# ZFP57 is a regulator of postnatal growth and life-long health

Geula Hanin [1], Boshra AlSulaiti[1], Kevin R. Costello[1], Hugo Tavares [1], Nozomi Takahashi[1], Liudmila A. Mikheeva[1], Anjuli Karmi Freeman[1], Shrina Patel [1], Benjamin Jenkins [2,3], Albert Koulman [2,3] & Anne C. Ferguson-Smith [1] ✉

Early-life factors, including postnatal nutrition, shape long-term health outcomes with epigenetic effects being implicated but poorly understood in the process. Zinc-finger protein 57 (ZFP57), is an epigenetic regulator of genomic imprinting, a process controlling gene expression based on parental origin with a vital role in prenatal growth. Here, we report an imprinting-independent function of ZFP57 in postnatal resource control via the mammary gland. ZFP57 influences multiple mammary gland phenotypes, including ductal branching and cellular homeostasis with its absence leading to significant differential gene expression related to alveologenesis and lactogenesis and altered milk composition. $Zfp57^{-/-}$ dams attenuate offspring growth, with impacts on offspring metabolic health; effects exacerbated when pups are raised by a dam of a different genotype than their birth mother. The study identifies ZFP57 as a major regulator of both pre and postnatal resource control in mammals, offering new insights into early factors influencing lifelong health.

Genomic imprinting is a mammalian epigenetic mechanism regulating gene expression according to parental origin[1]. It is predominantly controlled by DNA methylation, influencing multiple genes. While most studies established that imprinted genes regulate prenatal growth and development, evidence suggests that imprinting also supports postnatal development and maternal metabolism[2–6].

Both the establishment and maintenance of DNA methylation at imprinted and non-imprinted regions are highly regulated[1]. One of the key regulators is ZFP57, a master regulator of genomic imprinting required for normal development[7]. Ablation of $Zfp57$ causes loss of DNA methylation at imprinting control regions[8,9]. In mice, deletion of both the maternal and zygotic copies of $Zfp57$ early in development causes severe loss of DNA methylation at imprinted loci, resulting in embryonic lethality. In contrast, deletion of the zygotic $Zfp57$ copies only, is less severe causing partial neonatal lethality[7]. In humans, $Zfp57$ mutations are linked to multi-locus imprinting disturbances and transient neonatal diabetes mellitus[10].

The perinatal period significantly impacts offspring development. Various factors in early life can affect epigenetic status and gene expression, influencing lifelong health[11,12]. Over the years, imprinted genes have been linked to prenatal development[1] and nutrient transport via the placenta to the embryo[4,13]. However, maternal support of the offspring continues postnatally ensuring the survival of the young, manifested as maternal care and lactation. Some imprinted genes have been associated with these processes[2,3,14], and several are linked with adult metabolism[5,15,16]. Together, the perinatal and long-term effects of imprinting highlight it as a possible mechanism influencing lifelong health and disease risk.

The mammary gland, a crucial organ for lactation, develops during puberty and adulthood[17] and its proper development and functional differentiation are crucial for successful milk production[18]. During pregnancy, hormonal signals such as progesterone, prolactin and placental lactogen, drive mammary gland differentiation[19], which yields milk-producing alveoli. Breastmilk is essential for early offspring development and influences long-term health[20]. Breastfeeding is associated with a reduced risk of obesity and diabetes, diseases which are often associated with imprinted genes[5]. Some imprinted genes, including $Igf2$, $Grb10$ and $Mest$ were

[1]Department of Genetics, University of Cambridge, Cambridge, UK. [2]Core Metabolomics and Lipidomics Laboratory, Wellcome-MRC Institute of Metabolic Science, Addenbrooke's Treatment Centre, University of Cambridge, Cambridge, UK. [3]Wellcome-MRC Institute of Metabolic Science and Medical Research Council Metabolic Diseases Unit, Addenbrooke's Treatment Centre, University of Cambridge, Cambridge, UK. ✉e-mail: afsmith@gen.cam.ac.uk

previously implicated in mammary gland development and function[3,21,22]. Nevertheless, little is known about the role of epigenetic factors affecting lactation, postnatal growth and the developmental origins of health and disease.

In this work, we identify ZFP57 as a key regulator of postnatal resource control through the mammary gland, acting independently of its canonical imprinting role. We show that loss of ZFP57 alters mammary development, gene expression and milk composition, leading to impaired offspring growth and metabolic health, thereby linking this factor to both pre- and postnatal determinants of lifelong health.

## Results

### Zfp57 is expressed in adult somatic and mammary cell types

The key regulator of imprinting, *Zfp57*, mainly studied in the embryonic context[8,23,24], has been associated with multiple developmental processes[25,26]. To explore the relevance of *Zfp57* in post-mitotic tissues, we used RNA-seq to compare its expression across embryonic and adult mouse tissues. In embryos, *Zfp57* was highly expressed in various tissues, particularly in brain regions such as the neural tube, fore-, mid- and hindbrain. Other tissues such as the limb, heart and lung, also exhibited notable expression, while the liver showed the lowest expression (Supplementary Fig. 1A). *Zfp57* was highly expressed in the placenta and adult tissues such as the ovary, testis and brain regions, with lower expression in the lung, adipose tissue, kidney and mammary gland (Supplementary Fig. 1B). Experimental validation comparing embryonic levels to adult tissues confirmed high *Zfp57* expression in embryonic brain and cultured embryonic stem cells (ESCs). In adult tissues, *Zfp57* expression was reduced, though remained high in the ovary and lung, moderate in the spinal cord, heart and brain, with lower expression in the muscle, liver and kidney. It is

noteworthy that *Zfp57* expression is also low in the whole mammary gland (Fig. 1A).

Next, to compare the function of *Zfp57* in the embryo to the adult, we compared imprinted gene expression in wild-type (WT) and zygotic deletion of *Zfp57* (*Zfp57*[-/-]), a viable mouse model we previously described, which results in ZFP57 loss-of-function and partial loss of germline imprints[8,9]. Consistent with this, absence of *Zfp57* in embryonic brain affected the expression of several imprinted genes at E12.5, including *Zac1*, *Rasgrf1* and *Nnat* (Supplementary Fig. 1C).

Maternal endocrine effects impact lactation performance, directly affecting milk production and let-down through oxytocin, cortisol and prolactin[27,28]. The absence of imprinted gene expression in the hypothalamus and pituitary gland, where *Zfp57* levels are high, has been linked to impaired maternal care and reduced milk release[5,29]. To explore the role of ZFP57 in postnatal resource control, we quantified the expression of *Zfp57*-regulated imprinted genes[8] in WT and *Zfp57*[-/-] adult mice hypothalami and pituitary glands. In the hypothalamus, we observed changes in several imprinted genes regulated by *Zfp57*, including *Zac1*, *Rasgrf1*, *Snrpn* and *Nnat*, and a reduction in *Phlda2* levels (Fig. 1B). In the pituitary gland, both *Rasgrf1* and *Nespas* showed a similar pattern (Fig. 1C). Given that loss or partial loss of imprinting at most of the affected genes is found in *Zfp57* homozygous mutants, this indicates that imprinted gene regulation by ZFP57 is relevant in both embryonic and adult tissues. As imprinted genes are renowned for regulating foetal and postnatal growth, this led us to hypothesise that ZFP57 evolved to support pre- and also postnatal offspring growth and this might occur via the mammary gland.

To consider this hypothesis and to assess the contribution of *Zfp57* expression to the lactating mammary gland itself, we quantified *Zfp57* expression in five mammary cell populations from WT female mice on lactation day 2. We sorted basal cells, luminal differentiated

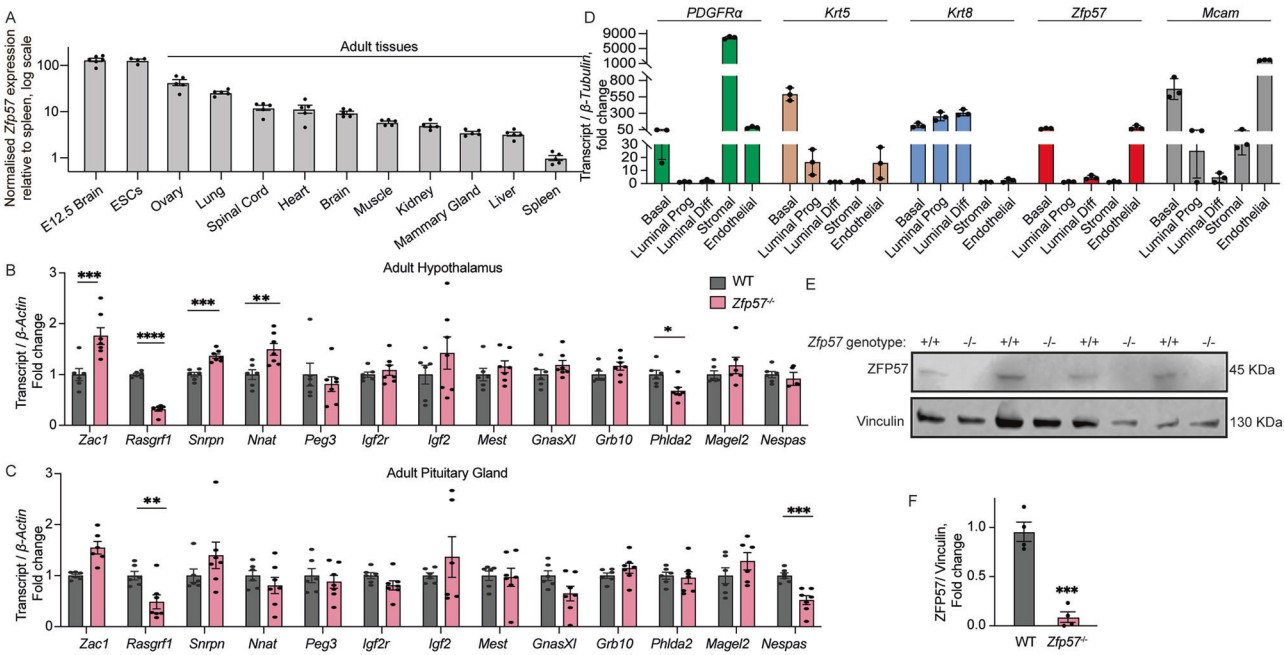

**Fig. 1 | ZFP57 is expressed in somatic cells and various mammary gland cell types throughout adulthood. A** Relative *Zfp57* expression determined by qPCR in embryonic brain, stem cells and adult mouse tissues from C57BL6/J mice. Values are normalised to *β-Tubulin*. n = 4 for ESCs, n = 5 for adult tissues, n = 6 for E12.5 brain. Data are mean ± SEM. Hypothalamic (**B**) and pituitary gland (**C**) expression levels of imprinted genes associated with ZFP57 or maternal behaviour and milk let-down in WT and *Zfp57*[-/-] mice (n = 6 for WTs, n = 7 for *Zfp57*[-/-]). Data are mean ± SEM, Two-tailed Student's *t*-test. Hypothalamus: *Zac1* p = 0.0038, *Rasgrf1* p = 8.6549E-09, *Snrpn* p = 0.00011, *Nnat* p = 0.0081, *Phlda* p = 0.0147; Pituitary gland: *Rasgrf1*

p = 0.0106, *Nespas* p = 0.00094. **D** PDGFRα, *Krt5*, *Krt8*, *Mcam* and *Zfp57* expression levels in sorted mammary cell types from WT mice at lactation day 2 (n = 3). Data are mean ± SEM normalised to *β-Tubulin* and displayed a fold change relative to the lowest expressing cell-type per transcript. Western blot (**E**) and quantification (**F**) of ZFP57 protein levels normalised to Vinculin in *Zfp57*[-/-] and WT mammary glands at lactation day 2 (n = 4 mice per genotype). Data are mean ± SEM, Two-tailed Student's *t*-test, p = 0.0010. *P < 0.05, **P < 0.01, ***P < 0.001, ****P < 0.0001. Source data are provided as a Source data file.

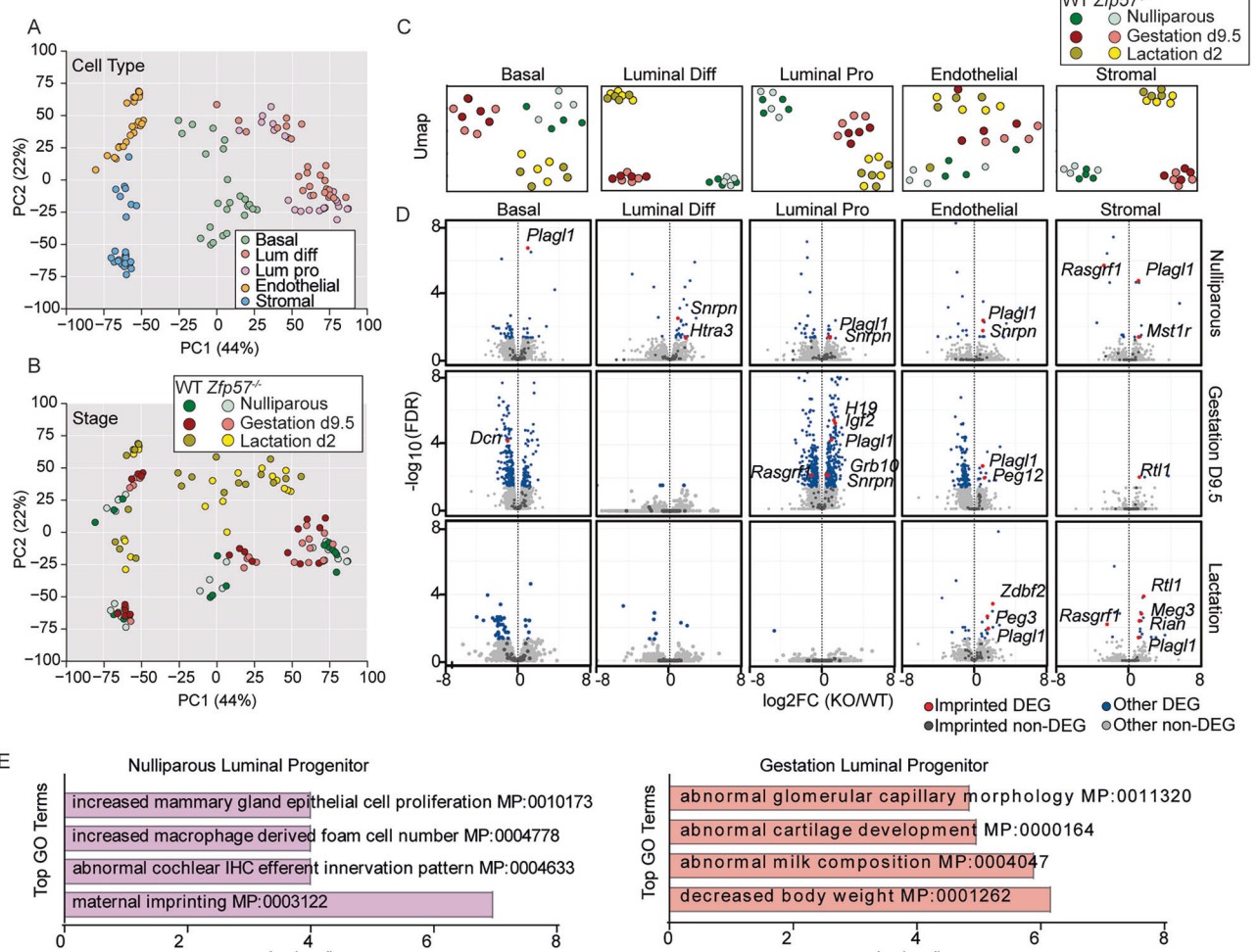

**Fig. 2 | Epigenetic and transcriptional profiling of mammary tissue in** *Zfp57⁻ᐟ⁻* **mice.** Principal component analysis (PCA) of mammary gland gene expression comparing *Zfp57⁻ᐟ⁻* and WT mice. Colours represent different purified cell-types (**A**) and developmental stage separated by genotype (**B**). **C** UMAP visualisation of cell-type specific RNA-seq libraries, coloured by stage and genotype. **D** Volcano plot representing differentially-expressed imprinted and non-imprinted genes in *Zfp57⁻ᐟ⁻* compared WT mice across 5 different cell populations and 3 developmental stages. Imprinted genes which display significant changes in expression between WT and *Zfp57⁻ᐟ⁻* animals are represented as red and labelled on the graph. All other differentially expressed genes are shown as blue. Differential gene expression is represented as -log10(FDR) plotted on the Y-axis, and Log2(FC) of normalised gene expression on the X-axis. **E** Pathway enrichment analysis of biological processes altered in *Zfp57⁻ᐟ⁻* luminal progenitor cells, in nulliparous and gestation time points. The top 4 GO terms are shown, arranged by their log(*p* value) performed in Enrichr using a one-sided Fisher's exact test with Benjamini-Hochberg correction. *n* = 4 mice/genotype cell type and stage.

cells, luminal progenitor cells, immune cells and endothelial cells (See methods). Supplementary Fig. 2A provides an overview of the gating strategy. *Zfp57* was expressed in basal and endothelial cells, co-expressing high levels of the markers *Krt5* and *Mcam*, respectively. Notably, luminal differentiated cells, luminal progenitors and stromal cells characterised by high expression of their respective markers *Krt8* (luminal) and PDGFRα (stromal), exhibit low levels of *Zfp57* expression (Fig. 1D). These findings suggest that ZFP57 may function specifically in basal and endothelial cells during adult mammary gland development.

To evaluate the presence of ZFP57 protein in mammary glands, we extracted protein from lactating mammary glands of both WT and *Zfp57⁻ᐟ⁻* mice. Western blot quantifications revealed presence of ZFP57 protein in WT mice albeit at low levels, and its absence in *Zfp57⁻ᐟ⁻* mice (Fig. 1E, F).

### ZFP57 shapes mammary development independently of imprinting

To decipher whether the absence of ZFP57 impacts gene expression in the adult mammary gland, we analysed 120 transcriptomes from 5 sorted cell populations isolated from 2 genotypes: WT and *Zfp57⁻ᐟ⁻* at 3 stages: nulliparous, gestation day 9.5, and lactation day 2. Characterisation of transcriptome-wide changes using principal component analysis (PCA) revealed that sample variation is primarily affected by cell type and stage but not genotype (Fig. 2A, B), indicating a lack of major transcriptional differences in *Zfp57⁻ᐟ⁻* glands and the general ability of the tissue to differentiate and function. Furthermore, cell-type-specific clustering showed that *Zfp57⁻ᐟ⁻* and WT maintain their clustering by stage. However, we could detect genotype-specific differences in luminal progenitors and endothelial cells during gestation and in stromal cells during lactation, consistent with the expression of *Zfp57* in the gland (Fig. 2C).

To study whether loss of *Zfp57* is associated with impaired genomic imprinting in the mammary gland, we plotted differentially-expressed genes (DEGs) between *Zfp57⁻ᐟ⁻* and WT mammary cell populations. Only 15 imprinted genes were differentially expressed in *Zfp57⁻ᐟ⁻* glands (Fig. 2D). Notably, none of the imprinted genes within the same imprinted control region showed reciprocal behaviour expected if imprinting were lost in the absence of ZFP57 (Fig. 2D,

Supplementary Fig. 3, Supplementary data 1). This indicates that perturbed expression levels of the 15 differentially expressed imprinted genes does not result from loss of ZFP57-mediated imprinting control.

Importantly, most of the DEGs were observed in luminal progenitor cells (Fig. 2D), which exhibit low expression levels of *Zfp57* (Fig. 1D). This suggests that the mammary gland phenotype observed may be an indirect effect or a downstream consequence of ZFP57 function loss at an earlier developmental stage or elsewhere. However, analysing the enriched pathways among the DEGs in *Zfp57*[-/-] luminal progenitors revealed associations with lactation. In particular, in the nulliparous stage, where among the top 4 GO terms was mammary gland epithelial proliferation. Similarly, during gestation day 9.5, when the highest number of DEGs are detected, the top 4 terms included milk composition and body weight (Fig. 2E, Supplementary data 2). Together, this implies that ZFP57 plays a role in regulating genes essential for normal mammary gland development and milk production independently of imprinted genes. This may be caused by functions of ZFP57 earlier in development and/or at distant sites regulating mammary gland physiology, such as the brain, liver and ovary. Further in-vivo analysis is required to elucidate these possibilities.

### ZFP57 affects mammary ductal branching during reproduction

To decipher whether transcriptional changes observed in *Zfp57*[-/-] mammary cells have functional consequences in the mammary gland, a tissue regulating nutritional resources postnatally, we focused on key developmental stages crucial for supporting offspring growth. We examined the architecture of nulliparous mammary glands, gestation days 4.5, 9.5, and 14.5, and lactation day 2. Comparing WT and *Zfp57*[-/-] revealed aberrant branching in *Zfp57*[-/-] mammary glands (Fig. 3A). Nulliparous *Zfp57*[-/-] mammary glands showed an increased number of branches and total secondary and tertiary branching points compared to WT controls, resembling early pregnancy glands in WT animals and consistent with the observed enrichment in the GO term 'mammary gland epithelial proliferation'. In contrast, on gestation days 4.5, 9.5 and 14.5, *Zfp57*[-/-] mice showed a significant reduction both in branches and in total secondary and tertiary branching points (Fig. 3A, B). This indicates that ZFP57 contributes to normal mammary branching and that its absence leads to precocious development, which could be a result of a local or systemic absence, potentially impacting tissue functionality. Interestingly, on lactation day 2, the gross appearance of the mammary gland seemed normal and fully populated with alveoli, suggesting potential compensation for the earlier defects and an ability to produce milk.

### ZFP57 regulates mammary cellular dynamics in vivo

Next, we tested the functionality of gland. Healthy mammary gland development relies on homeostasis between mammary cellular populations[30]. To investigate whether the aberrant branching in *Zfp57*[-/-] mice relates to abnormal homeostasis, we used flow cytometry to compare cellular proportions in *Zfp57*[-/-] and WT mammary glands. Loss of *Zfp57* resulted in a significantly imbalanced cellular composition throughout the mammary gland developmental cycle. *Zfp57*[-/-] glands had altered proportions of epithelial basal and luminal cells in nulliparous glands, at gestation day 9.5 and lactation day 2. Further quantification of luminal subpopulations revealed a higher fraction of differentiated luminal cells during gestation. During lactation, a significant proportion of them retained their progenitor state, potentially affecting tissue functionality (Supplementary Fig. 4A). Endothelial cells were decreased in *Zfp57*[-/-] glands compared to WTs in all stages (Supplementary Fig. 4B). The absolute cell numbers and changes per stage and genotype are presented (Supplementary Fig. 4C–E). These findings suggest that ZFP57 loss results in an imbalanced mammary cellular composition, particularly among epithelial cells.

Given the imbalance between proliferative and differentiated epithelial cells detected in flow cytometry, along with the enrichment of the GO term related to mammary gland epithelial proliferation, we immunostained mammary sections for the Ki67 proliferative marker. Quantification of Ki67-positive cells showed that *Zfp57*[-/-] nulliparous glands exhibited positive Ki67 cells in contrast to WT controls, consistent with the histological analysis. This difference was also observed on gestation day 9.5 but not earlier at gestation day 4.5 and lactation day 2, where no significant differences were detected (Fig. 3C, D and Supplementary Fig. 5). This supports the notion that premature proliferation occurs in *Zfp57*[-/-] nulliparous glands, when Ki67 is typically not detected, consistent with the previously observed transcriptional changes. This is followed by a subsequent burst of proliferation in mid-gestation, which serves as a compensatory mechanism for the reduced branching density.

Our data suggest that premature ductal proliferation occurs in mutants' nulliparous glands, followed by a significant decrease in ducts and branching points on gestation day 4.5, with a further burst of proliferation in gestation day 9.5. To explore whether the reduction in branching density is reflected at the molecular level, we performed TUNEL staining, revealing a significantly higher fraction of apoptotic mammary cells in *Zfp57*[-/-] compared to WT sections from nulliparous glands, through gestation day 4.5 and day 9.5 and lactation day 2. (Fig. 3E, F). Quantitative PCR analysis of differentiation- and lactation-related genes in mammary tissue confirmed that *Zfp57*[-/-] mice experience premature alveologenesis and lactogenesis in nulliparous glands, indicated by increased levels of *GATA3*, *Notch3*, *Elf5*, *Stat5a*, *Stat6*; while WT mammary glands typically exhibit these increases as they undergo alveologenesis and lactogenesis during pregnancy[31]. This was followed by a significant decrease in *GATA3*, *Elf5*, *Stat5a*, *Stat5b*, and *Stat6* transcript levels during gestation, concurrent with decreased expression of the milk genes *Csn2*, *Wap* and *Lalba*, contrasting with the expected increase in these transcripts during pregnancy. During lactation, most of these transcripts remain unchanged, except for *Notch3* and *Stat5b*, which remain lower compared to WT glands (Fig. 3G).

Together, our data show that loss of *Zfp57* results in precocious development of nulliparous mammary glands characterised by extensive secondary and tertiary branching, tissue proliferation and upregulated lactogenesis and alveologenesis-related genes. Moreover, during gestation, *Zfp57*[-/-] glands experience significantly decreased mammary ductal branching, extensive apoptosis and reduced activation of the transcriptional network that transforms the mammary gland into a physiologically-functional organ during lactation.

### Maternal *Zfp57* deficiency impacts birth outcomes

To assess the impact of the observed transcriptional and morphological aberrations in *Zfp57*[-/-] mammary glands on lactation and maternal nutritional resources, we monitored their offspring throughout their first weeks of life. We focused on two prominent mouse crosses: WT females crossed to a *Zfp57*[-/-] males, generating paternal heterozygous pups (*Zfp57*[+/-p]), or *Zfp57*[-/-] females crossed to WT males, generating maternal heterozygous pups (*Zfp57*[m/+]). Both crosses yield genetically identical offspring of a single genotype; however, they are epigenetically distinct[8,23]. Previous analysis has shown that while *Zfp57*[+/-p] have intact methylation imprints, the *Zfp57*[m/+] show a 20% reduction in the *Snrpn* methylation imprint[8]. However, their in-utero environment differs as they develop in either a WT or *Zfp57*[-/-] mother. The hypothalamus is involved in feeding circuitry and suckling[32], therefore, we next quantified changes in imprinted gene expression affected by *Zfp57* in this region. *Zfp57*[+/-p] offspring show no changes in hypothalamic imprinted gene expression compared to WT, while *Zfp57*[m/+] offspring exhibit increased hypothalamic expression of *Rasgrf1* and *Nespas* relative to both WT and *Zfp57*[+/-p] offspring. Expression of *Snrpn* and *Nnat* was higher in *Zfp57*[m/+] compared to WTs, and *Magel2* levels were higher in *Zfp57*[m/+] compared to *Zfp57*[+/-p] (Supplementary Fig. 6A). Partial loss of methylation may explain the increase in expression

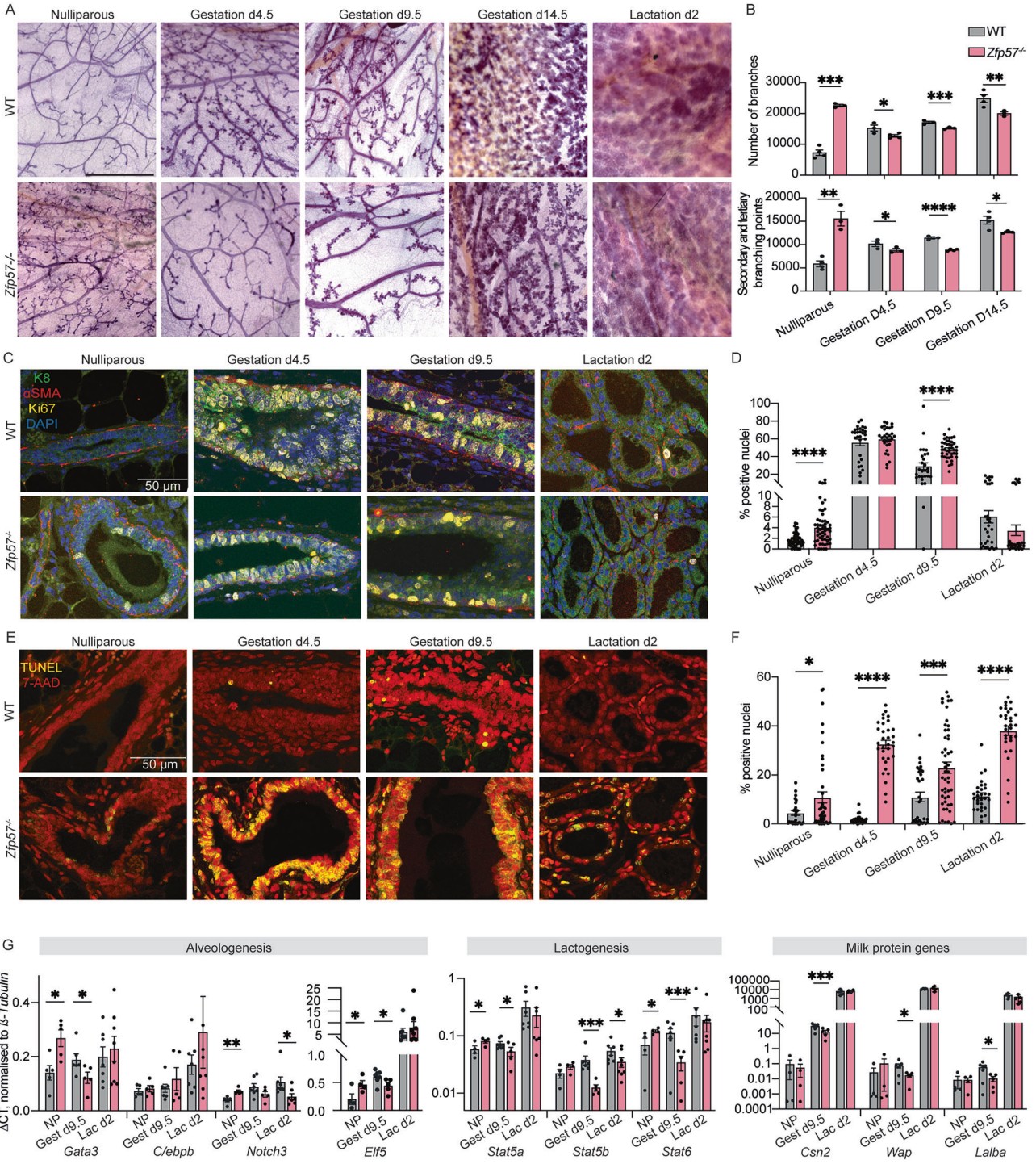

observed at the *Snrpn* gene in the *Zfp57*^m-/+ offspring. Next we explored offspring birth outcomes. *Zfp57*^-/- females did not display differences in pregnancy latency (Supplementary Fig. 6B), suggesting no major hormonal imbalance. This was supported by similar levels of circulating progesterone and oestradiol in *Zfp57*^-/- and WT nulliparous females (Supplementary Fig. 6C, D). *Zfp57*^m-/+ pups born to *Zfp57*^-/- dams showed a slight increase in birth weight (Fig. 4A). This could be attributed to longer gestation periods in *Zfp57*^-/- dams compared to WTs bred with *Zfp57*^-/- or *Zfp57*^+/- males (Fig. 4B). *Zfp57*^-/- dams generally had smaller litters compared to WT dams (Fig. 4C). This indicates that birth outcomes may be influenced by both the in-utero environment in *Zfp57*^-/- dams and intrinsic epigenetic differences of *Zfp57*^m-/+ pups. Notably, despite being slightly heavier, *Zfp57*^m-/+ pups born to *Zfp57*^-/-

dams displayed lower early postnatal survival during lactation compared to those born to WT or *Zfp57*^+/- dams (Fig. 4D). This could be explained by maternal or offspring factors. When we assessed milk secretion, we identified delayed appearance of milk spots in pups born to *Zfp57*^-/- dams (Fig. 4E). Additionally, impaired maternal behaviour, indicated by pup retrieval assay, may contribute to decreased pup survival (Fig. 4F). Nevertheless, our analysis did not reveal significant differences in pup contact, licking, or grouping within the nest (Supplementary Fig. 6E–I), except for a slight decrease in nest organisation score at P0 that did not persist (Supplementary Fig. 6E). Moreover, we found a minor decrease in pup body temperature at P1 and P2 which is unlikely to cause hypothermia (Supplementary Fig. 6I). Taken together, these findings reveal that *Zfp57*^m-/+ pups born to *Zfp57*^-/- dams

**Fig. 3 | *Zfp57^{-/-}* mice exhibit abnormal mammary gland morphology and cellular dynamics. A** Representative mammary gland whole mounts stained with Carmine Alum, from *Zfp57^{-/-}* and WT mice at nulliparous, gestation day 4.5, 9.5 and 14.5 and lactation day 2. Size bar represents 20 mm. **B** Quantified number of branches and secondary and tertiary branching points of mammary gland whole mounts. *n* = 3 (*Zfp57^{-/-}* nulliparous, gestation d14.5 and WT gestation d4.5); *n* = 4 (WT nulliparous, gestation d9.5, d14.5 and *Zfp57^{-/-}* gestation d4.5, d9.5). Number of branches: Nulliparous: *p* = 0.00106, gestation d4.5: *p* = 0.04, d9.5 *p* = 0.00015, d14.5 *p* = 0.0108. Secondary and tertiary branching: Nulliparous: *p* = 0.0020, gestation d4.5: *p* = 0.035, d9.5 *p* = 5.229E-05, d14.5 *p* = 0.025. Data are mean ± SEM, unpaired two-tailed t-test. Representative immunofluorescence (**C**) and quantification (**D**) of Ki67 in sectioned mammary glands from *Zfp57^{-/-}* mice compared with WT across nulliparous, gestation day 9.5 and lactation day 2 stages. KRT8 and αSMA served as luminal and basal markers, respectively. DAPI served as a nuclear marker. *n* = 3/ genotype and stage, 10 fields quantified/section. Nulliparous *p* = 1.80E-06; gestation d9.5 *p* = 1.88E-06. Data are mean ± SEM, unpaired two-tailed t-test. Representative images (**E**) and quantification (**F**) of apoptotic nuclei in *Zfp57^{-/-}* and WT mammary glands using TUNEL staining. 7-AAD was used as a nuclear marker. *n* = 5/

genotype for nulliparous; *n* = 3 for WT and *n* = 4 for *Zfp57^{-/-}* gestation d9.5; *n* = 3/ genotype for lactation d2, 10 fields quantified/section. Nulliparous: *p* = 0.035; gestation d4.5: *p* = 4.95E-25; d9.5 *p* = 0.0006; lactation d2 *p* = 2.41E-17. Data are mean ± SEM, unpaired two-tailed t-test. **G** Relative alveologenesis (left), lactogenesis (middle) and milk protein genes (right) related transcripts in whole mammary tissues from *Zfp57^{-/-}* mice compared with WT across nulliparous, gestation day 9.5 and lactation day 2 stages. For alveologenesis: *n* = 5 for nulliparous/genotype; *Zfp57^{-/-}* for gestation d9.5; *n* = 7 for WT gestation d9.5; *n* = 7 for WT lactation d2; *n* = 8 for *Zfp57^{-/-}* lactation d2. For lactogenesis: *n* = 4 for nulliparous/genotype; *n* = 5 for *Zfp57^{-/-}* for gestation d9.5; *n* = 7 for WT gestation d9.5; *n* = 7 for WT lactation d2; *n* = 8 for *Zfp57^{-/-}* lactation d2. Values are normalised to *β-Tubulin*. *Gata3*: NP *p* = 0.029; gest9.5 *p* = 0.031; *Notch3*: NP *p* = 0.005, lac d2 *p* = 0.017; *Elf5*: NP *p* = 0.033, gest9.5 *p* = 0.022; *Stat5a*: NP *p* = 0.037; gest9.5 *p* = 0.049; *Stat5b*: gest9.5 *p* = 0.005; lac d2 *p* = 0.046; *Stat6*: NP *p* = 0.034, gest9.5 *p* = 0.007; *Csn2*: gest9.5 *p* = 0.001; *Wap*: gest9.5 *p* = 0.050; *Lalba*: gest9.5 *p* = 0.050. Data are mean ± SEM. Two-way ANOVA with Tukey's corrections. *\*p* < 0.05, \*\**p* < 0.01, \*\*\**p* < 0.001, \*\*\*\**p* < 0.0001. Source data are provided as a Source data file.

---

show increased birth weight and poor perinatal survival compared to *Zfp57^{+/p}* pups born to WT dams.

To tease apart lactation-related maternal and offspring phenotypes, assess milk let-down and suckling, we monitored pup weight gain following separation from the dam. We compared WT and *Zfp57^{-/-}* dams raising their own *Zfp57^{m-/+}* and *Zfp57^{+/p}* biological litters, with dams of the same genotypes raising cross-fostered WT pups. We found that *Zfp57^{m-/+}* pups, but not WT pups, cross-fostered to *Zfp57^{-/-}* dam had delayed weight recovery when reunited with their *Zfp57^{-/-}* mothers (Fig. 4G). Throughout the experiments, pups were proximal to the nipple and often latched on, demonstrating no signs of delayed latching or lack of interest in suckling. This indicates that *Zfp57^{m-/+}* pups have impaired suckling.

### *Zfp57* loss alters milk composition offspring growth

To evaluate whether precocious development of *Zfp57^{-/-}* mammary glands impacts lactation and early postnatal offspring development, we monitored the growth of *Zfp57^{m-/+}*, *Zfp57^{+/p}* and WT pups raised by their *Zfp57^{-/-}* and WT biological mothers. Surprisingly, despite experiencing delayed milk spot appearance and a sucking defect, *Zfp57^{m-/+}* pups exhibited enhanced weight gain during lactation (Fig. 5A Supplementary Fig. 6K).

To distinguish the maternal-offspring in-utero effects from postnatal growth phenotypes, we conducted cross-fostering experiments with specific maternal and offspring genotype combinations. On postnatal day 1, WT, *Zfp57^{m-/+}* and *Zfp57^{+/p}* offspring were placed with either WT or *Zfp57^{-/-}* foster dams or kept with the biological mother. We assessed their weight gain during lactation and at weaning (Fig. 5B, Supplementary Fig. 7A–C). Both WT and *Zfp57^{+/p}* offspring showed similar growth and weaning weights when raised by, or cross-fostered to, a WT dam, indicating that the paternal *Zfp57* genotype did not influence postnatal growth trajectory and that being heterozygous per se did not influence weight outcome. *Zfp57^{m-/+}* pups exhibited significantly increased weight gain during lactation, compared to WT or *Zfp57^{+/p}* offspring, when raised by or cross-fostered to *Zfp57^{-/-}* dams. Surprisingly, regardless of their genotype, all the pups raised by, or cross-fostered to, *Zfp57^{-/-}* dams resulted in growth attenuation, with *Zfp57^{+/p}* pups exhibiting failure to thrive. This indicates that the *Zfp57^{-/-}* dam is compromised in her ability to support the normal postnatal growth of offspring.

The weight gain experienced by *Zfp57^{m-/+}* pups cross-fostered to *Zfp57^{-/-}* dams was exacerbated by cross-fostering to a WT dam, suggesting that maternal ZFP57 deficiency or exposure to a *Zfp57^{-/-}* in-utero environment predisposes these pups to excessive weight gain during lactation, which is further amplified if nursed by a mother of a different *Zfp57* genotype from their birth mother.

These experiments highlight the importance of concordance between gestational and nursing mother's genotype and emphasise the role of in-utero adaptation to postnatal resources provided by the birth mother. Notably, the failure of *Zfp57^{+/p}* pups to thrive was observed only when they were cross-fostered to *Zfp57^{-/-}* dams that differed in genotype from their birth mothers.

To further investigate whether the growth attenuation we observed in *Zfp57^{-/-}* relates to mammary gland function, we examined milk composition from *Zfp57^{-/-}* and WT dams and performed lipidomic analysis of milk collected on lactation day 8. We found that *Zfp57^{-/-}* milk contained higher oxidised triglycerides (36:0), and lower phosphatidylcholine levels (38:6 and 34:1) (Fig. 5C, Supplementary Data 3). Fewer phosphatidylcholines might impact phospholipid-coated lipid droplets size in milk and potentially body fat accumulation. Previous studies have indicated that dietary phospholipids provided to young mice can alter body fat accumulation[33]. Overall, this indicates that *Zfp57^{-/-}* mothers produce abnormal milk and this is associated with growth disparities observed in their nursing pups, regardless of their genotype. It also suggests that mother-offspring co-adaptation is associated with milk composition.

### Maternal *Zfp57* loss impairs offspring life-long health

Excessive childhood weight gain is a significant risk factor for cardiovascular and metabolic diseases[34]. To distinguish between pre- and postnatal influences on growth and metabolism, including the persistence or reversibility of the phenotypes into adulthood, we weighed *Zfp57^{m-/+}* and *Zfp57^{+/p}* offspring born to *Zfp57^{-/-}* and WT dams, alongside WT pups cross-fostered to *Zfp57^{-/-}* and WT dams, for 6 months.

We found that adult *Zfp57^{m-/+}* born to *Zfp57^{-/-}* dams continued to accumulate more weight than *Zfp57^{+/p}* from WT dams or WT offspring cross-fostered to *Zfp57^{-/-}* and WT dams, despite being fed the same chow diet and co-housed (Fig. 6A, B). Notably, the weight gain was exacerbated over time (Supplementary Fig. 8A). In contrast, *Zfp57^{+/p}* offspring, which initially failed to thrive when cross-fostered to *Zfp57^{-/-}* dams, caught up post-weaning, without long-term effects. At 6 months, both male and female *Zfp57^{m-/+}* offspring had higher fat mass, lower lean mass and a decreased lean/fat ratio compared to WT or *Zfp57^{+/p}* offspring (Fig. 6C, D, Supplementary Fig. 8B–D). Notably, *Zfp57^{m-/+}* offspring cross-fostered to WT dams exhibited significantly higher fat mass percentages than those raised by or cross-fostered to *Zfp57^{-/-}* dams.

Fasting glucose levels were higher in male, but not female *Zfp57^{+/p}* offspring cross-fostered to *Zfp57^{-/-}* dams compared to those raised by WT dams (Supplementary Fig. 8E, G). When challenged with intraperitoneal glucose tolerance test (IP-GTT), 'maladapted' *Zfp57^{m-/+}*

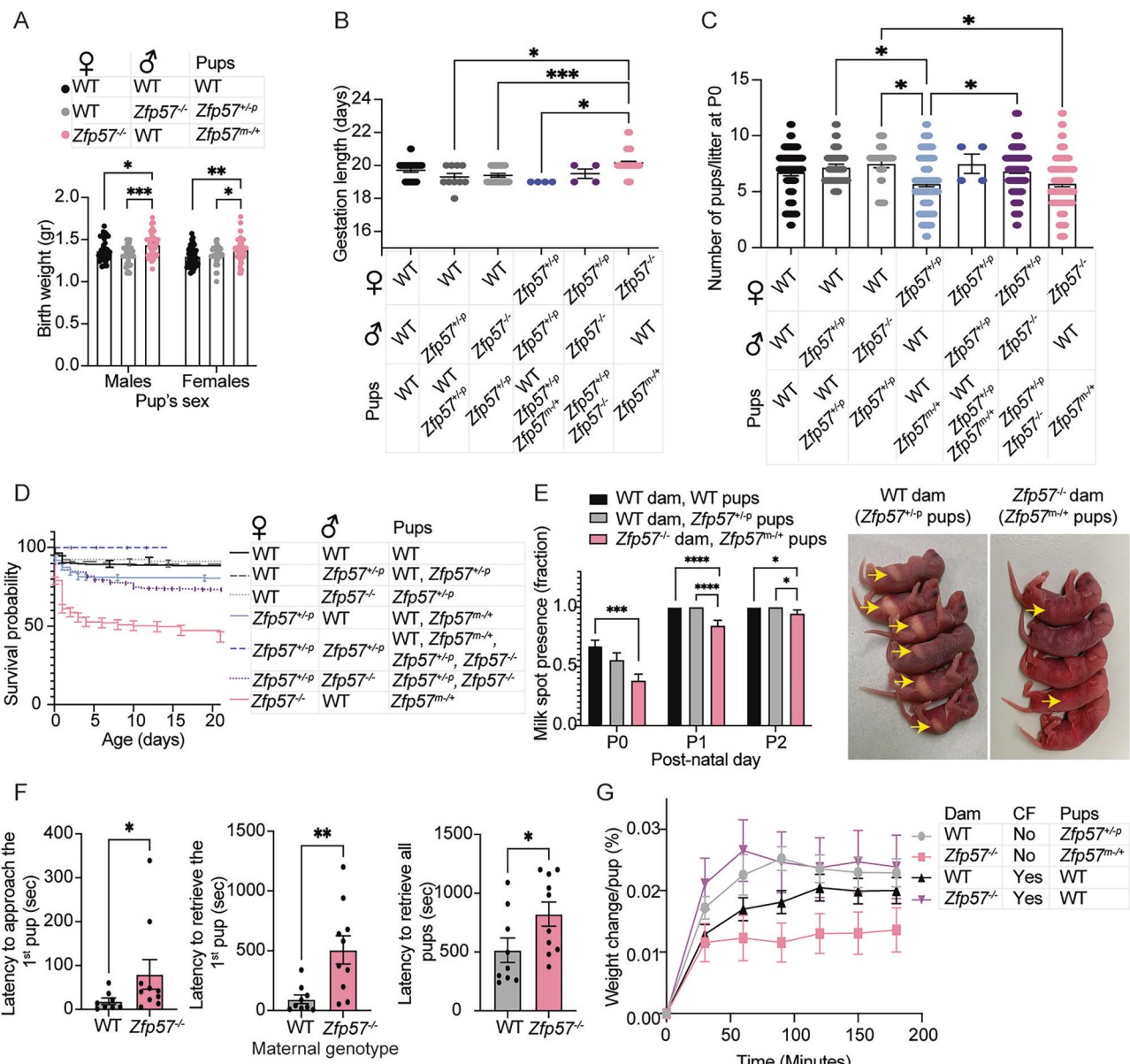

**Fig. 4 | *Maternal Zfp57 deficiency impacts* offspring birth parameters and suckling. A** Birth weight of heterozygous pups born to *Zfp57^-/-* or WT dam. WT: *n* = 36♂, 42♀ (13 litters); *Zfp57^+/-p*: *n* = 52♂, 42♀ (15 litters); *Zfp57^m-/+*: *n* = 58♂, 55♀ (16 litters). Males: *p* = 0.029 WT vs. *Zfp57^m-/+*: *p* = 4.004E-05 *Zfp57^+/-p* vs. *Zfp57^m-/+*, females: *p* = 0.007 WT vs. *Zfp57^m-/+*: *p* = 0.05 *Zfp57^+/-p* vs. *Zfp57^m-/+*. Data are mean ± SEM. One-way ANOVA with Tukey's corrections. **B** Length of gestation for each cross. *n* = 21 for WTxWT, *n* = 10 for WTx *Zfp57^+/-p*, *n* = 20 WTx *Zfp57^/-*, *n* = 4 *Zfp57^+/-p* x *Zfp57^+/-p* and *Zfp57^+/-p* x*Zfp57^/-*, *n* = 47 *Zfp57^/-* x WT. *p* = 0.014 WTx *Zfp57^+/-p* vs. *Zfp57^/-*xWT, *p* = 0.0006 WTx *Zfp57^/-* vs. *Zfp57^/-*xWT, *p* = 0.011 *Zfp57^/-* x WT.Data are mean ± SEM. One-way ANOVA with Tukey's corrections. **C** Number of pups per litter at postnatal day 0. *n* = 42 litters/WTxWT, *n* = 28/WTx *Zfp57^+/-p* *n* = 28/WTx *Zfp57^/-*, *n* = 70/*Zfp57^/-*x WT, *n* = 4/*Zfp57^/-*x *Zfp57^+/-p*, *n* = 83/*Zfp57^+/-p*x *Zfp57^/-* *n* = 54 for *Zfp57^/-*xWT. *p* = 0.045 WTx *Zfp57^+/-p* vs. *Zfp57^+/-p*x WT, *p* = 0.015 WTx *Zfp57^/-* vs. *Zfp57^+/-p*x WT, *p* = 0.026 WTx *Zfp57^/-* vs. *Zfp57^/-*xWT, *p* = 0.017 *Zfp57^+/-p*x WT vs. *Zfp57^+/-p*x *Zfp57^/-*. Data are mean ± SEM. One-way ANOVA with Tukey's corrections. **D** Kaplan–Meier curve of pups from different crosses, throughout lactation period. WTxWT *n* = 284, 42 litters, WTx *Zfp57^+/-p* *n* = 151, 28 litters, WTx *Zfp57^/-*, *n* = 158, 21 litters, *Zfp57^+/-p*xWT *n* = 407, 71 litters, *Zfp57^+/-p*x *Zfp57^+/-p* *n* = 15, 2 litters, *Zfp57^+/-p*x

*Zfp57^/-* *n* = 569, 83 litters, *Zfp57^/-*xWT *n* = 23, 6 litters. Log-rank test *P* < 0.0001 for *Zfp57^/-*xWT. **E** Quantification and representative pictures of pups with milk spots at postnatal days 0, 1, 2. *n* = 95 WT pups *n* = 82 *Zfp57^+/-p* pups, *n* = 80 *Zfp57^m-/+*, 13 litters/cross. WT vs. *Zfp57^m-/+* *p* = 0.0003 for P0, *p* = 4.65E-07 P1, *p* = 0.036 P2, *Zfp57^+/-p* vs. *Zfp57^m-/+* *p* = 4.62E-07 P1, *p* = 0.021 P2. Values are expressed as mean ± SEM. One-way ANOVA with Tukey's corrections. **F** Latency of *Zfp57^/-* and WT dam to approach and retrieve their pups. *n* = 65 pups from 10 litters for WT, *n* = 64 pups from 9 litters for *Zfp57^/-*. *p* = 0.025 for approaching the 1st pup, *p* = 0.0023 for retrieving the 1st pup, *p* = 0.034 for retrieving all pups. Values are mean ± SEM, two-tailed Student's *t*-test. **G** Pup weight gain following separation (time 0) and reunion with the dam. Maternal and offspring genotypes, as well as whether cross-fostering (CF) was performed, are indicated in the table. *n* = 32 pups from 6 litters for WT dam with *Zfp57^+/-p* pups, *n* = 20 pups from 5 litters for *Zfp57^/-*dam with *Zfp57^m-/+*pups, *n* = 37 pups from 5 litters for WT dam with WT pups, *n* = 28 pups from 6 litters for *Zfp57^/-* dam with WT pups. Values are mean ± SEM. Log-rank test. *P* < 0.05 values are mean relative concentrations (μM) ± SEM. *p < 0.05, **p < 0.01, ***p < 0.001. ***p < 0.001, ****p < 0.0001. Source data are provided as a Source data file.

offspring cross-fostered to WT dams performed worse than *Zfp57^m-/+* offspring raised by or cross-fostered to *Zfp57^/-* dams, with delayed normalisation of injected glucose, higher peak and endpoint glucose levels and a larger area under the curve (AUC) value (Fig. 6E,

Supplementary Fig. 8E–I). This suggests that *Zfp57^m-/+* offspring shows intensified glucose intolerance when fed by a mother carrying a genotype different from their birth mother and highlights co-adaptation as a mother-offspring compensatory mechanism.

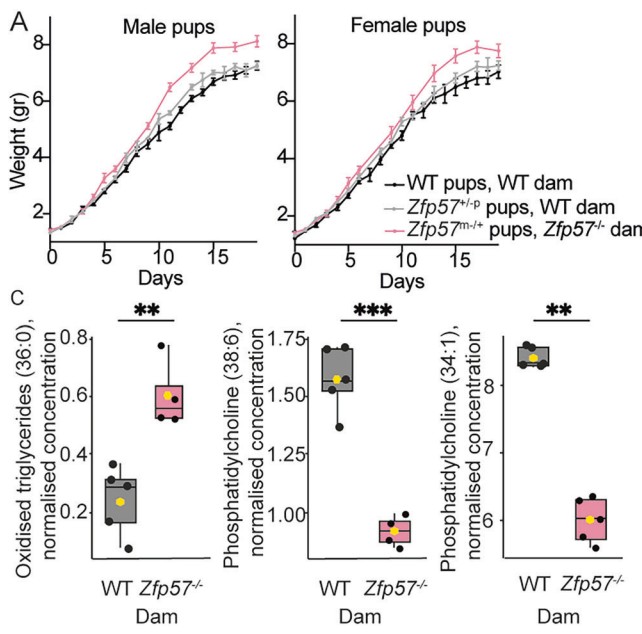

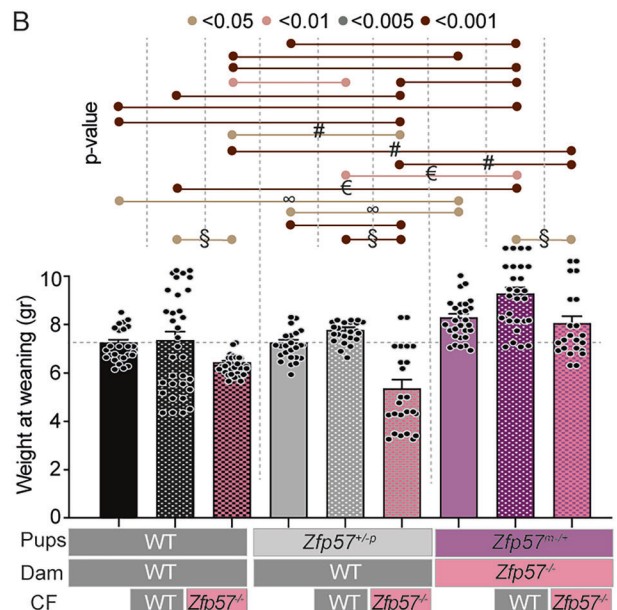

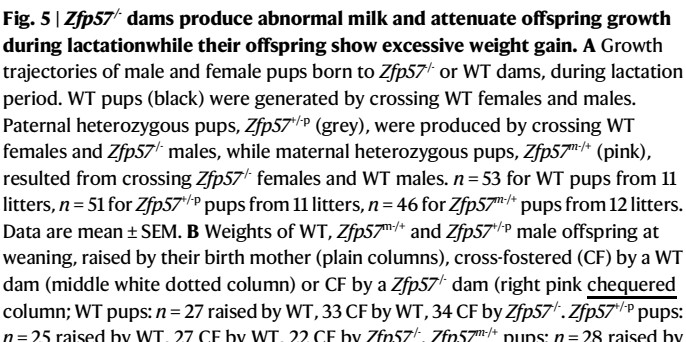

**Fig. 5 | *Zfp57⁻/⁻* dams produce abnormal milk and attenuate offspring growth during lactation while their offspring show excessive weight gain. A** Growth trajectories of male and female pups born to *Zfp57⁻/⁻* or WT dams, during lactation period. WT pups (black) were generated by crossing WT females and males. Paternal heterozygous pups, *Zfp57⁺/⁻ᵖ* (grey), were produced by crossing WT females and *Zfp57⁻/⁻* males, while maternal heterozygous pups, *Zfp57ᵐ/⁺* (pink), resulted from crossing *Zfp57⁻/⁻* females and WT males. n = 53 for WT pups from 11 litters, n = 51 for *Zfp57⁺/⁻ᵖ* pups from 11 litters, n = 46 for *Zfp57ᵐ/⁺* pups from 12 litters. Data are mean ± SEM. **B** Weights of WT, *Zfp57ᵐ/⁺* and *Zfp57⁺/⁻ᵖ* male offspring at weaning, raised by their birth mother (plain columns), cross-fostered (CF) by a WT dam (middle white dotted column) or CF by a *Zfp57⁻/⁻* dam (right pink chequered column; WT pups: n = 27 raised by WT, 33 CF by WT, 34 CF by *Zfp57⁻/⁻*. *Zfp57⁺/⁻ᵖ* pups: n = 25 raised by WT, 27 CF by WT, 22 CF by *Zfp57⁻/⁻*. *Zfp57ᵐ/⁺* pups: n = 28 raised by

KO, 27 CF by WT, 22 CF by *Zfp57⁻/⁻*. Symbols indicate significant comparison types: § Same pups fostered to different maternal genotypes, ∞ Non-cross-fostered pups, € Different pups fostered to WT dam, # Different pups fostered to *Zfp57⁻/⁻* dam. Exact p values provided as source data. Data are mean ± SEM. One-way ANOVA with Tukey's corrections. **C** Oxidised triglycerides (36:0) and phosphatidylcholine (38:6 and 34:1) in milk collected from *Zfp57⁻/⁻* and WT dams on lactation day 8. n = 5 for WT dam, n = 4 for *Zfp57⁻/⁻* dam. Box plots show the median (centre line), interquartile range (box), and whiskers extending to the minimum and maximum values, p = 0.007 for oxidised triglycerides (36:0), p = 0.0008 and p = 0.003 for phosphatidylcholine (38:6 and 34:1 respectively). Two-tailed Student's *t*-test. *p < 0.05, **p < 0.01, ***p < 0.001. ***p < 0.001, ****p < 0.0001. Source data are provided as a Source data file.

IP-GTT revealed intriguing dynamics in the 'maladapted' *Zfp57⁺/⁻ᵖ* offspring cross-fostered to *Zfp57⁻/⁻* dams, which failed to thrive as pups, with peak glucose levels 15 min post-injection, followed by a sharp decline. In contrast, *Zfp57⁺/⁻ᵖ* offspring raised by WT dams continually increased glucose levels up until 30- or 45-min post-injection, followed by clearance and normalisation (Fig. 6E). These findings suggest that early-life nutritional maladaptation has lasting metabolic consequences, independent of post-weaning weight normalisation. However, *Zfp57ᵐ/⁺* and *Zfp57⁺/⁻ᵖ* offspring performed similarly in an intraperitoneal insulin tolerance test (IP-ITT) (Supplementary Fig. 8J, K). Multi-parameter metabolic assessment (see Methods) revealed that *Zfp57ᵐ/⁺* offspring born to *Zfp57⁻/⁻* dams displayed a significantly higher fat oxidation (Fig. 6F) and lower carbohydrate oxidation (Supplementary Fig. 9A), despite similar levels of ambulatory activity and food intake (Supplementary Fig. 9B, C). Additionally, *Zfp57ᵐ/⁺* offspring born to *Zfp57⁻/⁻* dams showed lower respiratory quotient (Supplementary Fig. 9D), indicating a higher reliance on fat as an energy source compared to *Zfp57⁺/⁻ᵖ* offspring born to WT dams. While total energy expenditure was generally lower in *Zfp57ᵐ/⁺* compared to *Zfp57⁺/⁻ᵖ* offspring, this difference did not always reach statistical significance (Supplementary Fig. 9E). Notably, *Zfp57ᵐ/⁺* offspring cross-fostered to WT dams exhibited distinct metabolic differences from those raised by or cross-fostered to *Zfp57⁻/⁻* dams. During the dark period, *Zfp57ᵐ/⁺* offspring cross-fostered to WT dams (hence differing in genotype from their birth mothers) showed significantly higher fat oxidation compared to those raised by or cross-fostered to *Zfp57⁻/⁻* dams (Fig. 6F) and a markedly lower respiratory quotient in the dark (Supplementary Fig. 9D), indicating a metabolic shift associated with mismatched maternal genotype. These findings suggest that *Zfp57ᵐ/⁺*

offspring cross-fostered to WT dams (whose genotype was different from their birth mother) developed a poorer metabolic profile than those raised by *Zfp57⁻/⁻* dams, where the foster mother's genotype matched the birth mother.

*Zfp57⁺/⁻ᵖ* offspring cross-fostered to *Zfp57⁻/⁻* dams exhibited metabolic profiles including respiratory quotient, carbohydrate oxidation and total energy expenditure, similar to WT and to *Zfp57⁺/⁻ᵖ* offspring raised by or cross-fostered to WT dams (Supplementary Fig. 9A, D). However, they displayed increased fat oxidation during the dark period, unlike WT offspring cross-fostered to *Zfp57⁻/⁻* dams, but this was not associated with long-term metabolic effects (Fig. 6F, Supplementary Table 2). This suggests that, in contrast to offspring exposed in utero to an absence of maternal ZFP57, while *Zfp57⁺/⁻ᵖ* offspring faced significant early-life challenges, their later metabolic parameters largely normalise post-weaning.

Together, our findings indicate that *Zfp57ᵐ/⁺* offspring exposed prenatally to a *Zfp57⁻/⁻* maternal environment and epigenetically distinct from *Zfp57⁺/⁻ᵖ* offspring, undergo metabolic reprogramming, leading to long-lasting alterations in weight gain, body composition, metabolic rate and fat oxidation. In contrast, *Zfp57⁺/⁻ᵖ* and WT offspring do not show consistent long-term metabolic changes when cross-fostered to *Zfp57⁻/⁻* dams, despite transient alterations during lactation.

## Discussion
Our study identifies ZFP57 as a key modulator of postnatal nutritional resources, specifically affecting mammary gland development and milk composition.

Previous studies have suggested the importance of imprinting in the mammary gland[2], with some demonstrating functional roles for

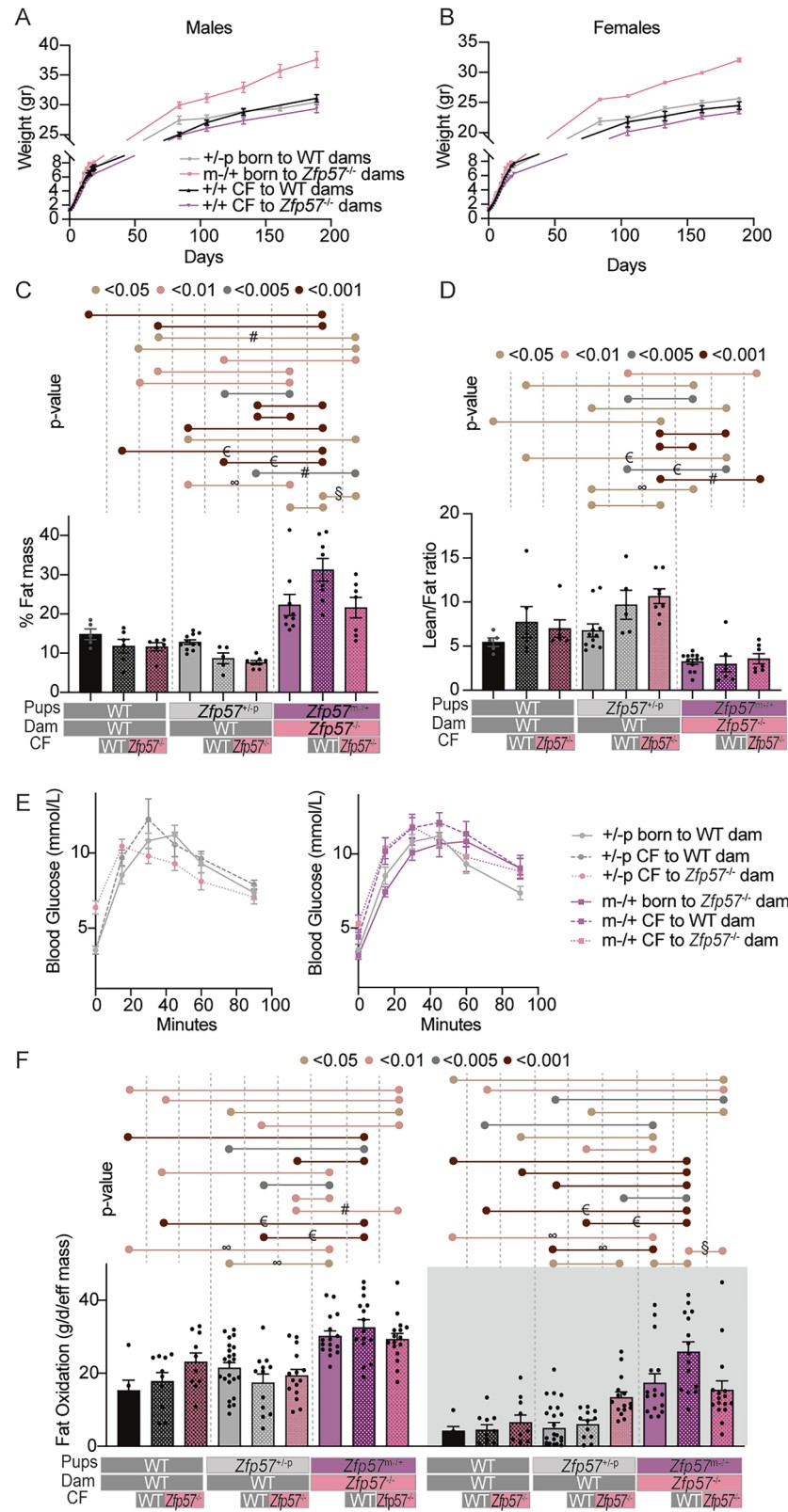

imprinted genes in this tissue[3,22]. However, while ZFP57 is a major imprinting regulator, our findings indicate that its role in the mammary gland is independent of imprinting mechanisms in the gland. This is evident from the absence of reciprocal changes in imprinted gene expression, such as increased *H19* and decreased *Igf2*. Instead of a complete loss of one gene and a doubling of expression in the other, both genes are upregulated. Moreover, only a small number of

imprinted genes display differential expression. Additionally, most of the DEGs were observed in luminal progenitor cells, which express very low levels of *Zfp57*. These observations suggest that ZFP57 is not acting directly in the adult mammary gland but influences the tissue indirectly, possibly through early developmental effects, transcriptional defects, inter-organ communication or secondary effects of other genes. Another possibility is that ZFP57 regulates stemness-like

**Fig. 6 |** *Zfp57*[m/+] **offspring born to Zfp57**[-/-] **dam present with long-lasting implications for energy metabolism, exacerbated by cross-fostering to a different maternal genotype.** Long-term growth trajectories of male (**A**) and female (**B**) *Zfp57*[+/p] and *Zfp57*[m/+] offspring born to *Zfp57*[-/-] or WT dams, and WT pups cross-fostered to *Zfp57*[-/-] or WT dams. *n* = 51 for *Zfp57*[+/p] pups (*n* = 27♂, 24♀) from 11 litters, *n* = 46 for *Zfp57*[m/+] pups (*n* = 21♂, 25♀) from 12 litters, *n* = 53 for WT pups CF to WT dam (*n* = 26♂, 27♀) from 8 litters, *n* = 49 for WT pups CF to *Zfp57*[-/-] dam (*n* = 24♂, 25♀) from 7 litters. Data are mean ± SEM. **C, D** Body composition assessment of WT, *Zfp57*[m/+] and *Zfp57*[+/p] male offspring in adulthood, raised by their birth mother (plain columns), cross-fostered (CF) by a WT dam (middle white dotted column) or CF by a *Zfp57*[-/-] dam (right pink chequered column), determined by TD-NMR and normalised to total body weight. **C** % Fat mass **D** Lean/Fat ratio. WT pups: *n* = 6 raised by WT, 6 CF by WT, 6 CF by *Zfp57*[-/-]. *Zfp57*[+/p] pups: *n* = 11 raised by WT, *n* = 5 CF by WT, *n* = 8 CF by *Zfp57*[-/-]. *Zfp57*[m/+] pups: *n* = 9 raised by KO, *n* = 8 CF by WT, *n* = 7 CF by *Zfp57*[-/-]. Exact *p* values provided as source data. Data are mean ± SEM. One-way ANOVA with Tukey's corrections. Symbols indicate significant comparison types: § Same pups fostered to different maternal genotypes, ∞ Non-cross-fostered pups, € Different pups fostered to WT dam, # Different pups fostered to *Zfp57*[-/-] dam.

**E** Intraperitoneal glucose tolerance test (IP-GTT) of male *Zfp57*[+/p] offspring (left) and *Zfp57*[m/+] offspring (right), raised by or cross-fostered to *Zfp57*[-/-] or WT dams. *Zfp57*[+/p] offspring born to WT dams is plotted twice for comparison. *Zfp57*[+/p] pups: *n* = 8 for pups born to WT, *n* = 6 for pups CF to WT, *n* = 7 for pups CF to *Zfp57*[-/-]. *Zfp57*[m/+] pups: *n* = 8 for pups born to *Zfp57*[-/-], *n* = 7 for pups CF to WT, *n* = 7 for pups CF to *Zfp57*[-/-]. Data are mean ± SEM. **F** Fat oxidation of male and female WT, *Zfp57*[m/+] and *Zfp57*[+/p] offspring at 6 months of age, raised by their birth mother (plain columns), cross-fostered (CF) by a WT dam (middle white dotted column) or CF by a *Zfp57*[-/-] dam (right pink chequered column), determined by the Promethion high-definition behavioural phenotyping system over 48 h period, dark period is represented by grey background. WT pups: *n* = 6 raised by WT, 10 CF by WT, 10 CF by *Zfp57*[-/-]; *Zfp57*[+/p] pups: *n* = 23 raised by WT, *n* = 12 CF by WT, *n* = 15 CF by *Zfp57*[-/-]; *Zfp57*[m/+] pups: *n* = 16 raised by KO, *n* = 15 CF by WT, *n* = 16 CF by *Zfp57*[-/-]. Exact *p* values provided as source data. Data are mean ± SEM. One-way ANOVA with Tukey's corrections. Upset plots below or above the bars represent *p* value based on colours *$p < 0.05$, **$p < 0.01$, ***$p < 0.001$. ****$p < 0.001$. Source data are provided as a Source data file.

properties in luminal progenitor cells during early development either intrinsically or systemically, supported by precocious proliferation occurring in nulliparous *Zfp57*[-/-] mammary glands. While *Zfp57* levels are low in these cells in adulthood, they exhibit the most pronounced gene expression changes, indicating a potential early-life influence. It is also worth considering that even lowly expressed genes can have regulatory effects. In this case, its absence may still influence basal or endothelial cells, where *Zfp57* is expressed at higher levels than other mammary cells. Conditional models of *Zfp57* can be used to decipher these possibilities, hence elucidating the potentially complex molecular mechanisms underlying the phenotypic outcome.

Analysing enriched pathways among the DEGs in mammary tissue highlighted top enriched terms related to mammary gland development, milk composition and body weight, features that are affected in *Zfp57*[-/-] dams or their offspring. This suggests that ZFP57 controls a specific set of mammary- and milk-related genes, consequently influencing these processes.

Our model suggests that ZFP57 regulates nutritional provision to offspring through two distinct mechanisms: prenatally, through genomic imprinting and postnatally through mammary gland and lactation-related genes.

A limitation of our study is the absence of Cre-line specifically deleting *Zfp57* in mammary cells. However, such a model will prevent us from examining the systemic physiological effects of ZFP57, since other vital organs that interact with the mammary gland during pregnancy and lactation, like the brain, pituitary gland, ovaries and placenta, will retain their *Zfp57* levels. *Zfp57*[m/+] offspring develop in *Zfp57*[-/-] dams and were thus exposed to an in-utero environment lacking maternal ZFP57. In addition to this maternal effect, these offspring also exhibit partial loss of DNA methylation at the *Snrpn* imprinting control region[8] and changes in gene expression[23]. Our data show that *Zfp57*[m/+] offspring have elevated hypothalamic expression of *Rasgrf1* and *Nespas*, but this is not due to loss of methylation imprinting and though they are involved in feeding circuitry, it may be a downstream effect. Therefore, while these gene expression changes are statistically significant, their causal role in the observed suckling defect remains uncertain. Nevertheless, some of the phenotypes observed in *Zfp57*[m/+] offspring likely reflect a combination of maternal in-utero environment and offspring-intrinsic epigenetic differences at *Snrpn*, distinguishing them from *Zfp57*[+/p] offspring. Our study challenges the prevailing notion of environmental co-adaptation between mother and offspring, revealing a genetic basis for this phenomenon. While most studies emphasise postnatal environmental exposures as determinants for lifelong health, our findings highlight an additional layer of genetically encoded regulation. We demonstrate that *Zfp57*[m/+] offspring, developing in a *Zfp57*[-/-] in-utero environment, have lifelong health implications, including higher birth weight, impaired suckling and enhanced early life weight gain, followed by metabolic syndrome hallmarks and obesity in adulthood. Additionally, we show that *Zfp57*[-/-] dams produce abnormal milk that attenuates offspring growth. Yet, later, their excessive weight gain is partially mitigated. This mitigation does not occur when they are cross-fostered to a WT dam.

The long-term implications observed in *Zfp57*[m/+] offspring further strengthen this observation, with *Zfp57*[m/+] cross-fostered to WT dams showing exacerbated metabolic syndrome hallmarks and obesity compared to when they are raised by their own or foster *Zfp57*[-/-] mothers. The differences are more pronounced during the dark phase when the animals are active. In contrast, *Zfp57*[+/p] offspring, experiencing a normal in-utero environment, show normal birth weight, suckling and weight gain, comparable to WT offspring. However, both WT and *Zfp57*[+/p] offspring show attenuated growth when nursed by a *Zfp57*[-/-] dam, with *Zfp57*[+/p] offspring failing to thrive during lactation. This suggests that the abnormal milk composition, characterised by increased oxidised triglycerides and reduced phosphatidylcholine, negatively impacts offspring growth, potentially in combination with altered maternal behaviour. Our findings demonstrate that the severe long-term health implications observed in *Zfp57*[m/+] offspring cross-fostered to WT dams are not seen in WT and *Zfp57*[+/p] offspring. Furthermore, the enhanced postnatal weight gain observed in *Zfp57*[m/+] pups appears to be an intrinsic feature and is attenuated when these pups are nursed by *Zfp57*[-/-] mothers. This supports the idea of maternal-offspring co-adaptation, whereby pups benefit more from milk produced by their biological mother or a dam sharing her genotype. Notably, in humans, *Zfp57* mutations are associated with transient neonatal diabetes. Further studies are required to determine whether *Zfp57*[m/+] pups exhibit a similar phenotype.

These findings highlight how prenatal developmental conditions can predispose offspring to long-term maladaptation, which can be exacerbated when postnatal nutrition resources are mismatched to their genetic background. This work has highlighted the synergistic nature of maternal genetic factors and postnatal nutritional influences.

The precise factors contributing to this genetic adaptation remain an open question. Exploring factors such as altered body composition in *Zfp57*[m/+] and failure to thrive in *Zfp57*[+/p], potentially relating to gene dosage and the genetics of the father's sperm, would be intriguing avenues for further investigation. Additionally, further investigation is required to determine whether cross-fostering itself alters milk composition and to delineate the specific impacts of milk components.

Overall, this study enhances our understanding of mother-offspring co-adaptation and its long-term effects on the life course. It sheds new light on the interplay between genetic factors and early-life nutrition, expanding our knowledge of this complex relationship.

## Methods

### Animals

Mouse work and the experiments for this study were approved by the University of Cambridge Animal Welfare and Ethical Review Body and performed under the UK Home Office Animals (Scientific Procedures) Act 1986. (Home Office project licence # PC213320E and PP8193772). *Zfp57* mutants[8] were backcrossed to 129aa background, a long term in-house colony of 129 Sv, for >30 generations and housed under a 12 h light/12 h dark photocycle 22 °C air temperature and 21% oxygen saturation with access to water and standard laboratory chow diet ad libitum (RM3, Special Dietary Services [SDS], Witham, UK) (11% fat, 7% simple sugars of energy contribution [%kcal]). Mutant lines are available from the authors upon request. Mice were mated at 12–16 weeks of age, embryos were collected at E12.5, and adult tissues, including the hypothalamus, pituitary glands and mammary glands, were collected from 12- to 14-week-old females to ensure a fully post-pubertal mammary gland. Adult hypothalamus and pituitary glands were collected from 6 animals per genotype, with 3 males and 3 females. Nulliparous mice were oestrus cycles matched and were dissected at oestrus. All *Zfp57*[-/-] dams used in this study were born in crosses of *Zfp57*[+/-] females x *Zfp57*[+/-] males, generating a zygotic knockout offspring, referred as *Zfp57*[-/-], which exhibits partial neonatal lethality as previously described[8]. In all experiments, adult female mice were euthanised by cardiac puncture followed by cervical dislocation, in accordance with approved institutional and national guidelines. Neonatal pups and embryos were euthanised by decapitation.

### Cross fostering

Heterozygous pups were generated by crossing *Zfp57*[-/-] female x WT male (producing maternal heterozygous offspring, referred to as *Zfp57*[m/+]) or WT female x *Zfp57*[-/-] male (producing paternal heterozygous offspring, referred to as *Zfp57*[+/p]) and cross-fostered within 24 h of each other to either *Zfp57*[-/-] or WT dams. In an additional experiment, WT pups from WTxWT matings bred alongside the experimental colonies were cross-fostered as control groups concurrently with the other experimental combinations. The original litter and the dam were removed from the dam's cage, and the cross-fostered pups were gently rolled in the nesting material of the fostering dam. Subsequently, the fostering dam was returned to her home cage with the new cross-fostered litter, which was weighed every 1–2 days until weaning. The number of pups was normalised to the size of the smaller litter.

### Suckling and milk let-down

Milk spot assessment was performed visually during weighing at postnatal days 0, 1 and 2 on litters with 5-6 pups. For suckling and milk let-down, pups and dams were separated at lactation day 8 for 4 h and were placed in proximity to each other in a heated cabinet. Following separation, the dam and pups were weighed using analytical scales immediately before their return to the nest (time 0), and at intervals of 30 min up to 3 h thereafter.

### Maternal behaviour

Dam behaviour was quantified in their home cage based on several parameters, including the quality of the built nest, scored 0-5 and performed on postnatal days 0–2. The rating score ranged from 0 to 5, with 0 being no nest and 5 being a neat nest with 90% of the nesting material used. For pup retrieval assay the mother and pups were removed from the cage without disturbing the nest, pups were weighed and scattered throughout the cage. The mother was then returned to the cage and her behaviour was recorded for 20 min. The time the dam took to approach and retrieve each pup back in the nest was recorded.

### Mouse milking

Mouse milking was performed on lactation day 8 on dams with at least 4 pups/litter. Pups and dams were separated for 4 h and placed in proximity to each other in a heated cabinet. Following the separation, dams were weighed and injected intraperitoneally with Ketamine (Ketavet, Zoetis, UK) 90 mg/Kg and Xylazine (Xylacare, Animalcare, UK) 10 mg/Kg in Saline, and 2 IU/Kg Oxytocin (sigma O4375-1000IU) in DDW. Milk was expressed manually using a gentle massage and collected using a P20 pipette.

### Multi-parameter metabolic assessment

TD-NMR (LF50H Minispec, Bruker, Coventry, UK) was used to determine longitudinal changes in body composition (lean and fat mass). Metabolic and activity profiles of the mice were assessed using the Promethion High-Definition Behavioral Phenotyping System (Sable Systems, Las Vegas, NV, USA) as described previously[35]. Data analysis was performed using ExpeDATA (Sable Systems, Las Vegas, NV, USA). The Respiratory quotient (RQ) was calculated as the ratio of $VCO_2$/$VO_2$. Total energy expenditure (TEE) was calculated as $VO_2$ x $(3.815 + 1.232 \times RQ)$, normalised to effective body mass calculated by ANCOVA, and expressed as kcal/h/kg eff. Mass. Fat oxidation (FO) was calculated as $FO = 1.69 \times VO_2 - 1.69 \times VCO_2$ and expressed as g/d/kg eff. Mass. Ambulatory activity and position were monitored simultaneously with the collection of the calorimetry data using XYZ beam arrays with a beam spacing of 0.25 cm.

### Glucose tolerance tests (GTT)

For intraperitoneal GTT (IP-GTT), mice fasted for 16 h and were then injected with 10% glucose (D-glucose, Sigma-Aldrich, MO, USA) at a 2 mg/kg dose in double-distilled water. Mice were bled from a tail clip. Blood glucose was measured before injection (time 0) and 15, 30, 60, 90 and 120 min post-injection using a handheld glucometer (Accu-Chek Performa, Roche, Mannheim, Germany). Blood glucose levels were compared between Heterozygous offspring born to either *Zfp57*[-/-] × WT (mat × pat) or WT × *Zfp57*[-/-] (mat × pat) crosses, at each time point using multiple unpaired t-tests corrected for multiple comparisons.

### Genotyping

*Zfp57*[-/-] mice were genotyped using the following primers: forward knockout, GAAAGTCCTGAATGCGTTGC, reverse knockout, GTGGGAAAGGGTTCGAAGTT, and *Zfp57* forward wildtype AGGACGTGGCAGTGTCTTTC and *Zfp57* reverse wildtype GACAAATGTCAGGTTCTTGAA. PCR product levels were then determined using agarose gel electrophoresis.

### ELISA

Circulating progesterone and oestradiol were determined in serum, using 17-beta-Oestradiol ELISA (Abcam ab108667), and progesterone ELISA (Enzo Life Sciences, ADI-900-001) according to manufacturer's instructions.

### Flow cytometry

Single-cell suspensions of mammary cells were prepared from WT or *Zfp57*[-/-] females at three different stages: nulliparous, gestation day 9.5 or lactation day 2. Pregnancy was confirmed during dissections. Mammary fragments were digested for 1 h at 37 °C using Krebs–Ringers – HEPES, 2.5 mM glucose (Sigma-Aldrich, D9434), 2% foetal bovine serum (Gibco, 16-000-044), 200 μM adenosine (Sigma-Aldrich, A9251), 1 mg/ml collagenase (Sigma-Aldrich, C2139), pH = 7.4. Further dissociation was performed using Trypsin-EDTA 0.25% (Gibco 25200072). Single cells were then filtered, and blocked with 10% normal rat serum (Sigma-Aldrich R9759-10ML) for 15 min at 4 °C and stained with the following antibodies: Anti-Mouse CD45 APC-eFluor780 (Invitrogen, 47-0451-82), Anti-Mouse CD31 PE-Cy7 (Invitrogen, 25-0311-82), Anti-Mouse TER-119 Biotin (Invitrogen, 13-5921-82), Anti-Mouse BP-1 Biotin (Invitrogen, 13-5891-81), Anti-Mouse EpCam Alexa Fluor647 (Biolegend, 118211), Anti-Mouse CD49b PE (Biolegend, 103506), Anti-Mouse CD49f Alexa Fluor488 (Biolegend, 313608).

Viability was assessed using Zombie Aqua dye (Biolegend 423101). We sorted basal cells (Lin$^-$CD31$^{neg}$CD45$^{neg}$EpCAM$^{lo}$ CD49f$^{hi}$), luminal differentiated cells (Lin$^-$CD31$^{neg}$CD45$^{neg}$EpCAM$^{high}$CD49f $^{low}$CD49b $^{low}$), luminal progenitor cells (Lin$^-$CD31$^{neg}$CD45$^{neg}$EpCAM$^{high}$CD49f $^{low}$CD49b $^{high}$), immune cells (CD45$^+$) and endothelial cells (CD31$^+$). The total number of sorted cells ranged between 350,000 on average for nulliparous glands and 750,000 for lactating glands. Subsequently, cell populations were sorted using BD ARIA II cell sorter (BD Biosciences). Data were analysed using FlowJo 10.8.1.

## RNA extraction

RNA was extracted from sorted cells or whole mammary gland tissue using miRNeasy micro kit (QIAGEN, 217084) or TRI reagent (Sigma-Aldrich, St. Louis, MO, USA), respectively, according to standard protocols. RNA concentration was determined using Qubit Fluorometer (Thermo Fisher), and RNA integrity was quantified using the 2100 Bioanalyzer instrument (Agilent).

## mRNA quantifications

mRNA levels were determined by quantitative reverse transcription PCR (RT-qPCR) using the RevertAid H Minus First Strand cDNA Synthesis kit (Thermo Scientific), followed by quantification in technical triplicates with Brilliant III Ultra-Fast SYBR® Green QPCR Master Mix (Agilent) on the LightCycler 480 Instrument (Roche). Relative expression was normalised to *β-Tubulin* expression unless otherwise noted and calculated using the ΔCt method.

Primers were designed using Primer3 software. Primer sequences are listed in Supplementary Table 1.

## Library generation and RNA sequencing

Libraries for RNA sequencing (RNA-seq) from sorted mammary cell populations, including basal, stromal, luminal differentiated, luminal progenitors and endothelial cells, were generated using SMARTer stranded total RNA-seq kit V2 – Pico input mammalian (TaKaRa bio, 634414) according to the manufacturer's instructions. Three developmental stages were selected, including nulliparous, gestation day 9.5 and lactation day 2, and included 8 animals/stage, 4 per genotype. The quality and RNA integrity number were assessed using Bioanalyzer 2100 (Agilent), Qubit fluorometer (Thermo Fisher) and sequenced using the NovaSeq 6000 system (Illumina).

RNA-seq processing pipeline was built using Snakemake v6.10.0[36]. Reads were trimmed using Cutadapt v3.4[37] with the following settings: Illumina TruSeq adaptors, -U4 --quality-cutoff 20, minimum length 50 base pairs. Read quality was assessed before and after trimming with FastQC v0.11.9 (Andrews 2010, http://www.bioinformatics.babraham.ac.uk/projects/fastqc). Further quality control metrics were obtained by aligning the reads to the standard GRCm38/mm10 mouse reference genome (Ensembl release 102) using STAR v2.7.9a[38], followed by RSeQC v4.0.0[39] for quality alignment control. Quality metrics were compiled using MultiQC v1.10.1[40]. Transcript quantification was performed using Salmon v1.5.0 in mapping-based mode[41]. Read count matrices were prepared using the R/Bioconductor package tximport v1.20.0[42]. DESeq2 v1.32.0[43] was used to detect differentially expressed protein-coding genes with at least 20 reads, false discovery rate (FDR) threshold of 0.05 and an absolute log2 fold-change threshold of 1.0. Differential expression gene analysis was performed separately for each cell type and stage. The DESeq2 package was also used to obtain normalised read counts using the variance stabilising transformation method.

UMAPS[44] were generated using normalised read counts obtained from DEseq2's 'variance stabilising transformation' method. UMAPs were made using 9 neighbours, with a min distance of 0.25, and using the Euclidean metric. Gene ontologies of differentially expressed genes were completed using Enrichr[45], and the top 4 most enriched terms were selected for visualisation.

## Milk lipidomic analysis

All solvents and additives were of HPLC grade or higher and purchased from Sigma-Aldrich (Haverhill, Suffolk, UK) unless otherwise stated.

The protein-precipitation liquid extraction protocol has been described previously[46]. Briefly, 10 μL of mouse milk was transferred into a 2 mL screw cap Eppendorf plastic tube (Eppendorf, Stevenage, UK). Immediately, 650 μL of chloroform was added to each sample, followed by thorough mixing. Then, 100 μL of the LIPID-IS (5 μM in methanol), 100 μL of the CARNITINE-IS (5 μM in methanol) and 150 μL of methanol was added to each sample, followed by thorough mixing. Then, 400 μL of acetone was added to each sample. The samples were vortexed and centrifuged for 10 min at -20,000 × $g$ to pellet any insoluble material. The supernatant was pipetted into separate 2 mL screw cap amber-glass auto-sampler vials (Agilent Technologies, Cheadle, United Kingdom). The organic extracts were dried down to dryness using a Concentrator Plus system (Eppendorf, Stevenage, UK) run for 60 min at 60 °C. The samples were reconstituted in 100 μL of 2: 1: 1 (propan-2-ol, acetonitrile and water, respectively) then thoroughly vortex. The reconstituted sample was transferred into a 250 μL low-volume vial insert inside a 2 mL amber glass auto-sample vial ready for liquid chromatography with mass spectrometry detection (LC-MS) analysis.

Full chromatographic separation of intact lipids was achieved using Shimadzu HPLC System (Shimadzu UK Limited, Milton Keynes, United Kingdom) with the injection of 10 μL onto a Waters Acquity UPLC® CSH C18 column (Waters, Hertfordshire, United Kingdom); 1.7 μm, I.D. 2.1 × 50 mm, maintained at 55 °C. Mobile phase A was 6:4, acetonitrile and water with 10 mM ammonium formate. Mobile phase B was 9:1, propan-2-ol and acetonitrile with 10 mM ammonium formate. The flow was maintained at 500 μL per minute through the following gradient: 0.00 min_40% mobile phase B; 0.40 min_43% mobile phase B; 0.45 min_50% mobile phase B; 2.40 min_54% mobile phase B; 2.45 min_70% mobile phase B; 7.00 min_99% mobile phase B; 8.00 min_99% mobile phase B; 8.3 min_40% mobile phase B; 10 min_40% mobile phase B. The sample injection needle was washed using 9:1, 2-propan-2-ol and acetonitrile. The mass spectrometer used was the Thermo Scientific Exactive Orbitrap with a heated electrospray ionisation source (Thermo Fisher Scientific, Hemel Hempstead, UK). The mass spectrometer was calibrated immediately before sample analysis using positive and negative ionisation calibration solution (recommended by Thermo Scientific). Additionally, the mass spectrometer scan rate was set at 4 Hz, giving a resolution of 25,000 (at 200 m/z) with a full-scan range of m/z 100 to 1800 with continuous switching between positive and negative mode.

Data processing—The instrument responses of the analytes were normalised to the relevant internal standard response (producing relative concentrations), these relative concentrations are corrected for the intensity for any extraction and instrument variations. Detailed methods for mouse milk lipidomic analysis are available in the Supplementary Information.

## Immunoblots

Samples were lysed in a solution containing 10 mM Tris HCl pH = 7.4, 1 M NaCl, 1 mM EGTA, and 1 % TX-100, and homogenised with a Kontes pellet pestle, incubated on ice for 10 min, centrifuged at 17,900 rcf, 4 °C, for 30 min, and the clear lysate collected. 40 μg protein samples were separated by standard SDS-PAGE procedures on 4–20% Mini-PROTEAN TGX gel (BioRad; 4561095) and ran at 90–130 V. After transfer to PVDF membrane (BioRad, 1704156EDU) using the BioRad TransBlot Turbo system (BioRad, 1704150ED)U, and blocking in 5% skimmed milk overnight at 4 °C, proteins were visualised using primary antibodies against ZFP57 (Abcam; ab45341; 1:250) and Vinculin (Abcam; ab129002; 1:5000) for 48 h at 4 °C. The membrane was washed with TBST for 30 min, followed by HRP-conjugated Goat anti-Rabbit secondary antibody incubation (Agilent Technologies; P044801-2; 1:10,000) for 2 h at RT, followed by a 30 min wash with

TBST. Membrane was developed with SuperSignal™ West Femto Maximum Sensitivity Substrate kit (Thermo Fisher; 34095) and exposed for 5 min on a Licor imager. Bands were quantified and analysed using densitometry with Fiji software. Original uncropped blots are provided as Source data.

## Immunohistochemistry
Paraffin slides were rehydrated by washing in xylene and decreasing ethanol concentrations in water. Heat-induced antigen retrieval was done by boiling slides for 10 min in 10 mM citrate buffer, pH 6. After washing, slides were incubated with 150 μl/slide of blocking buffer (4% horse serum, 0.05% TWEEN20 and 0.3% Triton X-100) for 60 min, followed by overnight incubation at 4 °C with primary antibody diluted in the blocking buffer. Slides were then washed and incubated with fluorescently- conjugated secondary antibodies for 2 h. Nuclear staining was done using 40,6-diamidino-2-phenylindole (DAPI, 5μg/μl for 5 min). Primary antibodies used were anti-K8 (DSHB; TROMA-I; 1:50), anti-αSMA (Novus Bio; NB300-978; 1:200), and anti-Ki67 (Cell Signalling; 12202; 1:200). Stained tissues were imaged using Leica SP8 confocal microscope.

Quantification was performed using Fiji software, using the free-drawing tool; ducts were circled and the intensity of Ki67 was measured, then normalised to DAPI.

## TUNEL staining
Mammary glands were fixed and sectioned at Nulliparous, gestation days 4.5, 9.5 and lactation day 2, and analysed for apoptosis using terminal deoxynucleotidyl transferase Br-dUTP nick end labelling (Abcam; ab66110) according to the manufacturer's instructions and using a 1:40 secondary antibody dilution. Z-stack images were taken on a Leica SP8 confocal microscope and analysed using FiJi software using the StarDist plugin to identify mammary ducts, calculating the percentage of BrdU intensity within the duct. Analysis included 3-5 animals per stage and genotype, and 27-60 fields per sample.

## Histology
Whole mounts of mammary tissues from nulliparous mice, gestation and lactation stages we fixed in 4% Paraformaldehyde at 4 °C, rinsed in Phosphate-buffered saline and stained overnight with carmine alum solution (Sigma-Aldrich C-1022), Aluminium Potassium sulphate (Sigma-Aldrich A-7167). Whole mounts were dehydrated and mounted for imaging using Zeiss Imager Apotome. Morphometric analysis was analysed using Fiji software, using the AnalyzeSkeleton plugin. Wholemount images were skeletonised and quantified. The total number of branches, as well as secondary and tertiary branching points were plotted.

## Statistics and reproducibility
Flow cytometry, milk lipidomic analysis and pup retrieval assay were performed blinded to genotype. All statistical analyses were performed using GraphPad Prism 8 Software. Statistical significance between two groups was determined by Mann-Whitney tests or two-tailed unpaired t-tests with Welch correction. Statistical significance between multiple groups was performed using one- or two-way ANOVA followed by Tukey's multiple comparisons tests, as appropriate. Differences were considered significant when *$P < 0.05$, **$P < 0.01$, ***$P < 0.001$, ****$P < 0.0001$. Data are presented as mean ± SEM. The number of samples or litter used for each experiment are indicated in figure legends.

Power calculations based on preliminary data were used to determine sample sizes. No data were excluded from the analyses. Samples were randomised for cross-fostering and metabolic testing. Flow cytometry, milk lipidomic analysis, and pup-retrieval assays were performed with investigators blinded to genotype during data acquisition and outcome assessment. All experiments were independently repeated with biologically distinct samples and representative results are shown.

## Reporting summary
Further information on research design is available in the Nature Portfolio Reporting Summary linked to this article.

## Data availability
The RNA-Seq data generated in this study have been deposited into the Gene Expression Omnibus database under accession number GSE248780. Lipidomics data has been deposited at MassIVE (https://massive.ucsd.edu) under the dataset identifier MSV000099723 [https://massive.ucsd.edu/ProteoSAFe/dataset.jsp?task=5aa276940c2047d1a12be0c4a529c5de]. All other data supporting the findings of this study are available within the article and its Supplementary Information files. Source data are provided with this paper.

## Code availability
The code has been deposited into Zenodo and can be accessed at https://doi.org/10.5281/zenodo.17454131.

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

## Acknowledgements

This work was funded by MRC grants MR/R009791/1 to A.C.F.-S. and MR/W003783/1 to A.C.F.-S. and G.H. was supported by the Royal Society Newton Fellowship and FEBS long-term fellowship. B.A. was supported by Qatar National Research Fund (GSRA8-l-1-0505-21030). The Cambridge NIHR BRC Cell Phenotyping Hub and the Cambridge Advanced Imaging Centre contributed support to the project. The authors thank Dr Leila Muresan for assisting in image quantification and analysis.

## Author contributions

A.C.F.S. and G.H. planned the experiments and interpreted the data. G.H. performed all the experiments, analysed the data, wrote the manuscript and prepared the figures. All authors contributed to the writing and editing of the manuscript. B.A. performed Tunel stainings, immunohistochemistry, confocal microscopy, image analysis, quantitative PCR and assisted with metabolic assays. K.R.C. performed RNA-seq analysis. H.T. performed RNA-seq differential expression analysis. N.T. collected and analysed embryonic imprinted gene expression. L.A.M. analysed expression in existing datasets. A.K.F. performed quantitative PCR, contributed to suckling assays and assisted with metabolic assays. S.P. assisted with RNA processing and library preparations. B.J. and A.K. analysed milk composition.

## Competing interests

The authors declare no competing interests.
