## [Transparent Peer Review file · Nature Communications]

ZFP57 is a regulator of postnatal growth and life-long health

Corresponding Author: Professor Anne Ferguson-Smith

Version 0:

Reviewer comments:

Reviewer #1

(Remarks to the Author)
Notes on ZFP57 manuscript

Very interesting premise, that contributes to our understanding of the interplay of genetics epigenetics in -utero and post-natal development. And the role lactation can play in this.
However, the current manuscript could lay out more clearly what are prenatal/in-utero programming effects and what are postnatal effect due to dam nurturing (maternal behavior and lactation)
General conclusion that both pup phenotype and dam phenotype are important seems valid, but the details are muddled in the reported analyses and it is really not clear what the effects of maternal postnatal nutrition/nurturing are vs the effect of the pups in utero exposure and genotype.

The manuscript could greatly benefit from focusing on this and using the relevant comparisons to draw conclusions.

Main concern

To address if the ZFP null dam has true intrinsic lactation defect one would have to use a conditional knock-out or transplantation experiment.

In the context of this manuscript at least all measurements should be performed comparing WT pups on WT dam and WT pups on ZFP58 null dams. This will still conflate maternal behavioral issues that might affect offspring's nutritional intake and mammary gland intrinsic defects, but at least one can say for sure that these effects are due to the dam in the postnatal window.

Furthermore, there is clear indication that the in-utero environment in the null dam affect live births and gestation duration and metabolic imprinting in the fetus.

Another consideration to take into account, all the null dams are presumably offspring of hemizygote or homozygote null dams so presumably been exposed to a ZFP57 +/- or -/- in utero and postnatal environment. The parental origin of null dams at least needs to be reported and taken into consideration for the interpretation.

From the presented cross fostering data it suggest that despite identified morphological histological and cellular differences identified in the null dams during mammary gland development and differentiation in pregnancy, and milk lipid content differences, this does not significantly affect growth of the WT offspring suckling the null gland. (fig 6 and fig s7 & S8)

Differences in growth of m-/+ could be because larger pups might provide a much stronger suckling stimulus of the gland and enhance lactation, WT glands can sustaining more growth of these pups. Is the % weight gain per pup relating to number of pups in the litter at weaning different for the m-/+ on WT vs on null dams? If the pup numbers become lower due to initial attrition the gland can sustain more demand.

Interesting to observe that the metabolic programming in m-/+ pups is radically different from p+/- pups, which only shows when nursed on ZFP57 null dams. Suggesting that the in-utero environment in ZFP 57 null mice is different? (as suggested by different birth weights and loner gestation)

m-/+ pups already heavier when born, grow heavier during nursing?

It seems that ZNP57 null dams attenuate growth of heterozygote pups (both m-/+ and +/-p), m-/+ do not grow as much on null as on WT dam, and the same is the case for +/-p offspring, while WT offspring is not affected by suckling on null dam.

Changed metabolism and signaling in the null dam resulting in different mammary gland development and milk composition probably also affects placental function or general metabolism during pregnancy.

Extended comments of figures and related text

Figure 1: and related text

Text relating to Figure 1d: "was expressed in basal and endothelial cells. However, the level of ZFP57 expression compared to the basal, luminal, and stromal markers Krt5, Krt8 and α SMA was lower. Notably, luminal differentiated, luminal progenitors and stromal cells exhibited the lowest expression (Figure 1D). "

This is confusing and it seems that the same is said 3 times, ZFP57 expression is only detected in basal and endothelial cells? Then authors go on to say that expression is lower than Krt5, Krt 8 and Sma, in which cell types? the basal cells or all cell types? Looks like it is high in the endothelial cells, compared to the cell type markers that are not expressed in the endothelial cells ... why no endothelial cell marker used? Then it seems that there is expression in the other cell types but very low.

Furthermore, in the current graph it is not possible to assess whether expression of some the genes is nearly at the same level as b-tubulin (or less than that 1-fold) in certain cell types, because some of the depicted genes are very highly expressed in the cell type

Firstly, I suggest grouping the graphs by gene, secondly use broken Y-axis if there is a large difference (and/or a logarithmic scale).

It is not clear what the "fold-change" relates to (expression relative to beta-tubulin? Or also relative to any expression of a gene in one of the celltypes?)

The legend and figure do not match up for 1E &F

Figure 2 and related text

A) Are the nulliparous wt and null mice in the same oestrus stage? And which one is that? Oestrus stage affects proliferation and apoptosis. Is it possible to assess alveolar development? (possibly from sections)

c) is there a difference in which cell type Ki67 is higher (basal vs luminal, lum prog?) or were you see the staining in ducts vs alveoli? There seems to be a difference in G9.5

D) What is quantified? Number of cells positive out of all cells (or epithelial cells) in the field? The positive K67 in the lactating ZFP null image looks like they are not in an alveolar structure.

E/F) same for TUNEL staining: is there a difference in which cell type TUNEL staining is higher (basal vs luminal, lum prog?) or just were you see the staining in ducts vs alveoli? What is quantified? Number of cells positive out of all cells (or epithelial cells) in the field?

G) is the expression of milk protein genes or fatty acid/ lipid synthesis affected?

Figure 3 and related text

The accompanying differential expression results should be shared as supplementary tables (DEG table and count tables) as well as the gene lists for go terms, (this can be retrieved from EnrichR using appyter).

Minimal expression of a gene does not mean that when it is knocked-out it will not have an effect, also besides the effect of knocking out ZFP57 in distant tissue that can affect mammary gland function, the cell-cell signaling within the mammary gland and the different cell types should be considered and discussed e.g. the possible effect of the the gene changes in the basal cells were Zfp57 is expressed at higher levels.

Figure 4 and related text

What are the authors trying to illustrate here? A defect in pups or in dams?

Prenatal (4A, B, C) indicate that there are prenatal/in-utero effects in pups born to null mothers, however that does is not addressing survival to birth of individual pup genotypes.

A) How does this compare to birthweight of WT born to WT ? It would be helpful to put this in context of weight of WT pups born to WT dams.

B/C) for null dams longer gestation and lower # of pups at p0 indicates that pups are dying in-utero. Is there a genotype difference in pups that survive to birth?

However, the studies shown related to postnatal development D and E conflate in-utero environment, postnatal behavior, mammary gland/lactation phenotype and pup genotype.

These should be properly cross fostered to address which pup phenotypes are due to mammary gland/ maternal behavior phenotypes/genotypes, first comparing WT dams and ZFP^{-/-} dams feeding WT pups. Also, there are obvious maternal behavior effect. How can one distinguish those from possible lactation defects?

D/E) are these nursed on their birth mothers?

D) conflates in-utero environment, with dam behavioral and lactation phenotypes as well as pup genotypes. Is there a difference in mortality between pup genotypes when there is more than one genotype in the litter?

E) Again, dam genotype and pup genotype conflate. Also are there still the same number of pups in the null dam litter at P2? If not the lack of difference can be because the demand is down and now the impaired gland can provide enough milk, assuming there is no effect of pup genotype on their ability to suckle. would be helpful to put this in context of Wt pups on

WT dams and WT pups on null dam.

G) By using WT pups the possibility of suckling issues based on genotype or imprint type could be ruled out.

H) is the relative weight gain significantly different? These graph indicate increased body weight but they start out heavier.

“Overall, this indicates that ZFP57^{-/-} mothers produce abnormal milk associated with growth disparities observed in their nursing pups, ultimately resulting in enhanced weight gain”

However, this seems to affect them^{-/+} pups. as is shown in subsequent cross fostering experiment the m^{-/+} pups also gain more weight when suckling a wild type dam. While there is no significant effect on growth of WT pups on null dams. If this is due to milk composition you would expect to see the same effect in WT pups on null dams. This does not add up and illustrated the importance of cross fostering studies.

Figure 5 and related text

There are clear metabolic differences between the offspring here but again the comparison of maternal +/- and paternal +/- on null dam vs WT is not a valid comparison, as it seems that the genotype and paternal inheritance (in-utero environment) affect the offspring's metabolism and growth.

To determine if the abnormal milk composition has an effect same pups should be assessed on different dams, preferably WT pups on WT and null dam.

To assess if the genotype of the offspring further affects the metabolic outcome later in life m^{-/+} and WT pups suckled on ZFP57 null dams should be compared.

Most of fig 4 and 5 is not as relevant as cross fostering experiments to really contribute to the understanding of what is due to pup genotype, in-utero environment, postnatal nutrition and nurturing. Fig4 A-C (with wt data included in A) show in-utero effects. Rest of these figures and experiments are too conflated to go beyond concluding there is “an effect” of deletion of ZNF57 and can be deleted IMO.

The cross-fostering experiments later described are better in enabling to start distinguishing between the different effects deletion of ZFP57 has and in which paternal context and interaction with dam and offspring. I do not see how most of what is show in fig 4 & 5 contributes I suggest to leave this out to keep a somewhat complicated concept due to possible parental & genotype effects and pre- and post-natal effects more to the point

Supplemental 5

5l) Should at least also be assessed using WT pups on WT dams and m^{-/+} pups on WT dams to determine what is due to genotype (and in utero environment) of the pups and what due to maternal postnatal factors (behavior, lactation)

Supplemental 6

A) Same as for 5l) (should include WT-WT and m^{-/+} pups on WT dams) also does this include all genotypes of pups born to WT? (why is the color in males different?)

B/C) Same as for 5l)

Figure 6 (supplemental fig 7 & 8) and accompanying text.

The interpretation of effect of suckling a genotypically different mother from it own genotype is per-se not incorrect but I think it can be brought back to what happens in utero and how the null dam rears the offspring postnatally, as mentioned earlier.

“These experiments highlight the importance of concordance between gestational and nursing mother's genotype and emphasises the role of in-utero adaptation to postnatal resources provided by the birth mother. “ This seems an overstatement

“Additionally, ZFP57^{m-/+} pups which showed the tendency to gain excessive weight when raised by ZFP57^{-/-} dams, exhibited greater weight gain when raised by a WT dam, which differed in genotype from their birth mothers” the gain weight in both cases just not as much on the null dams (just like the +/-p pups).

It is unclear from the graphs in fig6 if the metabolic analysis for the different offspring are different please show the AUC numbers in figure 6 not in supplemental. Also suggest including the growth curves in figure 6 and not the supplemental. Are the growth curves significantly different, can they be expressed as % weight gain? What is the effect of pup number in a litter on weight gain? The body condition and metabolic measures would greatly benefit from the inclusion of WT on WT data to know what “normal” is.

Discussion

“These findings indicate that ZFP57^{m-/+} offspring cross-fostered to WT dams whose genotype was different from their birth mother, showed an inferior metabolic profile than those raised by ZFP57^{-/-} dams whose genotype matched their birth mother” because they became heavier pre-weaning due to “better postnatal” environment?

“Our study identifies ZFP57 as a key modulator of postnatal nutritional resources, specifically affecting mammary gland development and milk production.” This only affects offspring if the offspring exposed to it is metabolically imprinted in utero due to epigenotype and maternal in utero environment.

Methods:

Minor notes

Animals: it is much appreciated that the nulliparous animals were oestrus matched, but it would be helpful to mention at which stage as the stage affects the morphological appearance of the gland and thus the interpretation of the null phenotype.

For the cell sorting, only used collagenase without any other means of dissociation of cell clusters you usually obtain only using collagenase? Did the authors assess the number of single cells obtained

Litter size matters in the stimulation of milk production, a smaller litter on a not optimally functioning gland can still do well, while a large litter would show lactation defects. What were the litter sizes?

Cells sorting: Only collagenase used to dissociate cells, no further dissociation, with trypsin or otherwise? Seems unusual as most often another dissociation step or enzyme is used to obtain single cells.

“Enrichment was calculated as per cent input” >>percent input

Reviewer #2

(Remarks to the Author)

Zfp57 is a regulator of postnatal growth and life-long health

The work presented explores the function of Zfp57 in mice focusing on the relationship between dams and their pups. Considerable work has been undertaken and there are some interesting phenotypic findings presented. The authors' main conclusion is that "Zfp57 functions in postnatal resource control via the mammary gland" with "life-long impacts on offspring metabolic health". Due to the complexity of the study and some lack of information in introduction and methodology, it is challenging to determine whether the authors interpretations are supported.

The authors state in their introduction "In mice, deletion of the maternal gene in oocytes and the zygotic copies in early embryos causes severe loss of methylation at imprinted loci, resulting in embryonic lethality" – this needs considerable additional detail since the study uses Zfp57^{-/-}, Zfp57^{m/+} and Zfp57^{+/-p} mice.

Interpretation of growth dynamics

The key issue lies with the use of genetically identical heterozygous offspring (Zfp57^{m/+} or Zfp57^{+/-p}) which may not be phenotypically identical. This is acknowledged in the discussion. The data presented could be interpreted to demonstrate that Zfp57^{m/+} or Zfp57^{+/-p} pups are phenotypically different with many of the findings due to these intrinsic differences, and not driven by resources from the mutant dam.

"Zfp57^{m/+} pups exhibited enhanced weight gain during lactation" and "Overall, this indicates that Zfp57^{-/-} mothers produce abnormal milk associated with growth disparities observed in their nursing pups, ultimately resulting in enhanced weight gain." The authors are suggesting that the milk quality is responsible for weight gain. However,

- 1) The Zfp57^{-/-} mutant dams have longer gestations (Fig 4B) so their pups will be heavier at birth as reported in Fig4A
- 2) The Zfp57^{-/-} mutant dams have smaller litters (Fig 4C) again consistent with heavier birthweight as reported in Fig4A
- 3) The pups are all mutant. The authors assume that Zfp57^{m/+} offspring and Zfp57^{+/-p} offspring are identical apart from the genetic status of the mother but - unless I have misunderstood the experimental design - Zfp57^{+/-p} offspring come from sires with 100% loss of Zfp57 in sperm and Zfp57^{m/+} offspring come from dams with 100% loss of Zfp57 in oocytes – could this impact gene expression in these offspring? They may be genetically identical but Zfp57^{m/+} and Zfp57^{+/-p} offspring could have differences in gene expression as a result of loss of Zfp57 in the respective parental germlines – which itself would be an interesting finding. Has this possibility been systematically excluded?
- 4) Fig4H – the growth dynamics of Zfp57^{m/+} and Zfp57^{+/-p} offspring are different but this could be interpreted to mean that Zfp57^{m/+} pups have intrinsically enhanced weight gain and/or Zfp57^{+/-p} pups have restricted weight gain as there is no WT control group

Similarly, it is not possible to conclude "Together, our findings indicate that early-life exposure induces metabolic reprogramming in Zfp57^{m/+} offspring, leading to long-lasting alterations in weight gain, body composition, metabolic rate and fat oxidation." Without a WT group, these findings in Fig5 could be interpreted in a number of ways

The critical experiment to tease apart the relationship between maternal Zfp57 and offspring Zfp57 is the fostering experiment. The authors have done well to illustrate the experimental design. It is, however, a complex experiment.

If I have interpreted Figure 6A and B correctly (and I may have this wrong)

Zfp57^{m/+} pups are generally heavier than fully WT

Zfp57^{m/+} pups fostered to WT dams are the heaviest

This could indicate an intrinsic growth advantage somewhat lost when the dam is mutant

Zfp57^{+/-p} pups are lighter when fostered to Zfp57 mutant dams but not when raised by WT dams – suggests Zfp57 mutant dams fail to provide sufficient nutrients.

The simplest interpretation is that loss of Zfp57 in the female germline results in growth advantage for pups and loss of Zfp57 in the male germline results in growth disadvantage for pups – most evident when their mothers are also mutant

WT pups raised by hom dam do not gain more weight excluding a role for maternally-driven weight gain.

Again, I may have misunderstood the experimental design.

Some information is missing from M&M for this work. M&M states “Heterozygous pups born to either Zfp57^{-/-} x WT (mat x pat) or WT x Zfp57^{-/-} (mat x pat) crosses within 24 hours of each other were used for cross-fostering” so where do the WT pups originate? More detail required. Was the WT colony bred and weighed alongside the experimental colonies? Although not indicated as significant, the overall WT control weight data appears lower than – for example - Zfp57^{+/-} pups with WT dams.

Also, not clear why only data for male pups presented?

Maternal behaviour

M&M and Figure 4F.

The details for assessment of maternal behaviour are insufficient and the indicated reference 2 is “Stringer, J. M., Suzuki, S., Pask, A. J., Shaw, G. & Renfree, M. B. Selected imprinting of INS in the marsupial. *Epigenetics Chromatin* 5, 14 (2012)” which is obviously incorrect. Detailed methods should be presented especially as the tests undertaken are not the standard ones done to assess maternal behaviour.

“Latency of Zfp57^{-/-} and WT dam to approach and retrieve their pups. n=64-65 pups from 9-10 litters” The data presented is for individual pups. This is not how data is analysed. It looks like a specific retrieval test was not undertaken. It would still be possible to present data for the 9-10 dams – ie latency to sniff 1st pup, latency to retrieve 1st pup and time to retrieve all pups to nest – assuming litter sizes comparable.

Introduction

“deletion of the maternal gene in oocytes and the zygotic copies in early embryos causes severe loss of methylation at imprinted loci, resulting in embryonic lethality. Deletion of the zygotic Zfp57 copy causes partial neonatal lethality⁷” I am struggling with the sentence – can the authors rewrite

Mammary development – when does the mammary gland start to differentiate? What hormonal signals? More detail required.

“little is known” Review all the other imprinted genes implicated in mammary gland development/function?

Results

Zfp57 - Non italicised capitals suggest the authors are referring to the protein. But, for example, the expression data is mRNA. Can the authors use Zfp57 (lower case and italics) when referring to mouse gene/mRNA product - and all mouse genes/mRNAs.

“In adults, Zfp57 was highly expressed in organs such as the placenta,” – the placenta is not an adult organ

The authors state “lower expression in adult somatic tissues including the lung and mammary gland” and then later “The placenta and mammary gland share similar functions, supporting offspring growth through nutritional resource control⁶. This led us to hypothesise that Zfp57 evolved as an upstream regulator supporting pre- and postnatal offspring growth.” This doesn't work logically. The authors can justify their focus just by saying imprinted genes are renowned for regulating fetal and postnatal growth raising the possibility that a master regulator might similarly influence these same parameters.

Similarly “Maternal behaviour impacts lactation performance and involves imprinted gene function in the hypothalamus and pituitary gland^{15,26}” Not clear what the authors mean here? In what way does maternal behaviour impact lactation performance?

“involves imprinted gene function in the hypothalamus and pituitary gland^{15,26}” Again, not clear. What do the authors mean by “involves”?

“we quantified the expression of Zfp57-regulated imprinted genes⁸ in adult mice hypothalami and pituitary glands” Male? Female? Add Zfp57-deficient or Zfp57^{-/-}

“a reduction in Phlda2 levels which may affect maternal care²⁷” why would reduced Phlda2 in adult mice hypothalami and pituitary glands impact maternal care?

“To investigate Zfp57 expression in the developing mammary gland” technically it has already developed by lactation day 2 (which I presume is postnatal day 2?).

“To evaluate the presence of Zfp57 protein in mammary glands” – did the authors use, for example, IHC or in situ to examine sites of expression?

“gestation days 4.5, 9.5, and 14.5,” When referring to events in the dam, it is usual to state whole days ie gestational days 5, 10 and 15.

The authors should provide weight data for mutant whole mammary glands as a proportion of total body weight.

“This indicates that Zfp57 contributes to normal mammary tertiary branching and that its absence leads to precocious development, potentially impacting tissue functionality” The authors need to specify that this phenotype could be due to either local deficiency or deficiency at another site. Needs some careful wording here.

“Quantification of Ki67-positive cells showed that Zfp57^{-/-} nulliparous glands exhibited positive Ki67 cells in contrast to WT controls, consistent with histological analysis. During gestation days 4.5, 9.5 and lactation day 2, no significant differences were observed (Figure 2C-D), further indicating premature proliferation in Zfp57^{-/-} nulliparous glands” This sentence needs to be worded more carefully ie are there only Ki67 +ve cells in mutant at G15 (14.5)?

“we analysed transcriptomes from 120 sorted cell populations at various stages: nulliparous, gestation day 9.5, and lactation day 2” – why not 4.5 and 14.5 to be consistent?

“Importantly, most of the DEGs were observed in luminal progenitor cells (Figure 3D), exhibiting minimal expression of Zfp57 (Figure 1C).” Not clear what the authors mean here?

“We found that Zfp57^{m-/+} pups had delayed weight recovery when reunited with their Zfp57^{-/-} mothers compared to Zfp57^{+/-} pups with their WT mothers.” Difficult to interpret as the genotype of both dams and offspring is different.

“receiving suboptimal maternal care” The authors do not show suboptimal care. Just delayed retrieval - which is a different factor. Remove.

M&M

129aa background – can the authors be more specific? Is this a long term “in house” colony? If not, state supplier

Mice were mated at 12-16 weeks of age – any particular reason for not mating earlier

Milk let down on postnatal day 9 /mouse milking on lactation day 8 – can the author settle on one term

Relative expression was normalised to β -Tubulin expression – can the authors confirm that only one reference gene was used? Has this gene shown to be stably expressed during mammary gland development and lactation? Cite reference
Supplementary Figure 1C Presume the axis is log scale – please add to label

Serum steroid hormones – why gestational d9.5 (also should be labelled gestational D10 or embryonic day 9.5) and lactation d2?

Supplementary Figure 5E – does the significance remain after testing for multiple measures made in P0,P1 and P2?

Reviewer #3

(Remarks to the Author)

Hanin et al present an interesting manuscript investigating the role of ZFP57 to regulate postnatal growth and life-long health. It is known that early life factors, such as pre- and post-natal nutrition, alter long term health via epigenetic mechanisms. ZFP57 is an epigenetic regulator of genomic imprinting with a known role in prenatal growth, and here they identify an imprinting-independent function of ZFP57 in postnatal control via the mammary gland.

The authors allude to a role for ZFP57 to regulate maternal care – could that be primary or secondary to the effects on offspring nutrition? This could be of interest to further elucidate the effect of functional changes in ZFP57.

Are there any other differences in the ZFP57^{-/-} mice that could contribute to their mammary gland development? Is their milk production normal? Is this a secondary effect on offspring development?

The cross-fostering data is interesting and clearly presents a role for the milk/mammary gland to contribute to offspring development. The authors also present a last metabolic phenotype in the offspring – what is the mechanism for this? Changes in body weight appear to contribute, as do altered fat oxidation in the offspring – is the fat different? This could be an important addition as it could allude to changes in adipose tissue function which could help further define the mechanism for ZFP57 to affect postnatal development.

Reviewer #4

(Remarks to the Author)

In the manuscript entitled, ZFP57 is a regulator of postnatal growth and life-long health, the authors present an intriguing discovery on the Kruppel-associated box-containing zinc-finger protein (ZFP57). ZPF57 is a master regulator of imprinting that acts to recruit DNA methyltransferases that lack the targeting domains required to recognize imprinting control regions. This is a confusing manuscript to read and not targeted to a general scientific audience as it lacks sentences that introduce the experiments conceptually and sentences that synthesize the potential meaning of the results.

Although ZPF57 is a master regulator of imprinting in the embryo, the authors find that it functions in an imprinting-independent fashion in the adult mammary gland. And, while there are a host of phenotypic changes in the mammary gland (involving alveologenesis), ZPF57 is expressed only in basal epithelial and stromal cells, even though it appears to function

by regulating the number and the function of alveolar progenitor cells (or at least luminal progenitor cells as shown here) -- a conundrum. Following the phenotypic analysis, the authors present a series of mating and cross-fostering experiments that further demonstrate how complex the consequences of ZFP57 loss is. Altogether, the data from cross-fostering suggest that concordance between gestational and nursing mothers' genotypes is important in the circumstance of ZFP57 loss and support the notion that there is in-utero adaptation to postnatal resources provided by the dam. This is an interesting message but the studies in the second half of the manuscript provide little insight into the studies in the first half. The question remains; how does ZFP57 mechanistically function in the mammary gland? How does it transcriptionally regulate mammary- and milk-related genes that govern nutritional provisioning? The authors do little to synthesize the manuscript's message, offering a one sentence paragraph in the discussion about the model. The bottom line is that this is an interesting study, but it remains preliminary and more experiments that address the mechanism of ZFP57 action are required. Below are more detailed comments on the figures.

Figure 1 starts out with data better shown in a supplementary figure -- RT-pPCR panel of imprinted gene expression in WT and ZFP57^{-/-} post-mitotic tissues. We learn that the absence of ZFP57 in the embryonic brain affected the expression of several imprinted genes at E12.5, including *Zac1*, *Rasgrf1* and *Nnat* (Figure 1A). But does it affect genes (e.g. *Igf2* and *H19*) in the reciprocal fashion typical of imprinting (*H19* expression is not examined)? Why is it important that both *Rasgrf1* and *Nespa5* showed a similar pattern? Does this suggest that in these tissues ZFP57 is not functioning in its canonical role as a master regulator of imprinting? Is this the conclusion? I ask because the authors simply state that the data show that ZFP57 regulates the expression of genes in adult tissues, which is a modest conclusion and why I suggest either the data belong in a supplementary figure or the authors interpret the data in a way that moves this manuscript's conclusions forward. In this figure they also FACS purify subpopulations of mammary epithelial cells to look at ZFP57 expression and find it expressed only in stromal and basal epithelial cells. A western of whole LD2 mammary glands reveals protein in WT but not KO tissue.

Figure 2 shows a phenotypic analysis of the mammary gland over the time course of gestation. It was initially surprising to me that the ZFP57^{-/-} dams could be evaluated since I understood from the literature that *Zfp57*-null mice exhibit embryonic lethality and loss of imprinting at many loci (Li et al., *Dev Cell*, 2008). But, it appears the loss-of-function effect is variable. Does this variable effect influence the interpretation of phenotypes in ZFP57^{-/-} adults?

The data show that nulliparous ZFP57^{-/-} glands are more branchy. N=3 MGs were quantified for the analysis but is this phenotype observed in every surviving ZFP57^{-/-} dam (i.e. what is the penetrance)? I note that the primary ductal structure in the ZFP57^{-/-} gland is similar to WT and what is being observed is an increase in secondary/tertiary branching. The authors should clarify their quantification. The reason the distinction is important is that the primary ductal structure is generated by endbud bifurcation and a defect in this would indicate a different mechanism than a defect in secondary/tertiary branching. To me, the precocious secondary/tertiary branching suggests precocious alveologensis or animals in diestrus (the methods say nulliparous glands were estrus matched, an important point). It is unclear how duct number was quantified and how this would differ from a branch point analysis. Interestingly, the phenotype is switched at pregnancy with ZFP57^{-/-} glands becoming less "branchy" than WT; it appears that there is almost no tertiary budding in the KO at this timepoint. However, the KO gland catches up over time.

Next the authors FACS purify cell populations at 3 timepoints and show a number of things including that the KO glands contain more luminal progenitor cells at LD2. To be clear, luminal progenitors are not alveolar progenitors, which can be distinguished by cKit expression (Shore et al., *Plos Genetics*, 2012); the analysis of AVPs may have provided better insight into the observed phenotype. The authors immunostain for Ki67 and obtain results that correspond to the phenotypic images: more proliferation in the KO in nulliparous glands, less at PD4.5, more at PD9 and no difference between genotypes at LD2. The issue here is that the authors quantify Ki67 by relative intensity...meaning that they are quantifying the intensity of staining across the nuclei (as shown by DAPI) rather than the proliferative index (% positive nuclei), which is the standard. It is true, there appears to be graded staining at PD4.5, maybe justifying an intensity approach to quantification. There is recent data to suggest that a cell's Ki67 expression level represents both its phase in the cell cycle and its cell cycle history (Miller et al., *Cell Reports* 24, 7/2018), but interpreting grades of Ki67 requires a more rigorous analysis. The authors should either apply this rigor or only count the percentage of strongly positive nuclei. Also intensity is spelled wrong on the Y-axis. This criticism applies to the TUNEL analysis as well, where the standard is the percentage of positive cells (not TUNEL area/duct area). Here, the immunostaining is heterogeneous, perhaps due to background because I also see staining around nuclei rather than in just in nuclei; a nuclear stain (DAPI) would be helpful for interpretation. The authors perform RT-qPCR and see increases in markers (e.g. *GATA3*, *Elf5* etc) consistent with premature alveologensis in the KO animals.

In Figure 3 the authors present transcriptome analysis from 120 sorted cell populations. Please make it clearer what "120 sorted populations" represent (from the methods: "Libraries for RNA sequencing (RNA-seq) from sorted mammary cells" which does not provide detailed information. How many animals at what stages and genotype? In any case, this analysis showed that variation is primarily affected by cell type and stage but not genotype. But they do see genotypic separation in luminal progenitor during gestation. What do the authors make of this observation? And is there genotypic separation in genes that regulate milk protein and milk lipid expression or just in genes that can be imprinted?

To evaluate whether ZFP57 is functioning in its capacity as a master regulator of genomic imprinting, differentially expressed genes (DEGs) were examined. Blue dots (other DEGs) apparently represent any significantly regulated gene (either a clearer explanation or the graphs clearly delineating significance would be helpful). But there is a difference between those imprinted genes that are differentially expressed and not. There are only 15 DEG imprinted genes in the KO and none of those in the same imprinted control region show reciprocal behavior, which suggests that ZFP57 is not functioning in its capacity as a master regulator of genomic imprinting. Most of the DEGs were observed in luminal progenitor cells and the authors link this to a potential role for ZFP57 in early developmental stages. Are the authors suggesting that ZFP57 functions in its role as master imprint regulator at earlier

stages and this is responsible for the postnatal phenotype? What do the authors mean by “early developmental stages”-- in the embryonic mammary gland? This is interesting and may hold the key to understanding ZFP57 function, but the GO terms are all about lactation so the manuscript is back to the investigating the phenotype. The authors conclude that loss of ZFP57 generates precocious development of the mammary gland independent of its role as an imprinting regulator and in the discussion, they state that their data suggest “ZFP57 is not acting directly in the mammary gland.” But intrinsic/extrinsic to the mammary gland is relatively easy to test because the mammary gland can easily be transplanted (Lawson DA, et al., Cold Spring Harb Protoc. 2015 Dec 2;2015(12). One can also test whether hormones(extrinsic) are responsible by ovariectomy followed by estrogen/progesterone treatment. A major deficiency in the manuscript is that this is where mechanism is dropped since “extensive in-vivo analysis is required to elucidate these possibilities.” But, these analyses should be done. Furthermore, if the authors find that ZFP57 is exerting its role through regulating gene transcription extrinsically through hormonal control, then ZFP57 ChIPseq, which has been performed by this group (Shi H, et al., Epigenetics Chromatin. 2019 Aug 9;12(1):49. doi: 10.1186/s13072-019-0295-4), could potentially be used in the pituitary gland to understand ZFP57 in this context. Using these types of in vivo analyses to investigate ZFP57 control of postnatal gene expression in luminal progenitor cells would achieve a more mechanistic understanding of the role of ZFP57 in mammary alveologenesis and a more impactful study.

Instead, the authors continued their analysis of ZFP57 by pursuing crosses that are typically used to understand imprinting. WT females crossed to a ZFP57^{-/-} males, generated paternal heterozygous pups (ZFP57^{+/-p}), and ZFP57^{-/-} females crossed to WT males generating maternal heterozygous pups (ZFP57^{m/+}). This section was confusing to read. But, the data show that m^{-/+} pups of ZFP57^{-/-} X WT cross have delayed appearance of stomach milk spots. Yet, ZFP57^{m/+} pups catch up and indeed, exhibit enhanced weight gain during lactation. Since altogether the data suggest either compromised milk let-down by ZFP57^{-/-} dams, or a suckling issue specific to ZFP57^{m/+} pups, the authors go back to characterizing differences between WT and KO dams but are unable to distinguish between these possibilities. They further find that KO dams have milk that contains higher levels of oxidized triglycerides and lower levels of phosphatidylcholines. The authors conclude that this “abnormal” milk may be associated with growth disparities observed in their nursing pups, and ultimately be responsible for enhanced weight gain, but they do not demonstrate it conclusively. Question: Why are light and dark data presented in Figures 5G-J? These studies are not well explained or discussed, even at the end of the manuscript when they are brought up again.

Certainly, it is not new that childhood weight gain is a risk factor for adult health. Here the authors also explore this and find that ZFP57^{m/+} offspring born to ZFP57^{-/-} dams continue to gain more weight and have impaired glucose tolerance among other things as demonstrated by a multi-parameter metabolic assessment system. The bottom line is that the authors find, as others have, that early-life exposures (through lactation) can induce metabolic reprogramming that can have long lasting effects (Picó C, et al., Lactation as a programming window for metabolic syndrome. Eur J Clin Invest. 2021 May;51(5):e13482. doi: 10.1111/eci.13482.)

Cross-fostering comes next and it is important as it can help differentiate between parental and offspring effects on traits. What happens when the ^{+/-p} pups are fostered by ZFP57^{-/-} dams? The answer is a severe failure to thrive during lactation but catch-up post-weaning. Moreover, ZFP57^{m/+} pups cross-fostered to a WT dam gain even more weight during lactation and display obesity at 6 mos. There is also data showing gender specific effects and data showing effects based on light/dark; none of the data are well framed or explained. All told, the studies suggest that concordance between gestational and nursing mother’s genotype is important in the circumstance of ZFP57 loss and supports the notion that there is in-utero adaptation to postnatal resources provided by the dam. This is interesting, but the authors provide no satisfactory explanation for the observation. Could this be due to the master imprinting function of ZFP57 embryonically or in “earlier developmental stages”? Or is it that ZFP57 functions to regulate hormonal control of lactation. Understanding this would shed light on what remains a mysterious transcriptional regulatory role of ZFP57 postnatally.

Version 1:

Reviewer comments:

Reviewer #1

(Remarks to the Author)

ZFP57 is a regulator of postnatal growth and life-long health

In this revised manuscript the authors have added additional cross fostering experiments that allowed the differentiation between in-utero effects and postnatal effects on pups of the lack of ZFP57, they also clarified and altered the writing extensively, which strengthens the manuscript. The data also shows that the altered mammary gland development, the resulting lactation and milk in the ZFP57^{nul} dam influences the growth and later health of offspring. Supporting their conclusion that ZFP57 is a regulator of both prenatal and postnatal growth and development and affect life-long health. Interestingly, the results indicate mother-offspring co-adaptation where in-utero effects on offspring are partially mitigated by altered postnatal nutrition (and nurturing?). Something seen in nature where different mammals have adapted their lactation strategy to align with the prenatal development and postnatal requirements of their offspring. The exact mechanism of how ZFP57 affects the function of the mammary gland and how fetal/placental- mammary crosstalk occurs needs further investigation, but this is a compelling question as well as what drives the maternal adaptation to the metabolic challenges of the offspring.

Is it know if the m^{-/+} offspring has transient neonatal diabetes as with human ZFP57 mutations?

Some minor notes

P1-L25: "functions" ...phenotypes or ... influences multiple aspects of MG development and function. Might be better, as the function of the mammary gland is not to branch etc (minor quibble)

P2-L13-14: mammary gland is a key requirement for milk production? that reads weird. Mean to say: that the proper development and functional differentiation of the MG is a key requirement for milk production?

P3-L7 absence of imprinted gene expression? or absence of the genes?

P3-L28-30: sentence does not read well

P3-L41-43: this suggests that there are 120 different cell populations in the mammary gland, suggest rephrasing so the following is clear: x cell populations at Y stages, 2 genotypes in Z replicates (3 stages, 5 cell types, 2 genotypes-WT and null, 4 replicates?)

P4-L10-12: I think that is maybe a bit too gratuitous conclusion, without analysis of DNA methylation, ZFP57 binding and more detailed expression analysis. Maybe the mammary gland lacks factors needed for these genes to be expressed.

How many imprinted genes are DEG in null embryos?

P4-L26: would like to argue that the liver is also an important other organ that contributes to mammary gland physiology (suggest adding here and in the discussion, especially as it has been shown that changes in imprinter genes affect the liver and alter lipid homeostasis during pregnancy, could that affect the lipid content of the milk/MG?)

P4-L42-44: it also looks like there are more alveolar structures in the virgin gland which would correlate with precocious development. this is one indication for precocious development, are there other indications? such as milk protein gene expression in luminal cells.? (in DEG analysis I see it in basal and endothelial cells??? that suggest cell populations that are not "clean") otherwise hyperplasia might be a better terminology?

Note on cell populations: MCAM and Krt8 expression in several populations suggest that the sorting did not result in clean populations which might be an issue in the interpretation of the mRNA-seq data. Enhanced by the expression of classic milk protein gene genes in basal and endothelial cell populations.

P5-L26-28: but in lactation there is less proliferation and more apoptosis so most likely less milk producing cells??

P5-L36-39: The qPCR results are not reflected in the RNA-seq data? any explanation?

P5-L41: Caution; levels of STAT5/6 are not the main indication of functionality, more important is if they are transducing the signal for Prl or other cytokines as indicated by their phosphorylation and nuclear translocation.

P6-L1: "results in precocious development of nulliparous mammary glands" hyper branching and possibly early alveologenesis but not as far as expression of milk protein genes. As mentioned before hyperplasia might be better term? are hormone levels the same?

P6-L26: "dams Compared to WT's", technically hat is compared to WT dams bred with +/- or -/- males, and not WT x WT. Paternal effects of in utero development and gestation cannot be rules out.

P7-L5: Figure 4G what is missing here are m-/+ pups on WT dams

Supplemental Figure 7AB: it would be helpful to depict the curves in A and B in a way that it is easy to compare how the same pups perform on Dams with a different genotype and use very different colors for each, the grays and pinks with different symbols and hatching is extremely hard to decipher. What one wants to establish here is that pups with the same genetic/epigenetic background perform differently when CF on WT vs null dam, so graphs should compare that, have CF to WT and CF to null in the same graph: add graphs for these comparisons.

Can you add statistics on the growth curves to show when the weights start differentiating or are there only statistical differences upon weaning?

Supplemental figure 8: 8H check legend on graph with legend, 8I needs legend on graph

Reviewer #2

(Remarks to the Author)

It is critically important that the reader understands that the offspring are heterozygous for the mutation but they are not identical.

As stated by authors in response to this point raised in initial review "The reviewer is correct that Zfp57m-/+ and Zfp57+/-p offspring are genetically identical, but because of a requirement for maternal ZFP57 in the oocyte (but not paternal ZFP57 in the sperm) to maintain imprints after fertilisation, they are epigenetically distinct."

Any phenotypic differences observed at birth and beyond between Zfp57m-/+ and Zfp57+/-p pups could be due to these intrinsic differences and not due to the maternal environment. It is not possible with this experimental design to tease apart these two factor ie in utero environment v intrinsic epigenetic differences. This does not detract from the findings of a lactation phenotype with metabolic consequences for offspring, but it is incorrect to conclude that "The study identifies ZFP57 as a major regulator of both pre and postnatal resource control".

Minor points

P2 line 15

“During pregnancy, hormonal signals such as progesterone and prolactin, drive mammary gland differentiation”
include placental lactogen

P3 line 12

“reduction in Phlda2 levels which may affect maternal care³²”

This reference refers to Phlda2 in the placenta not the hypothalamus

P4 line 11

“perturbed expression levels of imprinted genes does not result from loss of ZFP57-mediated imprinting control ”

This cannot be decisively concluded from the data presented. It could be that some changes are due to LOI.

P6 line 8 and line 27

“In-utero environment of Zfp57^{-/-} dams impacts offspring birth outcomes ”

Two issues: This new subtitle/statement could be misunderstood to suggest that the environment the mutant females experienced when they were in utero effects their pups' outcomes. However, the authors are referring to offspring's in utero environment. Even reworded - this is statement is not correct. The authors cannot conclude the pup phenotype is due to the maternal environment because both the dams and the pups are mutant in this model - see my next comment.

P6 line 15

“Both crosses yield genetically identical offspring of a single genotype”

As highlighted in first review and also indicated by authors in their response, the offspring are genetically identical but epigenetically distinct. This statement must be included in their results section and their discussion because it is critically important for the interpretation of the findings.

Line 27 “This indicates that the in-utero environment in Zfp57^{-/-} dams affects Zfp57^{m/+} pups.”

Again, the authors cannot make this statement - offspring are genetically identical but epigenetically distinct

Line 28

significantly upregulated/ downregulated

These terms can only be used if the authors have excluded gene changes resulting from changes in cellular composition.

Use higher/lower expression in bulk RNA from whole placenta.

Line 43 “Taken together, these findings suggest (?reveal/?demonstrate)that Zfp57^{m/+} pups born to Zfp57^{-/-} dams show increased birth weight and poor perinatal survival compared to heterozygous pups born to WT dams. “

Instead of heterozygous use Zfp57^{+/-} to be consistent

P7, lines 1 and line 19

The authors repeat the first line.

P7, lines 1-8

Need more clarity on the experimental design. Suggest “We compared WT and Zfp57^{-/-} dams raising their own Zfp57^{m/+} and Zfp57^{+/-} biological litters with fully wildtype litters, and included a cross-fostering paradigm where WT and mutant pups were fostered to dams of the three conditions.”?

P7 line 1 to P9 line 33

Appreciate the additional amount of work undertaken in the new cross fostering experiment. This is the critical experiment and key to interpreting the findings. The experimental design should be clearer in text and M&M. Generally this section is hard to follow.

M&M / figures

Presumably the authors have combined the original data from the non-fostered animals with the new data for the fostered animals ie two separate experiments? This needs to be clearly stated. And - if not performed concurrently - this should be acknowledged in discussion as a potential weakness.

Discussion

Needs to articulate more clearly that the heterozygous offspring are not identical – which may account for some phenotypes ie increased weight of Zfp57^{m/+} at birth.

Reviewer #3

(Remarks to the Author)

my comments have been sufficiently addressed.

Reviewer #4

(Remarks to the Author)

In this revised manuscript entitled, ZFP57 is a regulator of postnatal growth and life-long health, the authors present additional data about the role of Kruppel-associated box-containing zinc-finger protein (ZFP57) in the mammary gland, in the uterine environment and in the embryo. ZPF57 is a master regulator of imprinting that acts to recruit DNA methyltransferases lacking the targeting domains required to recognize imprinting control regions. Yet, it is not functioning in this canonical role, at least in the mammary gland. I am still unclear whether it functions canonically in the placenta or in the embryo. The manuscript is still confusing to read and not targeted to a general scientific audience. There are mistakes. It also lacks sentences that introduce the experiments conceptually and that synthesize the potential meaning of the results.

The major improvement to the manuscript was the addition of appropriate controls that allow the authors to better tease apart the role of the dam in the postnatal window. The most straight forward parts of this manuscript focus on the role of ZFP57 in the mammary gland (Figures 1-3) and the characterization of how abnormal milk produced by ZFP57^{-/-} glands (Figure 5C). The least straight forward focus on the in utero effects (Figure 4, 5, 6). Many interesting observations have been made, yet

the authors fail to synthesize any mechanistic insight from their results. The support for the take-home message that ZFP57 regulates nutritional provision to offspring through two distinct mechanisms: prenatally, through genomic imprinting and postnatally through mammary gland and lactation-related genes is not clearly articulated in the results or discussion section. In my opinion, this is two different manuscripts about nutritional provisioning and trying to put them together into one manuscript does not serve either the data or the reader. Furthermore, without mechanistic insight it is difficult to see how this manuscript reveals impactful information. It is not new that childhood weight gain is a risk factor for adult health. The authors find, as others have, that early-life exposures can induce metabolic reprogramming that can have long lasting effects (Picó C, et al., Lactation as a programming window for metabolic syndrome. *Eur J Clin Invest.* 2021 May;51(5):e13482. doi: 10.1111/eci.13482.). They also find that prenatal developmental conditions can predispose offspring to long-term maladaptation, something that others have also shown (<https://doi.org/10.1017/S0954579412000764>).

Below are comments concerning the new data, and a few observations from reviewing the manuscript again.

Figure 1-3 examines Zfp57 expression in five mammary cell populations. The authors use alphaSMA as a co-expressing marker for stromal cells but this doesn't work well in the mammary gland where alphaSMA marks basal cells. Here is an atlas of fibroblast gene expression during mammary gland development (<https://doi.org/10.1038/s44318-025-00422-3>); PDGFRalpha is a reasonable marker for mammary gland stroma. The FACS plots displayed in the Supplementary data (Supp Fig 4) do not show well isolated populations of cells; the gate calling appears non-standard (please see Shehata et al., doi: 10.1186/bcr3334.)

Figure 4 assesses the in-utero environment incorporating experiments suggested by the reviewers to disentangle the myriad of effects described in the original manuscript. This section is still very confusing. Below are my questions based on the understanding I gleaned after reading the section multiple times.

Here, the authors focused on "two prominent mouse crosses: WT females crossed to a Zfp57^{-/-} males, generating paternal heterozygous pups (Zfp57^{+/-}), or ZFP57^{-/-} females crossed to WT males generating maternal heterozygous pups (Zfp57^{m/+}). Both crosses yield genetically identical offspring of a single genotype,"

Ques: the crosses yield genetically identical offspring but are the epigenetically identical? In the introduction the authors said "Given that loss or partial loss of imprinting at most of the affected genes is found in Zfp57 homozygous mutants, this indicates that imprinted gene regulation by ZFP57 is relevant in both embryonic and adult tissues." Furthermore, a study has shown that showed "DNA methylation at a few imprinting control regions was partially lost without maternal Zfp57 in Zfp57 heterozygous mouse embryos derived from Zfp57 homozygous female mice. This suggests that maternal Zfp57 is essential for the maintenance of DNA methylation at a small subset of imprinted regions in mouse embryos." (doi: 10.3389/fcell.2022.784128). Can the authors please clarify how this result might affect the interpretation of their data?

Supp Fig 6a. In the text, the description of the data is "The hypothalamus is involved in feeding circuitry and suckling³⁵, therefore, we next quantified changes in imprinted gene expression affected by Zfp57 in this region. Zfp57^{m/+} offspring exhibit a few minor changes in hypothalamic expression of imprinted genes compared to WT, including Rasgrf1, Snrpn, Nnat and Nespas (Supplementary Figure 6A)." What is shown, but not stated, is that Zfp57^{+/-} offspring do not display changes in gene expression. Please state clearly that this comparison shows changes in Zfp57^{m/+} offspring compared to WT and Zfp57^{+/-} offspring. Furthermore, I am not sure why the authors downplay these changes by describing them as minor when they are significant, especially if they potentially contribute to the suckling defect? Or do the authors have a different explanation for the suckling defect? In any case, the authors should be careful about calling significant differences minor or at least justify their conclusion.

Supp Fig 6e. The figure legend does not match the figure so this is confusing. In Panel E, why are the authors calling out "paternal het" and "Zfp57^{-/-}" in the panel legend? In the text, the description of the data is "Zac1 and Snrpn were significantly upregulated in Zfp57^{m/+} placentas and Igf2 and Dlk1 were downregulated compared to Zfp57^{+/-} animals (Supplementary Figure 6E)." Aren't Zfp57^{m/+} placentas actually the placentas of the Zfp57^{-/-} dams? And doesn't paternal het mean WT placenta? Apparent mismatch between text and figures is confusing to the reader. It would be terrific if the authors made an effort to be consistent, and therefore clearer, in the use their own terminology.

The authors perform cross-fostering experiments to uncouple maternal and offspring perinatal phenotypes and conclude that "This indicates that Zfp57^{m/+} pups have impaired suckling." I understand the fact that Zfp57^{m/+} pups born to Zfp57^{-/-} dams are heavier, which suggests this could be a placental effect, but do the authors think impaired suckling is a placental effect? What do the authors think is the explanation for why Zfp57^{m/+} pups have impaired suckling (Figure 4G)? Is this due to the "few minor changes in hypothalamic expression of imprinted genes compared to WT, including Rasgrf1, Snrpn, Nnat and Nespas (Supplementary Figure 6A)." and, therefore, an offspring factor due to the fact that the "hypothalamus is involved in feeding circuitry and suckling", which they point out earlier in the previous paragraph? If so, can the author link these data together better for the reader? Wouldn't this be considered an offspring effect, even though this section is ostensibly about in utero effects? Alternatively, if hypothalamic expression is not the cause (because the changes are minor (but significant)), are the authors concluding the effect is due to placental imprinted gene expression (Supplementary Figure 6E) – an in utero effect? Authors: please clarify your thinking. And, if there is an offspring effect, consider modifying the section heading ("In-utero environment of Zfp57^{-/-} dams impacts offspring birth outcomes").

The point is that the authors test a number of hypotheses in this section, yet they conclude their paragraphs with a general statements about the findings ("Taken together, these findings suggest that Zfp57^{m/+} pups born to Zfp57^{-/-} dams show increased birth weight and poor perinatal survival compared to heterozygous pups born to WT dams." "This indicates that

Zfp57m-/+ pups have impaired suckling. ”), without a concluding statement about the biological mechanism that might explain these data. This lack of link to mechanism is a major problem with this manuscript: at least some data are here, but any link to biological mechanism is tenuously drawn.

The description of cross-fostering was very confusing:

“The original litter and the dam were removed from the dam's cage, and the cross-fostered pups were gently rolled in the nesting material of the fostering dam. Subsequently, the original dam was returned to her home cage with a new litter, which was weighed every 1-2 days until weaning. The number of pups was normalised to the size of the smaller litter.”
What is the new litter that goes back with the original dam? Are these the cross-fostered pups that have been rolled in the nesting material? If so, then they would be referred to as THE new litter. A new litter suggests a different, unspecified, litter. If you want the reader to understand your experiments, the authors should consider carefully editing their manuscript for clarity.

Figure 5 examines the offspring over time. A) The rationale for separating the data based on sex of the pup is not clear. In any case, the data appear similar. It would be helpful either in the text or in the Figure legend to remind the reader of the “two prominent mouse crosses” that I surmise are still the focus of this panel (“WT females crossed to a Zfp57-/- males, generating paternal heterozygous pups (Zfp57+/-p), or ZFP57-/- females crossed to WT males generating maternal heterozygous pups (Zfp57m-/+).”). The description “we monitored the growth of Zfp57m-/+, Zfp57+/-p and WT pups raised by Zfp57-/- and WT mothers.” is not particularly clear, especially since CF experiments have already been introduced. It is very helpful that the authors included the appropriate WT/WT control.

B) The rationale for looking only at male offspring is not clear (Legend: Weights of WT, Zfp57m-/+ and Zfp57+/-p male offspring at weaning). Given the number of statistical tests performed here, it is necessary for the authors to indicate the comparison group so when the text reads “Zfp57m-/+ pups exhibited significantly increased weight gain during lactation when raised by or cross-fostered to Zfp57-/- dams.” the reader needs to know the comparison group so that they can see the significant change on the graph. This is a very complicated panel so I may be missing something here, but I see that Zfp57m-/+ pups raised by Zfp57-/- dams and CF to WT have significantly increased weight gain during lactation compared to WT pups raised by WT dams or WT pups raised by WT dams and CF to WT dams or WT pups raised by WT dams and CF to Zfp57-/- dams. But, Zfp57m-/+ pups raised by Zfp57-/- dams do not have significantly increased weight gain during lactation compared to Zfp57m-/+ pups raised by Zfp57-/- dams and CF to WT or compared to Zfp57m-/+ pups raised by Zfp57-/- dams and CF to Zfp57-/- dams. That is, the possibilities in this graph are many. Maybe the authors are referring to a different comparison, but how would we know? The bottom line is that the authors need to clarify the language and indicate on the panel (maybe by using characters: #*) and in the text the comparisons, because it is not as if one can simply look at the figure and easily see the relevant comparison. Also, I don't think the reduction in weight at weaning for the Zfp57+/-p pups CF to Zfp57-/- dams can be classified as severe unless the authors can find a reference for this classification (PMCID: PMC3750667 PMID: 24209967).

They finish the section by concluding that Zfp57-/- dams are compromised in their ability to support the normal postnatal growth of offspring. This is followed up by an analysis of milk and the data support the authors' conclusion that Zfp57-/- mothers produce abnormal milk. The authors conclude that mother-offspring co-adaptation is associated with milk composition, but it would be helpful if they explain their thinking in terms of mother's genotype, in utero environment, pup epigenetics (which is generally dismissed) and development.

Figure 6 examines life-long health outcomes. The authors do a better job explaining their at least some of their comparisons. But, there are many comparisons to be made, and the authors do not call out all of them so again being explicit and indicating on the panel that comparisons that are being called out in the text would be very helpful. I also suggest that the salient results be tabulated in a way that makes them accessible to the reader. Sorting through the information, the authors find some differences and make conclusions concerning pup metabolic changes in response to the maternal environment and mother-offspring co-adaptation.

The limitation to the study pointed out in the discussion of not using a Cre-line to specifically delete Zfp57 in mammary cells because other organs will retain their Zfp57 levels misses the point about learning whether the alterations observed in the mammary gland are intrinsic to the gland (i.e. “the stemness of luminal progenitor cells during development”) or systemic influences (i.e. hormone levels, which the authors measured and found no differences in). Obtaining this level of understanding could help the authors draw mechanistic conclusions, something that is lacking in this manuscript.

Minor:

“Data show” for n/v agreement because the word data is a plural noun

Spell and punctuation check: there are many examples. Here is one: expressing (Fig Legend 1)

Version 2:

Reviewer comments:

Reviewer #1

(Remarks to the Author)

The authors have sufficiently addressed my concerns in this revision

Reviewer #2

(Remarks to the Author)

Authors have addressed my comments

Reviewer #4

(Remarks to the Author)

Reviewers have offered many suggestions to improve this manuscript entitled, ZFP57 is a regulator of postnatal growth and life-long health. In the first revision, this meant the authors included, among other improvements, a key control that was lacking from the original experimental design and more clarity in the interpretations of their experiments. In this revision, the authors used reviewers' comments to improve the intelligibility of the manuscript, yet this version is still confusing and not well communicated to a broad audience. Furthermore, the lack of a link to any specific mechanisms continues to be a major problem. Although ZPF57 is a master regulator of imprinting in the embryo, the authors find that it functions in an imprinting-independent fashion in the adult mammary gland. And, while there are a host of phenotypic changes in the mammary gland (involving alveologensis), ZPF57 is expressed only in basal epithelial and endothelial cells, even though it appears to function by regulating the number and the function of alveolar progenitor cells (or at least luminal progenitor cells as shown here) -- a conundrum. Following the phenotypic analysis, the authors present a series of mating and cross-fostering experiments that further demonstrate how complex the consequences of ZPF57 loss is. Altogether, the data from cross-fostering suggest that concordance between gestational and nursing mothers' genotypes is important in the circumstance of ZFP57 loss and support the notion that there is in-utero adaptation to postnatal resources provided by the dam. This is an interesting message but the studies in the second half of the manuscript provide little insight into the studies in the first half. Moreover, the conclusions that can be made are only tenuously drawn. In my opinion, the authors have provided the foundation for a more substantial investigation that might yield mechanistic insight, but such studies are not described in this manuscript.

REVIEWER COMMENTS

Reviewer #1 (Remarks to the Author):

Very interesting premise, that contributes to our understanding of the interplay of genetics epigenetics in -utero and post-natal development. And the role lactation can play in this. However, the current manuscript could lay out more clearly what are prenatal/in-utero programming effects and what are postnatal effect due to dam nurturing (maternal behavior and lactation)

General conclusion that both pup phenotype and dam phenotype are important seems valid, but the details are muddled in the reported analyses and it is really not clear what the effects of maternal postnatal nutrition/nurturing are vs the effect of the pups in utero exposure and genotype.

The manuscript could greatly benefit from focusing on this and using the relevant comparisons to draw conclusions.

Main concern

To address if the ZFP null dam has true intrinsic lactation defect one would have to use a conditional knock-out or transplantation experiment.

In the context of this manuscript at least all measurements should be performed comparing WT pups on WT dam and WT pups on ZFP58 null dams. This will still conflate maternal behavioral issues that might affect offspring's nutritional intake and mammary gland intrinsic defects, but at least one can say for sure that these effects are due to the dam in the postnatal window.

We found this comment highly valuable and conducted substantial new cross-fostering experiments, in which WT pups were cross-fostered to either WT or *Zfp57^{-/-}* dams (Figures 4-6, Supp. Figures 7-9). This enabled us to distinguish the *in-utero* effects from postnatal nutritional/nurturing effects as requested by the reviewer. These experiments included tracking the pups for six months, allowing us to determine that in-utero effects contribute to the phenotype observed in *Zfp57^{m/+}* influencing their birth parameters and metabolic performance throughout life. Additionally, our findings indicate that *Zfp57^{-/-}* dams produce milk with impaired composition, which is linked with attenuated pup growth during lactation. We also observed synergistic effects between pups and dams, driven by co-adaptation or maladaptation.

Furthermore, there is clear indication that the in-utero environment in the null dam affect live births and gestation duration and metabolic imprinting in the fetus.

We agree with the reviewer and have incorporated additional data from WT × WT intercrosses and cross-fostering experiments throughout the manuscript (Figures 4-6, Supplementary Figures 7-9). These additions allow us to better distinguish phenotypes arising from the in-utero environment experienced by *Zfp57^{m/+}* pups from those influenced by maternal factors.

Another consideration to take into account, all the null dams are presumably offspring of hemizygote or homozygote null dams so presumably been exposed to a ZFP57 +/- or -/- in utero and postnatal environment. The parental origin of null dams at least needs to be reported and taken into consideration for the interpretation.

All the *Zfp57^{-/-}* dams were born in crosses between a heterozygous female and a homozygous male, and thus were exposed to heterozygous dam *in-utero* and postnatally. We've added these details to the methods section (Page 11, lines 38-40).

From the presented cross fostering data it suggest that despite identified morphological histological and cellular differences identified in the null dams during mammary gland development and differentiation in pregnancy, and milk lipid content differences, this does not significantly affect growth of the WT offspring suckling the null gland. (fig 6 and fig s7 & S8)

The new data added to the manuscript as requested by the reviewer, including WT pups cross-fostered to either WT or *Zfp57^{-/-}* dams, demonstrates that the *Zfp57^{-/-}* dam significantly impairs offspring growth during lactation across all genotypes. Furthermore, the metabolic syndrome that WT pups experience when cross-fostered to a *Zfp57^{-/-}* mother, is less severe if the pup is a *Zfp57^{m/+}* and hence derived from a birth mother of the same genotype as the foster mum (Figure 6). These findings suggest a role for mother-offspring co-adaptation, indicating that while these offspring are predisposed to metabolic dysfunction, their phenotypes may be better managed when raised by a genetically mismatched foster mother.

Differences in growth of m/+ could be because larger pups might provide a much stronger suckling stimulus of the gland and enhance lactation, WT glands can sustaining more growth of these pups.

We appreciate this comment and have addressed it by conducting a suckling experiment (Figure 4G) with cross-fostered pups to distinguish phenotypes arising from the in-utero versus postnatal environment. Our findings indicate that *Zfp57^{m/+}* offspring exhibit impaired suckling, not a stronger sucking stimulus. However, despite this, they grow larger by weaning, likely due to differences in milk composition provided by the *Zfp57^{-/-}* dam.

Is the % weight gain per pup relating to number of pups in the litter at weaning different for the m/+ on WT vs on null dams? If the pup numbers become lower due to initial attrition the gland can sustain more demand.

We only included dams that retained the entire cross-fostered litter, with standardised litter sizes of 5-6 pups.

Interesting to observe that the metabolic programming in m/+ pups is radically different from p+/- pups, which only shows when nursed on ZFP57 null dams. Suggesting that the in-utero environment in ZFP 57 null mice is different? (as suggested by different birth weights and longer gestation)

m/+ pups already heavier when born, grow heavier during nursing?

It seems that ZNP57 null dams attenuate growth of heterozygote pups (both m/+ and +/-p), m/+ do not grow as much on null as on WT dam, and the same is the case for +/-p offspring, while WT offspring is not affected by suckling on null dam.

Yes - our data suggests that *Zfp57^{m/+}* and *Zfp57^{+/-p}* exhibit distinct phenotypes when nursed by either WT or *Zfp57^{-/-}* females (Figure 5B). To enhance clarity, we have reorganised this figure accordingly.

The new experiments included in this version of the manuscript (Figures 4, 5, and 6) confirm that, as the reviewer points out, the in-utero environment provided by *Zfp57^{-/-}* females differs from that of the wild type mother of the *Zfp57^{+/-p}* pups. Furthermore, these *Zfp57^{-/-}* dams attenuate growth across all genotypes. Notably, in *Zfp57^{m/+}* offspring, this attenuation appears to mitigate metabolic syndrome slightly (Figure 5B, 6C,6F), supporting the concept of co-adaptation. However, when these pups are nursed by a WT foster mother with a genotype differing from their birth mother, they exhibit signs of maladaptation (Figure 5B).

Changed metabolism and signaling in the null dam resulting in different mammary gland development and milk composition probably also affects placental function or general metabolism during pregnancy.

We agree with the reviewer's suggestion and have incorporated expression data from placentas into Supplementary Figure 6E. Notably, some of the affected genes do have established links to metabolism, including *Zac1*, which is associated with transient neonatal diabetes in humans, and *Dlk1*, which has multiple physiological functions including a role in regulating nutrient metabolism.

Extended comments of figures and related text

Figure 1: and related text

Text relating to Figure 1d: "was expressed in basal and endothelial cells. However, the level of ZFP57 expression compared to the basal, luminal, and stromal markers Krt5, Krt8 and α SMA was lower. Notably, luminal differentiated, luminal progenitors and stromal cells exhibited the lowest expression (Figure 1D). "

This is confusing and it seems that the same is said 3 times, ZFP57 expression is only detected in basal and endothelial cells? Then authors go on to say that expression is lower than Krt5, Krt 8 and Sma, in which cell types? the basal cells or all cell types? Looks like it is high in the endothelial cells, compared to the cell type markers that are not expressed in the endothelial cells ... why no endothelial cell marker used? Then it seems that there is expression in the other cell types but very low.

Thank you for this recommendation. We have added the endothelial marker MCAM (CD146) and revised the text accordingly (Page 3, lines 25-26). Our findings demonstrate that *Zfp57* is expressed in both mammary basal and endothelial cells. Additionally, Supplementary Figure 1B compares *Zfp57* expression levels to the median expression across various tissues, showing that while *Zfp57* expression in the mammary gland is close to the median, it does not reach the levels observed for key marker genes, highly expressed in these cells.

Furthermore, in the current graph it is not possible to assess whether expression of some the genes is nearly at the same level as b-tubulin (or less than that 1-fold) in certain cell types, because some of the depicted genes are very highly expressed in the cell type. Firstly, I suggest grouping the graphs by gene, secondly use broken Y-axis if there is a large difference (and/or a logarithmic scale).

It is not clear what the "fold-change" relates to (expression relative to beta-tubulin? Or also relative to any expression of a gene in one of the celltypes?)

We have adjusted the grouping as requested by the reviewer and updated the quantification to reflect relative expression compared to the lowest-expressing cell type for each transcript. This change has also been incorporated into the figure legend (Page 19, lines 12-15).

The legend and figure do not match up for 1E &F

Thank you - we have corrected the legend (Page 19, lines 15-16).

Figure 2 and related text

A) Are the nulliparous wt and null mice in the same oestrus stage? And which one is that? Oestrus stage affects proliferation and apoptosis.

We appreciate the reviewer for highlighting this important point. To ensure consistency, both WT and *Zfp57*^{-/-} samples were collected at the same oestrus stage (oestrus). This information has been added to the Methods section (Page 11, lines 37-38).

Is it possible to assess alveolar development? (possibly from sections)

We attempted to assess alveolar development in our staining; however, due to the nature of whole-mount samples including their variable thickness leading to variable staining intensity, it was challenging to ensure that only alveoli were quantified rather than background even after applying corrections.

c) is there a difference in which cell type Ki67 is higher (basal vs luminal, lum prog?) or were you see the staining in ducts vs alveoli? There seems to be a difference in G9.5

The DAPI and Ki67 staining are consistently clear across all slides and fields, whereas the K8 and α SMA stains appear more diffuse due to their cytoplasmic localisation. The absence of co-localisation makes it challenging to determine whether the Ki67 signal originates from luminal or adjacent basal cells, particularly in dense tissue. To address this, we have included additional representative images in Supplementary Figure 4, providing a qualitative comparison that highlights Ki67 signals in both compartments.

D) What is quantified? Number of cells positive out of all cells (or epithelial cells) in the field? The positive K67 in the lactating ZFP null image looks like they are not in an alveolar structure.

Our original quantification measured relative intensity compared to DAPI, however, in response to the reviewer's request, we have revised it to reflect the percentage of positive cells, as indicated by nuclear staining (Figure 3D).

E/F) same for TUNEL staining: is there a difference in which cell type TUNEL staining is higher (basal vs luminal, lum prog?) or just were you see the staining in ducts vs alveoli? What is quantified? Number of cells positive out of all cells (or epithelial cells) in the field?

In the original manuscript, we quantified the ratio of the positive signal area to the duct area. As per the reviewer's suggestion, we have revised this to quantify the percentage of positive cells/nuclei (Figure 3F). However, the kit used for this reaction does not allow for co-staining with other antibodies or cell-type markers, preventing us from directly linking the TUNEL staining to a specific cell type.

G) is the expression of milk protein genes or fatty acid/ lipid synthesis affected?

We thank the reviewer for this comment and added expression data for *Wap*, *Csn2* and *Lalba* to Figure 3G. All genes exhibit a significant reduction in expression during gestation, suggesting a delayed onset of lactogenesis.

Figure 3 and related text

The accompanying differential expression results should be shared as supplementary tables (DEG table and count tables) as well as the gene lists for go terms, (this can be retrieved from EnrichR using appyter).

We agree that this data is important and have added Supplementary Table 2, which includes all differentially expressed genes (DEGs) for each stage and cell type, as well as Supplementary Table 3, which provides the gene lists for the associated GO terms.

Minimal expression of a gene does not mean that when it is knocked-out it will not have an effect, also besides the effect of knocking out ZFP57 in distant tissue that can affect mammary gland function, the cell-cell signaling within the mammary gland and the different cell types should be considered and discussed e.g. the possible effect of the the gene

changes in the basal cells where *Zfp57* is expressed at higher levels.

Yes, we agree and thank the reviewer for this comment. We have incorporated this point into the discussion (Page 9, line 47 – Page 10, line 7).

Figure 4 and related text

What are the authors trying to illustrate here? A defect in pups or in dams?

Prenatal (4A, B, C) indicate that there are prenatal/in-utero effects in pups born to null mothers, however that does not address survival to birth of individual pup genotypes.

These panels refer to both the dam and the pups. Our new experiment further clarifies their individual contributions and we hope this is now clearer in the revised figures and text. Our data show that *Zfp57^{m/+}* offspring have a higher birth weight and impaired suckling (Figure 4A, 4G). Additionally, Figure 4D indicates that the complete absence of ZFP57 in utero leads to reduced pup survival, as other crosses producing *Zfp57^{m/+}* pups do not exhibit this survival deficit.

A) How does this compare to birthweight of WT born to WT? It would be helpful to put this in context of weight of WT pups born to WT dams.

We included WT × WT crosses to investigate this further. Our data indicate that *Zfp57^{m/+}* offspring exhibit increased birth weight, whereas *Zfp57^{+/-p}* do not show a significant difference compared to WT pups (Figure 4A).

B/C) for null dams longer gestation and lower # of pups at p0 indicates that pups are dying in-utero. Is there a genotype difference in pups that survive to birth?

The cross used in this experiment (*Zfp57^{-/-}* female × WT male) exclusively produces *Zfp57^{m/+}* offspring, and in this case, we observed prolonged gestation and a reduced number of pups. However, when comparing this to other crosses that also generate *Zfp57^{m/+}* alongside other genotypes (e.g. *Zfp57^{+/-p}* female with a *Zfp57^{+/-p}* or a WT male), we did not observe extended gestation or differences in the genotypes of the surviving pups.

However, the studies shown related to postnatal development D and E conflate in-utero environment, postnatal behavior, mammary gland/lactation phenotype and pup genotype. These should be properly cross fostered to address which pup phenotypes are due to mammary gland/ maternal behavior phenotypes/genotypes, first comparing WT dams and ZFP^{-/-} dams feeding WT pups. Also, there are obvious maternal behavior effects. How can one distinguish those from possible lactation defects?

To address this key point, we conducted cross-fostering experiments in which WT pups were fostered by either WT or *Zfp57^{-/-}* dams. These experiments revealed that *Zfp57^{m/+}* pups exhibit impaired suckling (Figure 4G). Pups were cross-fostered at postnatal day 1 to ensure successful fostering, but since most pup mortality occurs between P0 and P1, we cannot definitively determine whether these early deaths are driven by the dam or the pups. However, our data clearly demonstrate impaired suckling in *Zfp57^{m/+}* pups (Figure 4G).

Zfp57^{-/-} dams have a full knockout of *Zfp57*, which results in abnormal maternal behaviour, as noted by the reviewer and illustrated in Figure 4F. However, distinguishing maternal behaviour-related phenotypes from lactation-specific effects would require a tissue-specific knockout model, which is currently unavailable.

D/E) are these nursed on their birth mothers?

Yes, panels D and E include only pups with their birth mother. We have now added WT pups born to WT dams as a comparison to *Zfp57^{+/-p}* and *Zfp57^{m/+}* pups.

D) conflates in-utero environment, with dam behavioral and lactation phenotypes as well as pup genotypes. Is there a difference in mortality between pup genotypes when there is more than one genotype in the litter?

Our previous work demonstrated that this epigenetic difference affects methylation at specific differentially methylated regions (DMRs) within imprinted control regions (Takahashi et al., Cold Spring Harb Symp Quant Biol, 2015; 80:177-187, doi:10.1101/sqb.2015.80.027466). Methylation differences were observed in *Zfp57^{m/+}*, zygotic *Zfp57^{-/-}* and maternal-zygotic *Zfp57 MZ^{-/-}* embryos.

In mice, deletion of both the maternally inherited copy in oocytes and the zygotic copies in early embryos (*Zfp57MZ^{-/-}*) results in severe loss of methylation at imprinted loci, leading to embryonic lethality. Deletion of the zygotic ZFP57 copy (*Zfp57Z^{-/-}*) causes partial neonatal lethality. The reduced survival probability observed in panel D has been reported previously and occurs in heterozygous *Zfp57^{m/+}*, suggesting that in-utero factors influence survival during the P0-P1 period. We have cited this study in the introduction (Page 2, line 1).

E) Again, dam genotype and pup genotype conflate. Also are there still the same number of pups in the null dam litter at P2? If not the lack of difference can be because the demand is down and now the impaired gland can provide enough milk, assuming there is no effect of pup genotype on their ability to suckle. would be helpful to put this in context of Wt pups on WT dams and WT pups on null dam.

We added WT pups born to WT dams in Figure 4E and conducted suckling experiments with WT pups cross-fostered to either WT or *Zfp57^{-/-}* dams. Together, these data indicate that *Zfp57^{m/+}* pups exhibit delayed milk spot presence and impaired suckling. To ensure consistency in assessing milk spot presence, litter sizes were standardised to 5–6 pups, and litters with higher pup mortality at P2 were excluded from this experiment. These details have been added to the Methods section (Page 12, lines 8-10).

G) By using WT pups the possibility of suckling issues based on genotype or imprint type could be ruled out.

We appreciate the reviewer's suggestion and have added WT pups cross-fostered to either WT or *Zfp57^{-/-}* dams in Figure 4G. This allowed us to confirm that *Zfp57^{m/+}* pups exhibit impaired suckling, as WT pups fostered by *Zfp57^{-/-}* dams gain weight rapidly.

H) is the relative weight gain significantly different? These graph indicate increased body weight but they start out heavier.

We agree that *Zfp57^{m/+}* pups have a higher initial birth weight. However, even after normalising weight gain to birth weight and expressing the data as percentage weight gain, *Zfp57^{m/+}* pups still showed greater accumulation. This information has been added to Supplementary Figure 6F.

“Overall, this indicates that ZFP57^{-/-} mothers produce abnormal milk associated with growth disparities observed in their nursing pups, ultimately resulting in enhanced weight gain” However, this seems to affect them^{-/+} pups. as is shown in subsequent cross fostering experiment the m^{-/+} pups also gain more weight when suckling a wild type dam. While there is no significant effect on growth of WT pups on null dams. If this is due to milk composition

you would expect to see the same effect in WT pups on null dams. This does not add up and illustrated the importance of cross fostering studies.

We agree and thank the reviewer for this comment. We have conducted cross-fostering experiments to address this, details are provided below, in the next point.

Figure 5 and related text

There are clear metabolic differences between the offspring here but again the comparison of maternal +/- and paternal +/- on null dam vs WT is not a valid comparison, as it seems that the genotype and paternal inheritance (in-utero environment) affect the offspring's metabolism and growth.

To determine if the abnormal milk composition has an effect same pups should be assessed on different dams, preferably WT pups on WT and null dam.

As requested, we conducted these additional experiments and incorporated the findings into the manuscript. Cross-fostering WT pups to *Zfp57^{-/-}* dams confirmed that *Zfp57^{-/-}* dams attenuate pup growth regardless of pup genotype (Figure 5B). However, *Zfp57^{m/+}* pups exhibit hallmarks of metabolic issues, a developmental effect acquired in-utero, characterised by increased birth weight and accelerated pre-weaning weight gain and throughout life. Additionally, the abnormal milk produced by *Zfp57^{-/-}* dams negatively impacts WT pups by reducing their weight gain, demonstrating its influence independently of in-utero factors.

To assess if the genotype of the offspring further affects the metabolic outcome later in life *m/+* and WT pups suckled on ZFP57 null dams should be compared.

In this major revision, we incorporated a series of cross-fostering experiments to distinguish developmental effects in *Zfp57^{m/+}* pups that arise from gestation in a *Zfp57^{-/-}* uterus versus those influenced by postnatal rearing by a *Zfp57^{-/-}* dam. These findings are now presented in Figure 6 and Supplementary Figures 8–9. Our data indicate that the long-term metabolic outcomes in *Zfp57^{m/+}* pups are primarily developmental effects acquired in-utero, though the nursing mother also plays a role. When *Zfp57^{m/+}* pups are raised by *Zfp57^{-/-}* dams, which produce abnormal milk, they develop hallmarks of metabolic syndrome. However, these effects are further exacerbated when *Zfp57^{m/+}* pups are nursed by the genetically different WT mother. Additionally, WT pups raised by *Zfp57^{-/-}* dams exhibit attenuated pre-weaning weight gain (Figure 4B), though this effect is not long-lasting and normalises by six months (Figure 6A–D).

Most of fig 4 and 5 is not as relevant as cross fostering experiments to really contribute to the understanding of what is due to pup genotype, in-utero environment, postnatal nutrition and nurturing. Fig4 A-C (with wt data included in A) show in-utero effects. Rest of these figures and experiments are too conflated to go beyond concluding there is “an effect” of deletion of ZNF57 and can be deleted IMO.

The cross-fostering experiments later described are better in enabling to start distinguishing between the different effects deletion of ZFP57 has and in which paternal context and interaction with dam and offspring. I do not see how most of what is show in fig 4 & 5 contributes I suggest to leave this out to keep a somewhat complicated concept due to possible parental & genotype effects and pre- and post-natal effects more to the point

We conducted substantial new cross-fostering experiments (Figures 4-6, Supp. Figures 7-9). This enabled us to distinguish the *in-utero* from postnatal nutritional effects as requested by the reviewer.

Supplemental 5

5l) Should at least also be assessed using WT pups on WT dams and m-/+ pups on WT dams to determine what is due to genotype (and in utero environment) of the pups and what due to maternal postnatal factors (behavior, lactation)

This revision includes cross-fostering experiments with WT pups cross-fostered by either WT or *Zfp57^{-/-}* dams, clearly demonstrating that the phenotypes observed in *Zfp57^{m/+}* pups relate to their distinct in-utero environment. This is evident from differences in birth weight, where WT and *Zfp57^{+/-p}* pups have comparable weights and milk spot presence.

Supplemental 6

A) Same as for 5l) (should include WT-WT and m+/- pups on WT dams) also does this include all genotypes of pups born to WT? (why is the color in males different?)

B/C) Same as for 5l)

This revision includes cross-fostering WT pups to WT or *Zfp57^{-/-}* dams. Supplementary Figure 6A presents data exclusively from heterozygous pups. We did not perform GTT on the additional cross-fostered pups, as they have already undergone multiple longitudinal experiments, and an additional GTT would exceed the severity limits under the legal framework of our UK Government Home Office Animal Licence.

Figure 6 (supplemental fig 7 & 8) and accompanying text.

The interpretation of effect of suckling a genotypically different mother from its own genotype is per-se not incorrect but I think it can be brought back to what happens in utero and how the null dam rears the offspring postnatally, as mentioned earlier.

“These experiments highlight the importance of concordance between gestational and nursing mother’s genotype and emphasises the role of in-utero adaptation to postnatal resources provided by the birth mother. “ This seems an overstatement

Our newly added experiments, using WT pups that did not develop in *Zfp57^{-/-}* dams, emphasise the importance of genotype concordance between mother and pup. The new data further clarifies which phenotypes originate from in-utero influences and which are shaped by the postnatal environment, strengthening this conclusion.

“Additionally, *ZFP57^{m/+}* pups which showed the tendency to gain excessive weight when raised by *ZFP57^{-/-}* dams, exhibited greater weight gain when raised by a WT dam, which differed in genotype from their birth mothers” the gain weight in both cases just not as much on the null dams (just like the +/-p pups).

We agree. The new experiments (Figures 4, 5, and 6) provide additional data that enhance our understanding of these phenotypes, demonstrating that *Zfp57^{m/+}* pups develop a metabolic phenotype primarily influenced by their in-utero environment. Furthermore, *Zfp57^{-/-}* dams attenuate pup growth during lactation, which appears to have a protective effect on the metabolically compromised *Zfp57^{m/+}* pups.

It is unclear from the graphs in fig6 if the metabolic analysis for the different offspring are different please show the AUC numbers in figure 6 not in supplemental. Also suggest including the growth curves in figure 6 and not the supplemental.

In this revised version, we included WT pups cross-fostered to either WT or *Zfp57^{-/-}* females and tracked them from birth (Figures 4 and 5) through 6 months, at which point we conducted a series of metabolic tests now presented in Figure 6. To streamline the figures, we moved some metabolic measurements to Supplementary Figures 8 and 9 while

maintaining a primary focus on fat-related metabolic phenotypes. We also chose to include the GTT in the main figure, however, since AUC reflects only the total glucose response rather than the dynamics of the glucose challenge, we have placed the AUC analysis in the supplementary information (Supplementary Figure 8F, I). Additional explanations regarding glucose tolerance dynamics have been incorporated into the text (Page 8, lines 23-31).

Are the growth curves significantly different, can they be expressed as % weight gain? What is the effect of pup number in a litter on weight gain? The body condition and metabolic measures would greatly benefit from the inclusion of WT on WT data to know what “normal” is.

We agree with the reviewer and conducted additional cross-fostering experiments, as described. Additionally, we reorganised the figures, and the revised panel now appears in Figure 5B, displaying all significant differences. To ensure consistency, all litters were standardised to 5–6 pups, and this information has been added to the Methods section (Page 12, lines 8-9).

Discussion

“These findings indicate that ZFP57^{m/+} offspring cross-fostered to WT dams whose genotype was different from their birth mother, showed an inferior metabolic profile than those raised by ZFP57^{-/-} dams whose genotype matched their birth mother” because they became heavier pre-weaning due to “better postnatal” environment?

Zfp57^{m/+} offspring exhibited better metabolic adaptation when raised by a *Zfp57^{-/-}* dam, whereas those raised by a WT dam showed exacerbation of pre-existing metabolic syndrome hallmarks. We have revised the sentence for clarity (Page 10, lines 33-37). The measured outcomes included weight at weaning, fat content, and fat oxidation.

“Our study identifies ZFP57 as a key modulator of postnatal nutritional resources, specifically affecting mammary gland development and milk production.” This only affects offspring if the offspring exposed to it is metabolically imprinted in utero due to epigenotype and maternal in utero environment.

Our new data, which include cross-fostering WT pups to either *Zfp57^{-/-}* or WT dams, indicate that this effect is also relevant for cross fostered normal offspring that have not experienced *Zfp57* deficiency in utero. Despite developing in a typical environment, these pups still exhibit altered weight gain during maternal lactation indicating a postnatal effect.

Methods:

Minor notes

Animals: it is much appreciated that the nulliparous animals were oestrus matched, but it would be helpful to mention at which stage as the stage affects the morphological appearance of the gland and thus the interpretation of the null phenotype.

We added this information to the Methods section (Page 11, line 38).

For the cell sorting, only used collagenase without any other means of dissociation of cell clusters you usually obtain only using collagenase? Did the authors assess the number of single cells obtained

We also used Trypsin-EDTA which we omitted to note previously, and have now included this information in the Methods section. Additionally, we assessed the number of sorted single cells obtained, which averaged 350,000 for nulliparous glands and 750,000 for lactating glands. These details have also been added to the Methods section (Page 13, lines

32-33, 43-44).

Litter size matters in the stimulation of milk production, a smaller litter on a not optimally functioning gland can still do well, while a large litter would show lactation defects. What were the litter sizes?

Litter sizes were standardised to 5-6 pups/litter. This is now included in the methods section (Page 12, lines 8-9).

Cells sorting: Only collagenase used to dissociate cells, no further dissociation, with trypsin or otherwise? Seems unusual as most often another dissociation step or enzyme is used to obtain single cells.

We also used Trypsin-EDTA which we omitted to note previously, and have now included this information in the Methods section (Page 13, lines 32-33, 43-44).

“Enrichment was calculated as per cent input” >>percent input

Corrected.

Reviewer #2 (Remarks to the Author):

Zfp57 is a regulator of postnatal growth and life-long health

The work presented explores the function of Zfp57 in mice focusing on the relationship between dams and their pups. Considerable work has been undertaken and there are some interesting phenotypic findings presented. The authors' main conclusion is that "Zfp57 functions in postnatal resource control via the mammary gland" with "life-long impacts on offspring metabolic health". Due to the complexity of the study and some lack of information in introduction and methodology, it is challenging to determine whether the authors' interpretations are supported.

We have conducted substantial additional experiments and revised the manuscript, we believe these revisions have substantially improved the manuscript and support our conclusions. Our responses are detailed below.

The authors state in their introduction "In mice, deletion of the maternal gene in oocytes and the zygotic copies in early embryos causes severe loss of methylation at imprinted loci, resulting in embryonic lethality" – this needs considerable additional detail since the study uses Zfp57^{-/-}, Zfp57^{m/+} and Zfp57^{+/-p} mice.

We added additional details to the Methods section to clarify this point (Page 11, lines 41-43).

Interpretation of growth dynamics

The key issue lies with the use of genetically identical heterozygous offspring (Zfp57^{m/+} or Zfp57^{+/-p}) which may not be phenotypically identical. This is acknowledged in the discussion. The data presented could be interpreted to demonstrate that Zfp57^{m/+} or Zfp57^{+/-p} pups are phenotypically different with many of the findings due to these intrinsic differences, and not driven by resources from the mutant dam.

In this major revision we tested this and cross-fostered WT pups to both *Zfp57^{-/-}* and WT dams, allowing us to confirm that *Zfp57^{m/+}* exhibit distinct phenotypes compared to *Zfp57^{+/-p}*, including impaired suckling and increased weight gain throughout life. These cross-fostering experiments also helped disentangle the effects of maternal resource allocation. Our findings reveal that *Zfp57^{-/-}* dams produce milk with impaired composition, which is associated with reduced pup growth during lactation. Additionally, we observed synergistic interactions between pups and dams, driven by either co-adaptation or maladaptation. These findings indicate that both the pups' genotype and the resource allocations of their nursing mothers are important in offspring phenotype and that these two parameters interact with one another.

“*Zfp57^{m/+}* pups exhibited enhanced weight gain during lactation” and “Overall, this indicates that *Zfp57^{-/-}* mothers produce abnormal milk associated with growth disparities observed in their nursing pups, ultimately resulting in enhanced weight gain.” The authors are suggesting that the milk quality is responsible for weight gain. However,

- 1) The *Zfp57^{-/-}* mutant dams have longer gestations (Fig 4B) so their pups will be heavier at birth as reported in Fig4A
- 2) The *Zfp57^{-/-}* mutant dams have smaller litters (Fig 4C) again consistent with heavier birthweight as reported in Fig4A
- 3) The pups are all mutant. The authors assume that *Zfp57^{m/+}* offspring and *Zfp57^{+/-p}* offspring are identical apart from the genetic status of the mother but - unless I have misunderstood the experimental design - *Zfp57^{+/-p}* offspring come from sires with 100% loss of *Zfp57* in sperm and *Zfp57^{m/+}* offspring come from dams with 100% loss of *Zfp57* in oocytes – could this impact gene expression in these offspring? They may be genetically identical but *Zfp57^{m/+}* and *Zfp57^{+/-p}* offspring could have differences in gene expression as a result of loss of *Zfp57* in the respective parental germlines – which itself would be an interesting finding. Has this possibility been systematically excluded?

The reviewer is correct that *Zfp57^{m/+}* and *Zfp57^{+/-p}* offspring are genetically identical, but because of a requirement for maternal ZFP57 in the oocyte (but not paternal ZFP57 in the sperm) to maintain imprints after fertilisation, they are epigenetically distinct. Our previous work demonstrated that methylation at specific differentially methylated regions (DMRs) within some imprinting control regions is compromised in *Zfp57* mutants. There are, however, notable differences in imprint methylation between *Zfp57^{m/+}*, zygotic *Zfp57^{Z/-}*, and maternal-zygotic *Zfp57^{MZ/-}* embryos indicating contributions of both maternally and zygotically expressed ZFP57 to maintain imprints in mice, deletion of the maternal and the embryonic copies in early embryos leads to a severe loss of methylation at imprinted loci, resulting in embryonic lethality. Deletion of the zygotic *Zfp57* copies in homozygotes causes partial neonatal lethality as does absence of maternal *Zfp57* in oocytes. It is noteworthy that each imprinted DMR has a different demethylation pattern in the different mutants (Takahashi et al., Cold Spring Harb Symp Quant Biol, 2015; 80:177-187, doi:10.1101/sqb.2015.80.027466).

- 4) Fig4H – the growth dynamics of *Zfp57^{m/+}* and *Zfp57^{+/-p}* offspring are different but this could be interpreted to mean that *Zfp57^{m/+}* pups have intrinsically enhanced weight gain and/or *Zfp57^{+/-p}* pups have restricted weight gain as there is no WT control group

We added WT pups born to WT dams as a control group in Figure 4H, demonstrating that their weight gain closely resembles that of *Zfp57^{+/-p}* pups raised by their *Zfp57^{-/-}* mother. While *Zfp57^{m/+}* pups have a higher initial birth weight, their weight gain remains elevated even after normalisation to birth weight and expression as percentage weight gain. This additional analysis has been included in Supplementary Figure 6F.

Similarly, it is not possible to conclude “Together, our findings indicate that early-life exposure induces metabolic reprogramming in *Zfp57*^{m/+} offspring, leading to long-lasting alterations in weight gain, body composition, metabolic rate and fat oxidation. ” Without a WT group, these findings in Fig5 could be interpreted in a number of ways

The critical experiment to tease apart the relationship between maternal *Zfp57* and offspring *Zfp57* is the fostering experiment. The authors have done well to illustrate the experimental design. It is, however, a complex experiment.

We greatly appreciate this comment and conducted extensive new cross-fostering experiments, in which WT pups were fostered by either WT or *Zfp57*^{-/-} dams (Figures 4-6, Supplementary Figures 7-9). By tracking these pups for 6 months, we were able to confirm that in-utero effects play a significant role in shaping the *Zfp57*^{m/+} phenotype, influencing both birth parameters and long-term metabolic outcomes. Furthermore, our results indicate that *Zfp57*^{-/-} dams produce milk with an altered composition, which is associated with reduced pup growth during lactation. Additionally, we observed synergistic interactions between pups and dams, driven by either co-adaptation or maladaptation.

If I have interpreted Figure 6A and B correctly (and I may have this wrong)

Zfp57^{m/+} pups are generally heavier than fully WT
Zfp57^{m/+} pups fostered to WT dams are the heaviest
This could indicate an intrinsic growth advantage somewhat lost when the dam is mutant

Zfp57^{+/-} pups are lighter when fostered to *Zfp57* mutant dams but not when raised by WT dams – suggests *Zfp57* mutant dams fail to provide sufficient nutrients.

The simplest interpretation is that loss of *Zfp57* in the female germline results in growth advantage for pups and loss of *Zfp57* in the male germline results in growth disadvantage for pups – most evident when their mothers are also mutant

WT pups raised by hom dam do not gain more weight excluding a role for maternally-driven weight gain.

Again, I may have misunderstood the experimental design.

The reviewer has correctly identified the phenotype we observed. Given the lifelong metabolic consequences observed in *Zfp57*^{m/+} offspring, as well as the synergistic effects when they are raised by a *Zfp57*^{-/-} dam, we propose that bigger is not always better. In this case, excessive early growth appears to have detrimental metabolic effects that become apparent later in life.

Some information is missing from M&M for this work. M&M states “Heterozygous pups born to either *Zfp57*^{-/-} x WT (mat x pat) or WT x *Zfp57*^{-/-} (mat x pat) crosses within 24 hours of each other were used for cross-fostering” so where do the WT pups originate? More detail required. Was the WT colony bred and weighed alongside the experimental colonies? Although not indicated as significant, the overall WT control weight data appears lower than – for example - *Zfp57* ^{+/-} pups with WT dams.

Also, not clear why only data for male pups presented?

We thank the reviewer for identifying these missing details, which have now been added to the Methods section (Page 12, lines 1-3). WT pups were bred from WT x WT crosses and weighed alongside the experimental colony. Additionally, we incorporated new experiments

with WT controls, including weight data and statistical analyses, now presented in Figure 5. The data in Figure 5B specifically show male weights, while female weights, which follow the same pattern, have been included in Supplementary Figure 7.

Maternal behaviour

M&M and Figure 4F.

The details for assessment of maternal behaviour are insufficient and the indicated reference 2 is “Stringer, J. M., Suzuki, S., Pask, A. J., Shaw, G. & Renfree, M. B. Selected imprinting of *INS* in the marsupial. *Epigenetics Chromatin* 5, 14 (2012)” which is obviously incorrect. Detailed methods should be presented especially as the tests undertaken are not the standard ones done to assess maternal behaviour.

We have included additional details (Page 12, lines 21-26) and corrected the error regarding the reference, which has now been removed.

“Latency of *Zfp57*^{-/-} and WT dam to approach and retrieve their pups. n=64-65 pups from 9-10 litters” The data presented is for individual pups. This is not how data is analysed. It looks like a specific retrieval test was not undertaken. It would still be possible to present data for the 9-10 dams – ie latency to sniff 1st pup, latency to retrieve 1st pup and time to retrieve all pups to nest – assuming litter sizes comparable.

We reanalysed the data as requested by the reviewer and now present it in Figure 4F, showing latency to approach the first pup, latency to retrieve the first pup, and latency to retrieve all pups. All litters were standardised to 5–6 pups.

Introduction

“deletion of the maternal gene in oocytes and the zygotic copies in early embryos causes severe loss of methylation at imprinted loci, resulting in embryonic lethality. Deletion of the zygotic *Zfp57* copy causes partial neonatal lethality?” I am struggling with the sentence – can the authors rewrite

We added additional details to the Methods section to clarify this point (Page 11, lines 41-43). and rephrased this sentence (Page 1, lines 44-45).

Mammary development – when does the mammary gland start to differentiate? What hormonal signals? More detail required.

We have incorporated additional details into the introduction (Page 2, line 15).

“little is known” Review all the other imprinted genes implicated in mammary gland development/function?

We have expanded the text with additional details (Page 2, lines 18-20).

Results

Zfp57 - Non italicised capitals suggest the authors are referring to the protein. But, for example, the expression data is mRNA. Can the authors use *Zfp57* (lower case and italics) when referring to mouse gene/mRNA product - and all mouse genes/mRNAs.

We appreciate the reviewer for highlighting this, and we have corrected the formatting by italicising all gene names throughout the manuscript.

“In adults, Zfp57 was highly expressed in organs such as the placenta,” – the placenta is not an adult organ

Yes, we have changed the wording of this sentence (Page 2, lines 35-36).

The authors state “lower expression in adult somatic tissues including the lung and mammary gland” and then later “The placenta and mammary gland share similar functions, supporting offspring growth through nutritional resource control⁶. This led us to hypothesise that Zfp57 evolved as an upstream regulator supporting pre- and postnatal offspring growth.” This doesn’t work logically. The authors can justify their focus just by saying imprinted genes are renown for regulating fetal ad postnatal growth raising the possibility that a master regulator might similarly influence these same parameters.

We appreciate this comment and have revised the wording of the sentence accordingly (Page 2, lines 35-42).

Similarly “Maternal behaviour impacts lactation performance and involves imprinted gene function in the hypothalamus and pituitary gland^{15,26}” Not clear what the authors mean here? In what way does maternal behaviour impact lactation performance?

“involves imprinted gene function in the hypothalamus and pituitary gland^{15,26}” Again, not clear. What do the authors mean by “involves”?

We have clarified this sentence to better convey our intention, which was to highlight that hormones such as oxytocin, cortisol, and prolactin produced elsewhere influence mammary gland function including milk production and let-down. Additionally, we have included citations showing that the absence of several imprinted genes has been associated with impaired maternal care and reduced milk release. Given that *Zfp57* is highly expressed in the brain and that we show that imprinted genes are perturbed in the maternal brain regions in mutants, this may be a contributing factor. We have refined the wording and added further details for clarity (Page 3, lines 5-8).

“we quantified the expression of Zfp57-regulated imprinted genes⁸ in adult mice hypothalami and pituitary glands” Male? Female? Add Zfp57-deficient or Zfp57^{-/-}

We analysed expression in both sexes using six biological replicates (three males and three females). We have corrected the sentence to include the missing details and updated the Methods section accordingly (Page 3, line 10 and Page 11, lines 39-40).

“a reduction in Phlda2 levels which may affect maternal care²⁷” why would reduced Phlda2 in adult mice hypothalami and pituitary glands impact maternal care?

The cited paper demonstrates that dams exposed to a lower Phlda2 dose spent more time nursing and grooming their pups. Since the study did not use a tissue-specific deletion model, the observed phenotype could originate from multiple tissues, including the hypothalamus and pituitary gland.

“To investigate Zfp57 expression in the developing mammary gland” technically it has already developed by lactation day 2 (which I presume is postnatal day 2?).

We agree with the reviewer and corrected the wording from “developing” to “lactating” (Page 3, line 22). Lactation day 2 is postnatal day 2.

“To evaluate the presence of Zfp57 protein in mammary glands”– did the authors use, for example, IHC or in situ to examine sites of expression?

We agree with the reviewer that IHC or in situ analysis would have been valuable. However, despite our efforts, the available antibodies against ZFP57 are not suitable for IHC. Instead, we provide expression data in a cell-type-specific manner using sorted mammary gland cells (Figure 1D).

“gestation days 4.5, 9.5, and 14.5,” When referring to events in the dam, it is usual to state whole days ie gestational days 5, 10 and 15.

In this study, we used timed matings, and since mice typically mate during the dark phase of the light/dark cycle, we checked for plugs both in the morning and evening. We designate gestation day 0.5 for plugs detected in the morning and gestation day 1 for those found in the evening. We believe that using half-day designations provides greater accuracy.

The authors should provide weight data for mutant whole mammary glands as a proportion of total body weight.

We did not specifically measure the weight of the mammary glands during dissection. However, dam weights at the time of dissection showed no significant differences, and the mammary tissue did not appear to vary in size.

“This indicates that Zfp57 contributes to normal mammary tertiary branching and that its absence leads to precocious development, potentially impacting tissue functionality” The authors need to specify that this phenotype could be due to either local deficiency or deficiency at another site. Needs some careful wording here.

We agree with this comment, this sentence has been corrected (Page 4, line 36).

“Quantification of Ki67-positive cells showed that Zfp57^{-/-} nulliparous glands exhibited positive Ki67 cells in contrast to WT controls, consistent with histological analysis. During gestation days 4.5, 9.5 and lactation day 2, no significant differences were observed (Figure 2C-D), further indicating premature proliferation in Zfp57^{-/-} nulliparous glands” This sentence needs to be worded more carefully ie are there only Ki67 +ve cells in mutant at G15 (14.5)?

We appreciate the feedback and recognise that the previous version lacked clarity. We have revised the sentence accordingly (Page 5, lines 22-28).

“we analysed transcriptomes from 120 sorted cell populations at various stages: nulliparous, gestation day 9.5, and lactation day 2” – why not 4.5 and 14.5 to be consistent?

For RNA analysis, we selected a single mid-gestational time point as a representative, along with assessments at the nulliparous stage and during lactation. Figure 3 includes data from multiple gestational time points, including days 4.5, 9.5, and 14.5, expanding on the phenotypes observed.

“Importantly, most of the DEGs were observed in luminal progenitor cells (Figure 3D), exhibiting minimal expression of Zfp57 (Figure 1C).” Not clear what the authors mean here?

We have added further explanation and refined the wording for clarity (Page 4, lines 14-16). Essentially, we highlight that the most affected cell type in the mammary gland expresses low levels of *Zfp57*, suggesting that the observed effect may be indirect or a consequence of *Zfp57* loss during early development.

“We found that *Zfp57^{m-/+}* pups had delayed weight recovery when reunited with their *Zfp57^{-/-}* mothers compared to *Zfp57^{+/-}* pups with their WT mothers.” Difficult to interpret as the genotype of both dams and offspring is different.

We agree with this comment. To address it, we included two groups of WT pups cross-fostered to either WT or *Zfp57^{-/-}* dams (Figure 4G). Based on these findings, we can now conclude that *Zfp57^{m-/+}* pups exhibit a suckling defect, while WT pups nursed by *Zfp57^{-/-}* dams show normal weight recovery after being reunited with their foster mothers.

“receiving suboptimal maternal care” The authors do not show suboptimal care. Just delayed retrieval - which is a different factor. Remove.

We have removed this sentence.

M&M

129aa background – can the authors be more specific? Is this a long term “in house” colony? If not, state supplier

Yes, this study was conducted using a long-term in-house colony of 129Sv mice. We have now included this information in the Methods section (Page 11, lines 31-32).

Mice were mated at 12-16 weeks of age – any particular reason for not mating earlier

We aimed to ensure that the females’ mammary glands were fully developed before mating. In mice, mammary gland development continues even at 8 weeks of age. To ensure they were fully post-pubertal, we initiated mating at 12 weeks. This clarification has been added to the Methods section (Page 11, lines 38-39).

Milk let down on postnatal day 9 /mouse milking on lactation day 8 – can the author settle on one term

We have standardised the wording, and it now consistently appears as lactation day 8.

Relative expression was normalised to β -Tubulin expression – can the authors confirm that only one reference gene was used? Has this gene shown to be stably expressed during mammary gland development and lactation? Cite reference

Both *β -Tubulin* and *β -Actin* were used across the manuscript for normalising qPCR data. *β -Tubulin* was used in the mammary gland. Our RNAseq data indicates that this gene is relatively stable during mammary gland development.

Supplementary Figure 1C Presume the axis is log scale – please add to label

We have moved this panel to Figure 1A with additional data on brain embryonic expression of *Zfp57*, and corrected the label on the axis.

Serum steroid hormones – why gestational d9.5 (also should be labelled gestational D10 or embryonic day 9.5) and lactation d2?

We selected these time points to align with those used for mammary gland dissection, representing mid-gestation and early lactation.

Supplementary Figure 5E – does the significance remain after testing for multiple measures made in P0,P1 and P2?

Yes, we performed this correction in the statistical analysis.

Reviewer #3 (Remarks to the Author):

Hanin et al present an interesting manuscript investigating the role of ZFP57 to regulate postnatal growth and life-long health. It is known that early life factors, such as pre- and post-natal nutrition, alter long term health via epigenetic mechanisms. ZFP57 is an epigenetic regulator of genomic imprinting with a known role in prenatal growth, and here they identify an imprinting-independent function of ZFP57 in postnatal control via the mammary gland.

The authors allude to a role for ZFP57 to regulate maternal care – could that be primary or secondary to the effects on offspring nutrition? This could be of interest to further elucidate the effect of functional changes in ZFP57.

We appreciate this comment however determining whether this is a direct effect or a secondary consequence remains challenging and a substantial study outside the scope of this manuscript. Uncoupling maternal behaviour-related phenotypes from lactation-specific effects would require tissue-specific knockout models, which are not currently available.

Are there any other differences in the ZFP57^{-/-} mice that could contribute to their mammary gland development? Is their milk production normal? Is this a secondary effect on offspring development?

We are not aware of any additional differences in *Zfp57^{-/-}* mice. Based on the revised suckling experiment added to the manuscript, there is no indication of impaired milk production. WT pups nursed by a *Zfp57^{-/-}* dam after a separation period (Figure 4G) gained weight rapidly, suggesting sufficient milk production.

In the discussion, we state: “Overall, this suggests that ZFP57 is not acting directly in the mammary gland but instead influences the tissue indirectly, possibly through developmental effects, transcriptional defects, inter-organ communication, or secondary effects of other genes.”

The cross-fostering data is interesting and clearly presents a role for the milk/mammary gland to contribute to offspring development. The authors also present a last metabolic phenotype in the offspring – what is the mechanism for this? Changes in body weight appear to contribute, as do altered fat oxidation in the offspring – is the fat different? This could be an important addition as it could allude to changes in adipose tissue function which could help further define the mechanism for ZFP57 to affect postnatal development.

Our new data, including cross-fostering WT pups to either WT or *Zfp57^{-/-}* dams, confirms that the long-term metabolic phenotype observed in *Zfp57^{m/+}* offspring originates from in-utero influences. In contrast, WT pups nursed by *Zfp57^{-/-}* dams exhibit effects only during the lactation period, with no lasting impact later in life. The precise mechanism underlying this difference remains unclear. We anticipate that systemic changes in these animals may be influenced by both in-utero and postnatal factors.

Reviewer #4 (Remarks to the Author):

In the manuscript entitled, ZFP57 is a regulator of postnatal growth and life-long health, the

authors present an intriguing discovery on the Kruppel-associated box-containing zinc-finger protein (ZFP57). ZPF57 is a master regulator of imprinting that acts to recruit DNA methyltransferases that lack the targeting domains required to recognize imprinting control regions. This is a confusing manuscript to read and not targeted to a general scientific audience as it lacks sentences that introduce the experiments conceptually and sentences that synthesize the potential meaning of the results.

We recognise that the previous version lacked clarity and did not fully communicate the experiments and key findings effectively. To address this, we have made substantial revisions to both the experiments, text and figures, ensuring a clearer, more coherent message while simplifying the presentation of experiments and conclusions.

Although ZPF57 is a master regulator of imprinting in the embryo, the authors find that it functions in an imprinting-independent fashion in the adult mammary gland. And, while there are a host of phenotypic changes in the mammary gland (involving alveologenesis), ZPF57 is expressed only in basal epithelial and stromal cells, even though it appears to function by regulating the number and the function of alveolar progenitor cells (or at least luminal progenitor cells as shown here) -- a conundrum. Following the phenotypic analysis, the authors present a series of mating and cross-fostering experiments that further demonstrate how complex the consequences of ZPF57 loss is. Altogether, the data from cross-fostering suggest that concordance between gestational and nursing mothers' genotypes is important in the circumstance of ZPF57 loss and support the notion that there is in-utero adaptation to postnatal resources provided by the dam. This is an interesting message but the studies in the second half of the manuscript provide little insight into the studies in the first half. The question remains; how does ZPF57 mechanistically function in the mammary gland? How does it transcriptionally regulate mammary- and milk-related genes that govern nutritional provisioning? The authors do little to synthesize the manuscript's message, offering a one sentence paragraph in the discussion about the model. The bottom line is that this is an interesting study, but it remains preliminary and more experiments that address the mechanism of ZPF57 action are required. Below are more detailed comments on the figures.

Figure 1 starts out with data better shown in a supplementary figure – RT-pPCR panel of imprinted gene expression in WT and ZFP57^{-/-} post-mitotic tissues. We learn that the absence of ZFP57 in the embryonic brain affected the expression of several imprinted genes at E12.5, including *Zac1*, *Rasgrf1* and *Nnat* (Figure 1A). But does it affect genes (e.g. *Igf2* and *H19*) in the reciprocal fashion typical of imprinting (*H19* expression is not examined)? Why is it important that both *Rasgrf1* and *Nespa5* showed a similar pattern? Does this suggest that in these tissues ZFP57 is not functioning in its canonical role as a master regulator of imprinting? Is this the conclusion? I ask because the authors simply state that the data show that ZFP57 regulates the expression of genes in adult tissues, which is a modest conclusion and why I suggest either the data belong in a supplementary figure or the authors interpret the data in a way that moves this manuscript's conclusions forward. In this figure they also FACS purify subpopulations of mammary epithelial cells to look at ZFP57 expression and find it expressed only in stromal and basal epithelial cells. A western of whole LD2 mammary glands reveals protein in WT but not KO tissue.

We have added *H19* expression data at E12.5. The significance of the imprinted genes we highlight lies in their increased sensitivity to DNA methylation changes, as demonstrated by our group in Takahashi et al., *Genes Dev.* 2019. In this study, we are investigating the zygotic deletion of *Zfp57*, meaning maternally derived *Zfp57* remains present. However, some of the imprinted genes analysed are still regulated by *Zfp445*. Among the genes we tested, both *Zac1* and *Rasgrf1* are controlled by an imprinted control region that undergoes complete loss of DNA methylation following zygotic *Zfp57* deletion.

In response to the reviewer's recommendation, we have reorganised the data, moving the

embryonic findings to Supplementary Figure 1C. To enhance clarity and coherence, we now present the adult data in Figure 1, which includes qPCR analysis of *Zfp57* levels in adult tissues and ESCs. Additionally, this panel now features *Zfp57* expression in the embryonic brain.

Rasgrf1 has previously been shown to influence pup growth, potentially linking it to lactation, while *Nespas* exhibits a similar pattern in the pituitary gland and has been associated with social behaviour. Given their potential relevance to our study, we have referenced these findings. Overall, our results suggest that ZFP57 may have distinct roles during embryonic and adult stages.

Figure 2 shows a phenotypic analysis of the mammary gland over the time course of gestation. It was initially surprising to me that the ZFP57^{-/-} dams could be evaluated since I understood from the literature that *Zfp57*-null mice exhibit embryonic lethality and loss of imprinting at many loci (Li et al., Dev Cell, 2008). But, it appears the loss-of-function effect is variable. Does this variable effect influence the interpretation of phenotypes in ZFP^{-/-} adults?

We recognise that the different types of *Zfp57* deletions were not clearly communicated in the previous version. To clarify this, we have revised and restructured the introduction and results section.

Deletion of *Zfp57* in mothers resulting in null oocytes and also the zygotic copies in early embryos leads to a severe loss of methylation at imprinted loci, resulting in embryonic lethality (maternal-zygotic mutants). In contrast, in animals homozygous for the mutation derived from heterozygous parents also resulting in null embryos, partial neonatal lethality occurs. In this study, we specifically focus on null mothers who have survived from that cross.

The data show that nulliparous ZFP57^{-/-} glands are more branchy. N=3 MGs were quantified for the analysis but is this phenotype observed in every surviving ZFP57^{-/-} dam (i.e. what is the penetrance)?

Yes, we observed this in all *Zfp57*^{-/-} females tested.

I note that the primary ductal structure in the ZFP57^{-/-} gland is similar to WT and what is being observed is an increase in secondary/tertiary branching. The authors should clarify their quantification. The reason the distinction is important is that the primary ductal structure is generated by endbud bifurcation and a defect in this would indicate a different mechanism than a defect in secondary/tertiary branching. To me, the precocious secondary/tertiary branching suggests precocious alveologenesis or animals in diestrus (the methods say nulliparous glands were estrus matched, an important point). It is unclear how duct number was quantified and how this would differ from a branch point analysis. Interestingly, the phenotype is switched at pregnancy with ZFP57^{-/-} glands becoming less “branchy” than WT; it appears that there is almost no tertiary budding in the KO at this timepoint. However, the KO gland catches up over time.

All animals used for the nulliparous stage were in oestrus. We analysed the Carmine Alum-stained images using Fiji and the “AnalyzeSkeleton” plugin. This approach allowed us to transform the raw image of the ducts into a skeleton-like representation recapitulating the structure of the mammary gland and quantify junctions, triple and quadruple points, and branches.

The number of ducts corresponds to the total number of branches in the whole mount, while total branching points represent the number of junctions, including both secondary and tertiary branches. We have added this information to the Methods section (Page 17, lines 6-8) and updated the Y-axis titles in Figure 3B for greater clarity.

Next the authors FACS purify cell populations at 3 timepoints and show a number of things including that the KO glands contain more luminal progenitor cells at LD2. To be clear, luminal progenitors are not alveolar progenitors, which can be distinguished by *cKit* expression (Shore et al., Plos Genetics, 2012); the analysis of AVPs may have provided better insight into the observed phenotype.

We used flow cytometry to compare cellular proportions between *Zfp57^{-/-}* and WT cells. We appreciate the importance of additional insights into luminal cells and have explored the RNA-seq data from sorted mammary cells to address this point. Our analysis indicates that *cKit* expression in the luminal compartment is similar between *Zfp57^{-/-}* and WT cells, suggesting that we are unable to pinpoint a specific progenitor population associated with the observed phenotype.

The authors immunostain for Ki67 and obtain results that correspond to the phenotypic images: more proliferation in the KO in nulliparous glands, less at PD4.5, more at PD9 and no difference between genotypes at LD2. The issue here is that the authors quantify Ki67 by relative intensity....meaning that they are quantifying the intensity of staining across the nuclei (as shown by DAPI) rather than the proliferative index (% positive nuclei), which is the standard. It is true, there appears to be graded staining at PD4.5, maybe justifying an intensity approach to quantification. There is recent data to suggest that a cell's Ki67 expression level represents both its phase in the cell cycle and its cell cycle history (Miller et al., Cell Reports 24, 7/2018), but interpreting grades of Ki67 requires a more rigorous analysis. The authors should either apply this rigor or only count the percentage of strongly positive nuclei. Also intensity is spelled wrong on the Y-axis.

We appreciate the reviewer for highlighting this point. As suggested, we have revised the quantification to reflect the percentage of positive cells/nuclei (Figure 3D). To ensure accuracy, we applied a threshold that considers a cell positive only if its Ki67 signal exceeds a defined intensity level, excluding nuclei with faint signals from the count.

This criticism applies to the TUNEL analysis as well, where the standard is the percentage of positive cells (not TUNEL area/duct area). Here, the immunostaining is heterogeneous, perhaps due to background because I also see staining around nuclei rather than in just in nuclei; a nuclear stain (DAPI) would be helpful for interpretation. The authors perform RT-qPCR and see increases in markers (e.g. GATA3, Elf5 etc) consistent with premature alveologenesis in the KO animals.

We have revised the data to now display the percentage of positive cells (Figure 3F). To ensure a comprehensive representation of the tissue, we used 10µm sections. Due to differences in focal planes, Ki67 signals may sometimes appear outside the nucleus. However, our quantification includes background correction, and we use 7AAD as a nuclear marker, which is shown in red.

In Figure 3 the authors present transcriptome analysis from 120 sorted cell populations. Please make it clearer what “120 sorted populations” represent (from the methods: “Libraries for RNA sequencing (RNA-seq) from sorted mammary cells” which does not provide detailed information. How many animals at what stages and genotype?

The libraries consist of 5 distinct mammary cell populations: stromal, basal, endothelial, luminal progenitors, and luminal differentiated cells. We selected 3 representative stages—nulliparous, gestation day 9.5, and lactation day 2. Each time point includes samples from 8 animals, with 4 per genotype. These details have been added to the Methods section (Page 14, lines 27-32).

In any case, this analysis showed that variation is primarily affected by cell type and stage but not genotype. But they do see genotypic separation in luminal progenitor during gestation. What do the authors make of this observation? And is there genotypic separation in genes that regulate milk protein and milk lipid expression or just in genes that can be imprinted?

Figures 2A and 3B illustrate that the majority of transcriptional changes are driven by cell type and developmental stage, which is expected given that different mammary cell populations have distinct functions, reflected in their transcriptional profiles. A clear separation by genotype would indicate a widespread transcriptional disruption, potentially leading to mammary gland dysfunction. However, a closer examination of each cell population individually (Figure 2C) reveals that certain populations, such as luminal progenitors during gestation and stromal cells during lactation, show some genotype-dependent differences. While these changes are not dramatic, they prompted us to further investigate differentially expressed genes (Figure 2D) and perform a Gene Ontology term enrichment analysis (Figure 2E), which identified a GO term related to milk composition. The list of differentially expressed genes has been added as Supplementary Table 3. Among these genes, several are associated with milk lipid metabolism and imprinting, including *Plin2* and *Fas*, as well as imprinted genes such as *Meg3*, *Igf2*, *Rasgrf1*, *Snrpn*, and *Plagl1*. While these genes likely contribute to the observed phenotype, their differential expression may be an indirect effect of *Zfp57* mutation.

To evaluate whether ZFP57 is functioning in its capacity as a master regulator of genomic imprinting, differentially expressed genes (DEGs) were examined. Blue dots (other DEGs) apparently represent any significantly regulated gene (either a clearer explanation or the graphs clearly delineating significance would be helpful). But there is a difference between those imprinted genes that are differentially expressed and not. There are only 15 DEG imprinted genes in the KO and none of those in the same imprinted control region show reciprocal behavior, which suggests that ZFP57 is not functioning in its capacity as a master regulator of genomic imprinting.

The Y axis in Figure 2D represents significance, displayed as $-\log_{10}(\text{FDR})$. The dots in blue and red represent DEGs which show significant differences between the genotypes. We have added additional explanation to the figure legend.

Most of the DEGs were observed in luminal progenitor cells and the authors link this to a potential role for ZFP57 in early developmental stages. Are the authors suggesting that ZFP57 functions in its role as master imprint regulator at earlier stages and this is responsible for the postnatal phenotype? What do the authors mean by “early developmental stages”-- in the embryonic mammary gland?

We propose that ZFP57 serves a dual function, prenatally as a master regulator of imprinting, as extensively studied by our lab and others, and postnatally as a regulator of nutritional resources through the mammary gland. A limitation of our model is that it involves a global *Zfp57* deletion rather than a tissue-specific one, making it challenging to precisely determine the location and timing of ZFP57 activity. As a result, the exact mechanism remains unclear and will require more refined mouse models for further investigation. Given that *Zfp57* is highly expressed in embryonic stem cells (as shown in Figure 1A) and that the most pronounced effects in the mammary gland occur in luminal cells, which share stem-like properties, it is possible that ZFP57 plays a role in this tissue during embryonic development, with the observed differential expression being a downstream consequence. Additionally, we raise the possibility in the text that other tissues may also be involved. We have expanded the discussion to further explore these potential mechanisms (Page 10, lines 4-11).

This is interesting and may hold the key to understanding ZFP57 function, but the GO terms are all about lactation so the manuscript is back to the investigating the phenotype. The authors conclude that loss of ZFP57 generates precocious development of the mammary gland independent of its role as an imprinting regulator and in the discussion, they state that their data suggest “ZFP57 is not acting directly in the mammary gland.” But intrinsic/extrinsic to the mammary gland is relatively easy to test because the mammary gland can easily be transplanted (Lawson DA, et al., Cold Spring Harb Protoc. 2015 Dec 2;2015(12). One can also test whether hormones(extrinsic) are responsible by ovariectomy followed by estrogen/progesterone treatment. A major deficiency in the manuscript is that this is where mechanism is dropped since “extensive in-vivo analysis is required to elucidate these possibilities.” But, these analyses should be done.

We appreciate the reviewer’s point and acknowledge its importance. However, our expression data (Figure 1A, D) indicate that *Zfp57* is not highly expressed in mammary gland cells, suggesting that the precocious mammary gland development observed may be a secondary effect driven by systemic influences. The ovaries, which express high levels of *Zfp57*, are a compelling candidate, however, our measurements of serum progesterone and 17 β -estradiol showed no significant differences between WT and *Zfp57*^{-/-} dams in nulliparous females (oestrus-matched), at gestation day 9.5, or at lactation day 2. This data has been added to the supplementary information (Supplementary Figure 6C-D). The precise developmental timing of ZFP57’s function remains undetermined and may occur early in development, with systemic consequences manifesting in adulthood. While ovariectomy could provide some insights, it would only address a narrow developmental window, can only be performed after weaning, and may have limited applicability in fully elucidating the mechanism.

Furthermore, if the authors find that ZFP57 is exerting its role through regulating gene transcription extrinsically through hormonal control, then ZFP57 ChIPseq, which has been performed by this group (Shi H, et al., Epigenetics Chromatin. 2019 Aug 9;12(1):49. doi: 10.1186/s13072-019-0295-4), could potentially be used in the pituitary gland to understand ZFP57 in this context. Using these types of in vivo analyses to investigate ZFP57 control of postnatal gene expression in luminal progenitor cells would achieve a more mechanistic understanding of the role of ZFP57 in mammary alveologenesis and a more impactful study.

Our group previously performed ChIP for ZFP57 in ESCs, where its expression is significantly higher compared to adult tissues. However, when we attempted ChIP in adult somatic tissues, the low expression levels resulted in a poor signal-to-noise ratio, preventing us from obtaining meaningful insights. In the manuscript, we discuss the possibility that many of the observed phenotypes stem from early developmental events that have lasting effects on health throughout life.

Instead, the authors continued their analysis of ZFP57 by pursuing crosses that are typically used to understand imprinting. WT females crossed to a ZFP57^{-/-} males, generated paternal heterozygous pups (ZFP57^{+/-p}), and ZFP57^{-/-} females crossed to WT males generating maternal heterozygous pups (ZFP57^{m-/+}). This section was confusing to read.

We apologise for the lack of clarity in the original version of the manuscript. In this revision, we have made an effort to improve the presentation by clearly explaining the experiments and their justification at the beginning of each section, hopefully making the manuscript more straightforward and coherent. Additionally, the new in-vivo data have allowed us to draw further conclusions that were not included in the previous version. We believe these improvements have resulted in a significantly stronger manuscript.

But, the data show that m-/+ pups of ZFP57-/- X WT cross have delayed appearance of stomach milk spots. Yet, ZFP57m-/+ pups catch up and indeed, exhibit enhanced weight gain during lactation. Since altogether the data suggest either compromised milk let-down by ZFP57-/- dams, or a suckling issue specific to ZFP57m-/+ pups, the authors go back to characterizing differences between WT and KO dams but are unable to distinguish between these possibilities. They further find that KO dams have milk that contains higher levels of oxidized triglycerides and lower levels of phosphatidylcholines. The authors conclude that this “abnormal” milk may be associated with growth disparities observed in their nursing pups, and ultimately be responsible for enhanced weight gain, but they do not demonstrate it conclusively.

In this major revision, we incorporated a series of cross-fostering experiments to distinguish between developmental effects in *Zfp57^{m-/+}* pups resulting from gestation in a *Zfp57^{-/-}* uterus and those influenced by postnatal rearing by *Zfp57^{-/-}* dams. Cross-fostering WT pups to *Zfp57^{-/-}* dams confirmed that these dams attenuate pup growth regardless of genotype (Figure 5B). However, *Zfp57^{m-/+}* pups exhibit hallmarks of metabolic syndrome, an effect acquired in utero, characterised by increased birth weight and accelerated weight gain during lactation and throughout life. Additionally, the abnormal milk produced by *Zfp57^{-/-}* dams negatively impacts WT pups by reducing their weight gain, highlighting its influence independent of in-utero factors.

Question: Why are light and dark data presented in Figures 5G-J? These studies are not well explained or discussed, even at the end of the manuscript when they are brought up again.

The metabolic parameters we measured are influenced by circadian rhythms, light/dark cycles, and the animals' activity levels. To provide a comprehensive assessment of metabolic status, we present results from both the light and dark phases, enhancing our ability to classify behaviours more accurately. These data are now included in Figure 6F and Supplementary Figure 9. Additionally, we have provided further explanation in the Methods section and expanded on this point in the discussion (Page 10, lines 40-41).

Certainly, it is not new that childhood weight gain is a risk factor for adult health. Here the authors also explore this and find that ZFP57m-/+ offspring born to ZFP57-/- dams continue to gain more weight and have impaired glucose tolerance among other things as demonstrated by a multi-parameter metabolic assessment system. The bottom line is that the authors find, as others have, that early-life exposures (through lactation) can induce metabolic reprogramming that can have long lasting effects (Picó C, et al., Lactation as a programming window for metabolic syndrome. *Eur J Clin Invest.* 2021 May;51(5):e13482. doi: 10.1111/eci.13482.)

Cross-fostering comes next and it is important as it can help differentiate between parental and offspring effects on traits. What happens when the +/-p pups are fostered by ZFP57-/- dams? The answer is a severe failure to thrive during lactation but catch-up post-weaning. Moreover, ZFP57m-/+ pups cross-fostered to a WT dam gain even more weight during lactation and display obesity at 6 mos. There is also data showing gender specific effects and data showing effects based on light/dark; none of the data are well framed or explained.

We acknowledge that our initial explanation did not fully capture the complexity of maternal and offspring contributions. To address this, we have extensively revised both the text and figures. The additional cross-fostering experiments we conducted offer a clearer understanding of these interactions. Furthermore, we have refined the text to improve clarity and framing, and we hope that the current version presents the findings more effectively.

All told, the studies suggest that concordance between gestational and nursing mother's genotype is important in the circumstance of ZFP57 loss and supports the notion that there is in-utero adaptation to postnatal resources provided by the dam. This is interesting, but the authors provide no satisfactory explanation for the observation. Could this be due to the master imprinting function of ZFP57 embryonically or in "earlier developmental stages"? Or is it that ZFP57 functions to regulate hormonal control of lactation. Understanding this would shed light on what remains a mysterious transcriptional regulatory role of ZFP57 postnatally.

Our observations, including the newly added cross-fostering experiments, indicate that the phenotypes observed in *Zfp57^{m/+}* pups originate from the absence of ZFP57 in utero. Our previous work has shown that *Zfp57* reaches its peak expression during these developmental stages, where it functions as a master regulator of imprinting (Takahashi et al., Cold Spring Harb Symp Quant Biol, 2015; 80:177-187, doi:10.1101/sqb.2015.80.027466). Several ZFP57-regulated genes are known to influence metabolism, including *Zac1*, which is associated with transient neonatal diabetes, *Igf2*, a mitogen and growth enhancer and *Dlk1*, which regulates nutrient metabolism. It is possible that one or more of these genes contribute to the lifelong metabolic phenotype observed, and further targeted studies will be needed to investigate their specific roles. Additionally, we found no evidence that ZFP57 directly regulates hormonal control of lactation, and we provide these measurements in Supplementary Figure 6.

REVIEWER COMMENTS

Reviewer #1 (Remarks to the Author):

ZFP57 is a regulator of postnatal growth and life-long health

In this revised manuscript the authors have added additional cross fostering experiments that allowed the differentiation between in-utero effects and postnatal effects on pups of the lack of ZFP57, they also clarified and altered the writing extensively, which strengthens the manuscript. The data also shows that the altered mammary gland development, the resulting lactation and milk in the ZFP57^{nul} dam influences the growth and later health of offspring. Supporting their conclusion that ZFP57 is a regulator of both prenatal and postnatal growth and development and affect life-long health.

Interestingly, the results indicate mother-offspring co-adaptation where in-utero effects on offspring are partially mitigated by altered postnatal nutrition (and nurturing?). Something seen in nature where different mammals have adapted their lactation strategy to align with the prenatal development and postnatal requirements of their offspring. The exact mechanism of how ZFP57 affects the function of the mammary gland and how fetal/placental- mammary crosstalk occurs needs further investigation, but this is a compelling question as well as what drives the maternal adaptation to the metabolic challenges of the offspring.

Is it known if the m-/+ offspring has transient neonatal diabetes as with human ZFP57 mutations?

Thank you for this thoughtful feedback. We agree that the findings raise compelling questions about the mechanisms of maternal adaptation and mother-offspring co-adaptation. While we have not directly tested for transient neonatal diabetes in the *Zfp57^{m-/+}* offspring, we appreciate the relevance of this point given the human phenotype. This is an important avenue for future investigation and we have noted this in the discussion.

Some minor notes

P1- l25: "functions" ...phenotypes or ... influences multiple aspects of MG development and function. Might be better, as the function of the mammary gland is not to branch etc (minor quibble)

We have revised the text to "mammary gland phenotypes" to more accurately reflect our intention.

P2-L13-14: mammary gland is a key requirement for milk production? that reads weird. Mean to say: that the proper development and functional differentiation of the MG is a key requirement for milk production?

Thanks for this refinement, we have corrected the phrasing of this sentence.

P3-L7 absence of imprinted gene expression? or absence of the genes? Thank you for spotting this ambiguity. We intended to refer to absence of expression of imprinted genes rather the absence of the genes themselves and have revised the sentence accordingly.

P3-L28-30: sentence does not read well. We appreciate the feedback and have revised this sentence to improve readability.

P3-L41-43: this suggests that there are 120 different cell populations in the mammary gland, suggest rephrasing so the following is clear: x cell populations at Y stages, 2 genotypes in Z replicates (3 stages, 5 cell types, 2 genotypes-WT and null, 4 replicates?) **We have corrected the phrasing of this sentence.**

P4-L10-12: I think that is maybe a bit too gratuitous conclusion, without analysis of DNA methylation, ZFP57 binding and more detailed expression analysis. Maybe the mammary gland lacks factors needed for these genes to be expressed. **We revised this and limited the conclusion only to those 15 differentially expressed imprinted genes.**

How many imprinted genes are DEG in null embryos?

While we have not performed RNA-seq on null embryos, our qPCR data (Supplementary Fig. 1C) indicates that only a small number of imprinted genes, specifically *Zac1*, *Rasgrf1*, and *Nnat*, are differentially expressed in E12.5 brains.

This is consistent with previous extensive work by our group and others. For example, in Takahashi et al. (Cold Spring Harb Symp Quant Biol 2015; doi:10.1101/sqb.2015.80.027466), zygotic *Zfp57* knockout, as used in our study, resulted in partial loss of DNA methylation but only at a small number of imprinted genes – notably *Zac1*, *Snrpn*, and *Rasgrf1*. Further supporting this, Jiang et al. (PNAS 2020; <https://doi.org/10.1073/pnas.2005377118>) found that in whole embryos, these loci were among the most affected.

P4-L26: would like to argue that the liver is also an important other organ that contributes to mammary gland physiology (suggest adding here and in the discussion, especially as it has been shown that changes in imprinter genes affect the liver and alter lipid homeostasis during pregnancy, could that affect the lipid content of the milk/MG?) **We agree and added the liver to this sentence.**

P4-L42-44: it also looks like there are more alveolar structures in the virgin gland which would correlate with precocious development. this is one indication for precocious development, are there other indications? such as milk protein gene expression in luminal cells.? (in DEG analysis I see it in basal and endothelial cells??? that suggest cell populations that are not "clean") otherwise hyperplasia might be a better terminology?

We agree that the *Zfp57*^{-/-} appear to exhibit more alveolar-like structures, which alongside the positive Ki67 staining of the *Zfp57*^{-/-} nulliparous gland (Figure 3C) support the interpretation of precocious development.

Regarding the expression of milk protein genes: at this early stage, we did not observe a marked upregulation of key milk protein genes in the qPCR, which aligns with the fact that their robust expression typically occurs later in pregnancy. Nonetheless, we do see increased expression of key transcription factors involved in alveolar differentiation and early lactogenesis, including *Gata3*, *Elf5*, *Stat5a*, and *Stat6* (Figure 3G) which further supports the interpretation of premature developmental progression.

With respect to the differential expression of *Wap* in basal and endothelial populations, we note that while *Wap* is commonly used as a luminal marker, expression can occur in other cell types, particularly observed in RNAseq. This has

been reported in mammary epithelial and non-epithelial cells, including fibroblast (<https://doi.org/10.1038/s44318-025-00422-3>), and WAP-Cre–driven deletions have been observed in non-epithelial cells such as mammary adipocytes (<https://doi.org/10.1038/s41418-023-01146-9>).

Taken together, the histological findings, proliferation marker expression, and upregulation of developmental transcription factors point to precocious development.

Note on cell populations: MCAM and Krt8 expression in several populations suggest that the sorting did not result in clean populations which might be an issue in the interpretation of the mRNA-seq data. Enhanced by the expression of classic milk protein gene genes in basal and endothelial cell populations.

We appreciate the reviewer’s thoughtful comment. It is important to note that marker expression patterns can vary during lactation compared to the nulliparous state, and some degree of overlap between populations has been observed in previous studies (e.g., Bach et al., 2017 DOI: 10.1038/s41467-017-02001-5). While *Krt8* is strongly expressed in luminal cells as expected, low-level expression in basal populations has been reported in scRNAseq (e.g. at Bach et al) and may reflect transitional or mixed cell states. In classifying cell populations, we relied on a combination of established markers rather than single-gene assignments, aiming to maximise specificity and robustness. We believe this approach supports the reliability of our mRNA-seq interpretations, though we acknowledge the inherent complexity of cellular identity in lactating tissue and the possibility of some contamination given that these are bulk purified.

P5-L26-28: but in lactation there is less proliferation and more apoptosis so most likely less milk producing cells?? The suckling assay performed on WT pups cross fostered to either WT or *Zfp57^{-/-}* dams, showed no difference in milk let-down or overall volume, as indicated by comparable pup weight gain following separation and reunion (Figure 4G). These findings suggest that, despite potential cellular alterations, milk production remains functionally intact.

P5-L36-39: The qPCR results are not reflected in the RNA-seq data? any explanation? The qPCR analysis was performed on whole mammary tissue, which reflects the combined gene expression of all cell types present. In contrast, the RNAseq data were generated from sorted cell populations, capturing transcriptional profiles from specific subtypes rather than the entire cellular context. The RNAseq data from sorted populations revealed clear enrichment of pathways and biological processes consistent with our histological and immunofluorescence findings, including mammary epithelial proliferation and changes in milk composition, supporting the relevance and sensitivity of this approach in capturing key aspects of the tissue-level phenotype.

P5-L41: Caution; levels of STAT5/6 are not the main indication of functionality, more important is if they are transducing the signal for Prl or other cytokines as indicated by their phosphorylation and nuclear translocation.

We rephrased this sentence to reflect that we refer to transcript levels and not protein levels or functionality.

P6-L1: “results in precocious development of nulliparous mammary glands” hyper

branching and possibly early alveologenesis but not as far as expression of milk protein genes. As mentioned before hyperplasia might be better term?
are hormone levels the same?

We believe the phenotype is better characterised as precocious development rather than hyperplasia. as total cell numbers did not differ significantly between WT and *Zfp57^{-/-}* glands at any time point. Our findings suggest that specific developmental processes, such as ductal branching, are initiated earlier than expected in nulliparous glands, but these changes are neither sustained nor exacerbated at later stages. Furthermore, circulating levels of oestrogen and progesterone were not significantly different between groups (Supplementary Figure 6C–D), which makes a hormonally driven hyperplastic response less likely.

P6-L26: “dams Compared to WT’s”, technically that is compared to WT dams bred with +/- or -/- males, and not WT x WT. Paternal effects of in utero development and gestation cannot be ruled out. Yes, we corrected the text accordingly and tried to emphasise this point throughout the revised manuscript.

P7-L5: Figure 4G what is missing here are m-/+ pups on WT dams.
In this experiment, we focused on cross-fostering WT pups to both WT and *Zfp57^{-/-}* dams, in order to specifically dissect the maternal contribution to the observed phenotype, independent of pup genotype. This approach effectively uncouples maternal and offspring effects of the *Zfp57*. The data in Figures 5 and 6 include additional analysis of *Zfp57^{m/+}* pups in the growth and metabolism contexts.

Supplemental Figure 7AB: it would be helpful to depict the curves in A and B in a way that it is easy to compare how the same pups perform on Dams with a different genotype and use very different colors for each, the grays and pinks with different symbols and hatching is extremely hard to decipher. What one wants to establish here is that pups with the same genetic/epigenetic background perform differently when CF on WT vs null dam, so graphs should compare that, have CF to WT and CF to null in the same graph: add graphs for these comparisons.
We have revised Supplemental Figure 7A-B to combine the relevant comparisons as suggested.

Can you add statistics on the growth curves to show when the weights start differentiating or are there only statistical differences upon weaning? We have added statistical analyses to the growth curves, showing that differences emerge during the lactation period.

Supplemental figure 8: 8H check legend on graph with legend, 8I needs legend on graph We have amended the legend accordingly.

Reviewer #2 (Remarks to the Author):

It is critically important that the reader understands that the offspring are heterozygous for the mutation but they are not identical.

As stated by authors in response to this point raised in initial review “The reviewer is correct that *Zfp57^{m/+}* and *Zfp57^{+/-}* offspring are genetically identical, but because of a requirement for maternal ZFP57 in the oocyte (but not paternal ZFP57 in the

sperm) to maintain imprints after fertilisation, they are epigenetically distinct.”

Any phenotypic differences observed at birth and beyond between *Zfp57^{m/+}* and *Zfp57^{+/-p}* pups could be due to these intrinsic differences and not due to the maternal environment. It is not possible with this experimental design to tease apart these two factors i.e. in utero environment v intrinsic epigenetic differences. This does not detract from the findings of a lactation phenotype with metabolic consequences for offspring, but it is incorrect to conclude that “The study identifies ZFP57 as a major regulator of both pre and postnatal resource control”.

Thank you for raising this. Our group has extensively studied the role of *Zfp57* in maintaining DNA methylation at imprinted control regions (Takahashi et al., *Cold Spring Harb Symp Quant Biol* 2015, doi:10.1101/sqb.2015.80.027466; Takahashi et al. *Genes Dev* 2019, doi:10.1101/gad.320069.118; Shi et al., *Epigenetics Chromatin* 2019, <https://doi.org/10.1186/s13072-019-0295-4>) which demonstrate that the maternal-zygotic mutant is the genotype where most imprints are lost in contrast to the zygotic mutants (the genotype of the mothers used here) where only partial methylation and selective disruption of imprinting occurs only at *Zac1*, *Snrpn* and *Rasgrf1*.

We agree that it is essential to clarify the distinction between *Zfp57^{m/+}* and *Zfp57^{+/-p}* offspring which have the potential to be epigenetically different, however, we have previously characterised the epigenetic differences at imprinted genes and found that *Zfp57^{+/-p}* have normal methylation imprints (Li et al. DOI 10.1016/j.devcel.2008.08.014). Consistent with this, there is no significant difference in the expression of imprinted genes in *Zfp57^{+/-p}* compared to WTs (Supp Fig 6A). Our previous data has shown that the *Zfp57^{m/+}* are epigenetically different compared to *Zfp57^{+/-p}* but in a very minor sense, they have only partially lost their imprints at *Snrpn*, from 50% to 30% (Takahashi et al., doi:10.1101/sqb.2015.80.027466). This may explain the increase in expression of *Snrpn* compared to WTs we describe (Supp Fig 6A).

Other effects on imprinted genes in the *Zfp57^{m/+}* are therefore not explained by epigenetic differences between the *Zfp57^{m/+}* and *Zfp57^{+/-p}*, and could be either a cause or a consequence of their phenotype which we interpret as being conferred by the maternal environment.

Major epigenetic difference at imprints occurs in offspring in which both the maternally-inherited and paternally-inherited and oocyte-derived ZFP57 have been deleted.

Nonetheless, we have amended the manuscript accordingly to ensure this is clearly articulated and we have toned down the title.

Importantly, many of the phenotypes we describe are independent of the pups' epigenotype. For example, we observe precocious mammary gland development and altered gene expression in *Zfp57^{-/-}* dams, along with abnormal milk composition (including reduced phosphatidylcholines and increased oxidised triglycerides). These changes are present regardless of the pup genotype and reflect an intrinsic role for ZFP57 in the maternal regulation of postnatal nutritional support. Moreover, cross-fostering experiments further support this conclusion: WT pups fostered by *Zfp57^{-/-}* dams exhibit reduced weight gain compared to those nursed by WT dams (Figure

5B), indicating that maternal ZFP57 deficiency compromises lactational support independently of pup genotype.

Our conclusion that ZFP57 acts as a regulator of both prenatal and postnatal resource control is therefore well supported by the data. Nonetheless, we now state more explicitly that some observed phenotypes could stem from epigenetic differences between heterozygous offspring, and we have revised the discussion to clarify the interplay between epigenotype and environment.

Minor points

P2 line 15

“During pregnancy, hormonal signals such as progesterone and prolactin, drive mammary gland differentiation”
include placental lactogen **Done**

P3 line 12

“reduction in Phlda2 levels which may affect maternal care³²”

This reference refers to Phlda2 in the placenta not the hypothalamus **Thank you for this comment, we have removed this reference and statement.**

P4 line 11

“perturbed expression levels of imprinted genes does not result from loss of ZFP57-mediated imprinting control ”

This cannot be decisively concluded from the data presented. It could be that some changes are due to LOI.

We agree, the text has been amended to refer specifically to the 15 differentially-expressed imprinted genes.

P6 line 8 and line 27

“In-utero environment of Zfp57^{-/-} dams impacts offspring birth outcomes ”

Two issues: This new subtitle/statement could be misunderstood to suggest that the environment the mutant females experienced when they were in utero effects their pups' outcomes. However, the authors are referring to offspring's in utero environment. Even reworded - this is statement is not correct. The authors cannot conclude the pup phenotype is due to the maternal environment because both the dams and the pups are mutant in this model - see my next comment.

We agree the original subtitle is ambiguous and could be misinterpreted. We have revised it to clarify that maternal ZFP57 deficiency impacts the observed birth outcomes. We have updated the text to reflect that the phenotype may not be solely attributed to the maternal environment.

P6 line 15

“Both crosses yield genetically identical offspring of a single genotype”

As highlighted in first review and also indicated by authors in their response, the offspring are genetically identical but epigenetically distinct. This statement must be included in their results section and their discussion because it is critically important for the interpretation of the findings.

Thank you. We have now included this clarification explicitly in both the results and discussion sections to ensure accurate interpretation of our findings.

Line 27 “This indicates that the in-utero environment in *Zfp57*^{-/-} dams affects *Zfp57m*^{-/+} pups.”

Again, the authors cannot make this statement - offspring are genetically identical but epigenetically distinct

We have amended the manuscript accordingly.

Line 28

significantly upregulated/ downregulated

These terms can only be used if the authors have excluded gene changes resulting from changes in cellular composition. Use higher/lower expression in bulk RNA from whole placenta.

We corrected the phrasing of this sentence.

Line 43 “Taken together, these findings suggest (?reveal/?demonstrate)that *Zfp57m*^{-/+} pups born to *Zfp57*^{-/-} dams show increased birth weight and poor perinatal survival compared to heterozygous pups born to WT dams. “

Instead of heterozygous use *Zfp57*^{+/-p} to be consistent

Thank you, we have corrected this.

P7, lines 1 and line 19

The authors repeat the first line. We edited the first sentence.

P7, lines 1-8

Need more clarity on the experimental design. Suggest “We compared WT and *Zfp57*^{-/-} dams raising their own *Zfp57m*^{-/+} and *Zfp57*^{+/-p} biological litters with fully wildtype litters, and included a cross-fostering paradigm where WT and mutant pups were fostered to dams of the three conditions.”?

Thank you for the suggestion. We agree that clarifying the experimental design is important for reader understanding. We have revised the sentence to more clearly describe the comparisons made, including the use of biological litters and cross-fostering across WT and *Zfp57*^{-/-} dams.

P7 line 1 to P9 line 33

Appreciate the additional amount of work undertaken in the new cross fostering experiment. This is the critical experiment and key to interpreting the findings. The experimental design should be clearer in text and M&M. Generally this section is hard to follow.

Thank you for this feedback, we have revised the materials and methods section of the manuscript to provide a clearer and more structured description of the experimental design and procedures. We hope this revised version improves clarity and better conveys the rationale and methodology underlying this key experiment.

M&M / figures

Presumably the authors have combined the original data from the non-fostered animals with the new data for the fostered animals ie two separate experiments? This needs to be clearly stated. And - if not performed concurrently - this should be acknowledged in discussion as a potential weakness.

To clarify, the original data included cross-fostering experiments, which were conducted simultaneously with dams raising their biological litters, ensuring consistent conditions across groups. In the revised version, we added additional

cross-fostering groups involving WT pups and WT or *Zfp57*^{-/-} dams. These were performed concurrently with the other experimental combinations and integrated into the revised figure. We have stated this in the revised methods section and clarified the timing of the experiments to ensure transparency.

Discussion

Needs to articulate more clearly that the heterozygous offspring are not identical – which may account for some phenotypes ie increased weight of *Zfp57*^{m-/+} at birth. We agree that it is important to acknowledge the differences between the heterozygous offspring. To clarify this point, we have added a paragraph to the discussion highlighting these distinctions and their potential implications for interpreting the observed outcomes.

Reviewer #3 (Remarks to the Author):

my comments have been sufficiently addressed.

Reviewer #4 (Remarks to the Author):

In this revised manuscript entitled, ZFP57 is a regulator of postnatal growth and life-long health, the authors present additional data about the role of Kruppel-associated box-containing zinc-finger protein (ZFP57) in the mammary gland, in the uterine environment and in the embryo. ZFP57 is a master regulator of imprinting that acts to recruit DNA methyltransferases lacking the targeting domains required to recognize imprinting control regions. Yet, it is not functioning in this canonical role, at least in the mammary gland. I am still unclear whether it functions canonically in the placenta or in the embryo. The manuscript is still confusing to read and not targeted to a general scientific audience. There are mistakes. It also lacks sentences that introduce the experiments conceptually and that synthesize the potential meaning of the results.

The major improvement to the manuscript was the addition of appropriate controls that allow the authors to better tease apart the role of the dam in the postnatal window. The most straight forward parts of this manuscript focus on the role of ZFP57 in the mammary gland (Figures 1-3) and the characterization of how abnormal milk produced by ZFP57^{-/-} glands (Figure 5C). The least straight forward focus on the in utero effects (Figure 4, 5, 6). Many interesting observations have been made, yet the authors fail to synthesize any mechanistic insight from their results. The support for the take-home message that ZFP57 regulates nutritional provision to offspring through two distinct mechanisms: prenatally, through genomic imprinting and postnatally through mammary gland and lactation-related genes is not clearly articulated in the results or discussion section. In my opinion, this is two different manuscripts about nutritional provisioning and trying to put them together into one manuscript does not serve either the data or the reader. Furthermore, without mechanistic insight it is difficult to see how this manuscript reveals impactful information. It is not new that childhood weight gain is a risk factor for adult health. The authors find, as others have, that early-life exposures can induce metabolic reprogramming that can have long lasting effects (Picó C, et al., Lactation as a programming window for metabolic syndrome. Eur J Clin Invest. 2021

May;51(5):e13482. doi: 10.1111/eci.13482.). They also find that prenatal developmental conditions can predispose offspring to long-term maladaptation, something that others have also shown (<https://doi.org/10.1017/S0954579412000764>).

We thank the reviewer for their thorough and detailed feedback. In this revision, we have made substantial efforts to improve the clarity of the manuscript. We have enhanced the conceptual framing of each experimental section and expanded the Materials and Methods, Results and Discussion.

While we acknowledge that early-life exposures and their long-term effects on offspring health have been previously described, we believe our findings offer a novel perspective. Most existing studies focus on environmentally driven maternal-offspring co-adaptation. By contrast, our data support a genetically encoded mechanism, driven by the maternal loss of ZFP57, that alters both prenatal and postnatal nutritional provisioning. We now articulate this concept more clearly throughout the manuscript and highlight how our results challenge existing models by providing evidence for a genetically encoded, rather than solely environmentally mediated, coordination between maternal physiology and offspring growth.

Below are comments concerning the new data, and a few observations from reviewing the manuscript again.

Figure 1-3 examines Zfp57 expression in five mammary cell populations. The authors use alphaSMA as a co-expressing marker for stromal cells but this doesn't work well in the mammary gland where alphaSMA marks basal cells. Here is an atlas of fibroblast gene expression during mammary gland development (<https://doi.org/10.1038/s44318-025-00422-3>); PDGFRalpha is a reasonable marker for mammary gland stroma. The FACS plots displayed in the Supplementary data (Supp Fig 4) do not show well isolated populations of cells; the gate calling appears non-standard (please see Shehata et al., doi: 10.1186/bcr3334.)

Thank you for this comment; we are familiar with the work conducted by Rosa Pascual and colleagues and appreciate the recommendation. In response, we have replaced the qPCR data with *Pdgfra* as a stromal marker as suggested. Regarding the FACS gating strategy, we follow the marker combinations described by Shehata et al. However, we note that their study used the C57BL/6J strain, whereas our experiments were conducted in mice on a 129aa genetic background. Strain-specific differences are known to affect gene expression, including cell-surface markers, which may influence both population distribution and the appearance of FACS plots. We have added the details on the mouse lines in the Methods section.

Figure 4 assesses the in-utero environment incorporating experiments suggested by the reviewers to disentangle the myriad of effects described in the original manuscript. This section is still very confusing. Below are my questions based on the understanding I gleaned after reading the section multiple times.

Here, the authors focused on “two prominent mouse crosses: WT females crossed to a Zfp57^{-/-} males, generating paternal heterozygous pups (Zfp57^{+/-p}), or ZFP57^{-/-} females crossed to WT males generating maternal heterozygous pups (Zfp57^{m-/+}). Both crosses yield genetically identical offspring of a single genotype,”

Ques: the crosses yield genetically identical offspring but are the epigenetically identical? In the introduction the authors said “Given that loss or partial loss of imprinting at most of the affected genes is found in *Zfp57* homozygous mutants, this indicates that imprinted gene regulation by ZFP57 is relevant in both embryonic and adult tissues.” Furthermore, a study has shown that showed “DNA methylation at a few imprinting control regions was partially lost without maternal *Zfp57* in *Zfp57* heterozygous mouse embryos derived from *Zfp57* homozygous female mice. This suggests that maternal *Zfp57* is essential for the maintenance of DNA methylation at a small subset of imprinted regions in mouse embryos.” (doi: 10.3389/fcell.2022.784128). Can the authors please clarify how this result might affect the interpretation of their data?

Our group has previously characterised *Zfp57*'s role in maintaining DNA methylation at imprinted control regions (Li et al., 2008; Takahashi et al., *Cold Spring Harb Symp Quant Biol* 2015. doi:10.1101/sqb.2015.80.027466; *Genes Dev* 2019, doi:10.1101/qad.320069.118; Shi et al., *Epigenetics Chromatin* 2019, <https://doi.org/10.1186/s13072-019-0295-4>). These studies demonstrated that maternal-zygotic *Zfp57* knockouts (MZ^{-/-}) result in severe loss of methylation at multiple imprinted loci and embryonic lethality, while zygotic knockouts (Z^{-/-}) exhibit partial perinatal lethality and more selective effects on imprinted regions.

In the present study, the *Zfp57*-deficient dams are zygotic knockouts (Z^{-/-}), not maternal-zygotic (MZ^{-/-}), and therefore the embryos from which these dams arise, retain a maternal supply of ZFP57 during early development. As shown in Takahashi et al., this results in partial methylation loss, most notably at *Zac1*, *Snrpn*, and *Rasgrf1*, but it is substantially less severe than in MZ^{-/-} embryos that lose the majority of their imprints and die (see Fig. 3, blue vs. red lines in Takahashi et al. 2015).

Our group's previous data has shown that the *Zfp57*^{m/+} are epigenetically different compared to *Zfp57*^{+/-p} but in a very minor sense, they have only partially lost their imprints at *Snrpn*, from 50% to 30% (Takahashi et al., doi:10.1101/sqb.2015.80.027466). This may explain the increase in expression of *Snrpn* compared to WTs we describe (Supp Fig 6A). Other effects on imprinted genes in the *Zfp57*^{m/+} are therefore not explained by epigenetic differences between the *Zfp57*^{m/+} and *Zfp57*^{+/-p}, and could be either a cause or a consequence of their phenotype which we interpret as being conferred by the maternal environment. Major epigenetic difference at imprints occurs in offspring in which both the maternally-inherited and paternally-inherited and oocyte-derived ZFP57 have been deleted.

We have added clarification of these distinctions to the Materials and Methods to ensure readers can more easily follow the genetic and epigenetic context of our breeding strategy. We also emphasised these distinctions in the Discussion.

Supp Fig 6a. In the text, the description of the data is “The hypothalamus is involved in feeding circuitry and suckling³⁵, therefore, we next quantified changes in imprinted gene expression affected by *Zfp57* in this region. *Zfp57*^{m/+} offspring exhibit a few minor changes in hypothalamic expression of imprinted genes compared to WTs, including *Rasgrf1*, *Snrpn*, *Nnat* and *Nespas* (Supplementary

Figure 6A).” What is shown, but not stated, is that *Zfp57*^{+/-p} offspring do not display changes in gene expression. Please state clearly that this comparison shows changes in *Zfp57*^{m/+} offspring compared to WTs and *Zfp57*^{+/-p} offspring. Furthermore, I am not sure why the authors downplay these changes by describing them as minor when they are significant, especially if they potentially contribute to the suckling defect? Or do the authors have a different explanation for the suckling defect? In any case, the authors should be careful about calling significant differences minor or at least justify their conclusion

We have now clarified in both the results text and the legend to Supplementary Figure 6A that *Zfp57*^{+/-p} offspring do not exhibit changes in hypothalamic imprinted gene expression compared to wild-type, while some changes are observed in *Zfp57*^{m/+} offspring relative to both wild-type and *Zfp57*^{+/-p}.

We acknowledge that *Rasarf1* and *Nespa5* are differentially expressed in *Zfp57*^{m/+} compared to *Zfp57*^{+/-p} offspring, however, our previous work shows that there is no change in methylation at these loci (Takahashi et al.), therefore these changes could be downstream effects of their phenotype. Therefore, while the changes in gene expression are statistically significant, the extent to which they causally contribute to the observed suckling defect remains unclear. We now reflect this interpretation in the discussion.

Supp Fig 6e. The figure legend does not match the figure so this is confusing. In Panel E, why are the authors calling out “paternal het” and “*Zfp57*^{-/-}” in the panel legend? In the text, the description of the data is “*Zac1* and *Snrpn* were significantly upregulated in *Zfp57*^{m/+} placentas and *Igf2* and *Dlk1* were downregulated compared to *Zfp57*^{+/-p} animals (Supplementary Figure 6E).” Aren’t *Zfp57*^{m/+} placentas actually the placentas of the *Zfp57*^{-/-} dams? And doesn’t paternal het mean WT placenta? Apparent mismatch between text and figures is confusing to the reader. It would be terrific if the authors made an effort to be consistent, and therefore clearer, in the use their own terminology.

We amended the figure to maintain the terminology consistent with the rest of the manuscript.

To improve the clarity and readability of the manuscript, we also reassessed the contribution of the placental data to the overall findings. Given the potential for confusion and the fact that these observations are consistent with previously published reports (Takahashi et al., *Cold Spring Harb Symp Quant Biol* 2015, doi:10.1101/sqb.2015.80.027466; *Genes Dev* 2019; Jiang et al., <https://doi.org/10.1073/pnas.2005377118>), we have opted to streamline this section and focus the manuscript more directly on the novel aspects of our study.

We agree that both maternal genotype (*Zfp57*^{-/-} dams) and offspring epigenetics (*Zfp57*^{m/+}) can influence imprinted gene expression, and we now clarify this more precisely in the main text.

The authors perform cross-fostering experiments to uncouple maternal and offspring perinatal phenotypes and conclude that “This indicates that *Zfp57*^{m/+} pups have impaired suckling.” I understand the fact that *Zfp57*^{m/+} pups born to *Zfp57*^{-/-} dams are heavier, which suggests this could be a placental effect, but do the authors think impaired suckling is a placental effect? What do the authors think is the explanation

for why *Zfp57*^{m/+} pups have impaired suckling (Figure 4G)? Is this due to the “few minor changes in hypothalamic expression of imprinted genes compared to WT, including *Rasgrf1*, *Snrpn*, *Nnat* and *Nespas* (Supplementary Figure 6A).” and, therefore, an offspring factor due to the fact that the “hypothalamus is involved in feeding circuitry and suckling”, which they point out earlier in the previous paragraph? If so, can the author link these data together better for the reader? Wouldn't this be considered an offspring effect, even though this section is ostensibly about in utero effects? Alternatively, if hypothalamic expression is not the cause (because the changes are minor (but significant)), are the authors concluding the effect is due to placental imprinted gene expression (Supplementary Figure 6E) – an in utero effect? Authors: please clarify your thinking. And, if there is an offspring effect, consider modifying the section heading (“In-utero environment of *Zfp57*^{-/-} dams impacts offspring birth outcomes”).

In this model, where the *Zfp57*^{m/+} pups develop in the *Zfp57*^{-/-} dams, the observed phenotypes (including increased birth weight, impaired suckling) may result from either a maternal in-utero effect, intrinsic offspring epigenetic effects or a combination of the two. Yet, *Zfp57*^{m/+} offspring which differ epigenetically from *Zfp57*^{+/-}, exhibit only a 20% reduction in the methylation of *Snrpn*. This likely accounts for the increased hypothalamic expression of *Snrpn* in *Zfp57*^{m/+} relative to WT (Supplementary Figure 6A). However, all other changes in hypothalamic imprinted gene expression, cannot be attributed to this epigenetic difference and may instead represent either a cause or a consequence of their phenotype, which we interpret as being driven by the maternal deficiency of *Zfp57*. To better reflect this nuance, we have revised the subheading and the relevant text to clarify our interpretation.

The point is that the authors test a number of hypotheses in this section, yet they conclude their paragraphs with a general statements about the findings (“Taken together, these findings suggest that *Zfp57*^{m/+} pups born to *Zfp57*^{-/-} dams show increased birth weight and poor perinatal survival compared to heterozygous pups born to WT dams.” “This indicates that *Zfp57*^{m/+} pups have impaired suckling. ”), without a concluding statement about the biological mechanism that might explain these data. This lack of link to mechanism is a major problem with this manuscript: at least some data are here, but any link to biological mechanism is tenuously drawn.

Our RNAseq data reveals a range of differentially expressed genes which are being affected by the knockout.

Most of these differentially expressed genes are not imprinted, and among those that are, we do not observe patterns consistent with loss of imprinting. This suggests that the underlying mechanism may extend beyond canonical imprinting and could involve early developmental disruptions or imprinting-independent effects. This represents a novel mechanistic insight into ZFP57 function.

Notably, this is the first study to investigate *Zfp57* in the context of mammary gland biology. Our cross-fostering experiments, which include wild-type pups, demonstrate that *Zfp57*^{-/-} dams produce milk that negatively impacts both short- and long-term offspring health. These findings point to a previously unexplored role for *Zfp57* in postnatal metabolic regulation via maternal tissues.

Given that *Zfp57* regulates a broad epigenetic network, rather than a single target gene, pinpointing a singular causal mechanism within this global knockout model is inherently complex. However, we have clearly stated in the discussion that this limitation can be addressed in future studies using tissue-specific and temporally controlled knockouts. Overall, our findings provide an important foundation for understanding how maternal epigenetic regulators like *Zfp57* influence offspring development and health, and open new directions for mechanistic exploration.

The description of cross-fostering was very confusing:

“The original litter and the dam were removed from the dam's cage, and the cross-fostered pups were gently rolled in the nesting material of the fostering dam. Subsequently, the original dam was returned to her home cage with a new litter, which was weighed every 1-2 days until weaning. The number of pups was normalised to the size of the smaller litter.”

What is the new litter that goes back with the original dam? Are these the cross-fostered pups that have been rolled in the nesting material? If so, then they would be referred to as THE new litter. A new litter suggests a different, unspecified, litter. If you want the reader to understand your experiments, the authors should consider carefully editing their manuscript for clarity. **Thank you for this comment. We have now revised the phrasing of this methods section as suggested.**

Figure 5 examines the offspring over time. A) The rationale for separating the data based on sex of the pup is not clear. In any case, the data appear similar. It would be helpful either in the text or in the Figure legend to remind the reader of the “two prominent mouse crosses” that I surmise are still the focus of this panel (“WT females crossed to a *Zfp57*^{-/-} males, generating paternal heterozygous pups (*Zfp57*^{+/-p}), or *ZFP57*^{-/-} females crossed to WT males generating maternal heterozygous pups (*Zfp57*^{m-/+}).”). The description “we monitored the growth of *Zfp57*^{m-/+}, *Zfp57*^{+/-p} and WT pups raised by *Zfp57*^{-/-} and WT mothers.” is not particularly clear, especially since CF experiments have already been introduced. It is very helpful that the authors included the appropriate WT/WT control.

We separated the data by sex because male and female pups exhibit different growth trajectories, with males typically being heavier, thus making it important to present sex-specific plots to avoid confounding comparisons.

To add clarity regarding the breeding crosses and maternal context, we have updated the figure legend to explicitly describe the genetic crosses and clarified in the main text that these pups were raised by their biological mothers.

B) The rationale for looking only at male offspring is not clear (Legend: Weights of WT, *Zfp57*^{m-/+} and *Zfp57*^{+/-p} male offspring at weaning). Given the number of statistical tests performed here, it is necessary for the authors to indicate the comparison group so when the text reads “*Zfp57*^{m-/+} pups exhibited significantly increased weight gain during lactation when raised by or cross-fostered to *Zfp57*^{-/-} dams.” the reader needs to know the comparison group so that they can see the significant change on the graph. This is a very complicated panel so I may be missing something here, but I see that *Zfp57*^{m-/+} pups raised by *Zfp57*^{-/-} dams and CF to WT have significantly increased weight gain during lactation compared to WT pups raised by WT dams or WT pups raised by WT dams and CF to WT dams or

WT pups raised by WT dams and CF to *Zfp57*^{-/-} dams. But, *Zfp57*^{m/+} pups raised by *Zfp57*^{-/-} dams do not have significantly increased weight gain during lactation compared to *Zfp57*^{m/+} pups raised by *Zfp57*^{-/-} dams and CF to WT or compared to *Zfp57*^{m/+} pups raised by *Zfp57*^{-/-} dams and CF to *Zfp57*^{-/-} dams. That is, the possibilities in this graph are many. Maybe the authors are referring to a different comparison, but how would we know? The bottom line is that the authors need to clarify the language and indicate on the panel (maybe by using characters: #*) and in the text the comparisons, because it is not as if one can simply look at the figure and easily see the relevant comparison. Also, I don't think the reduction in weight at weaning for the *Zfp57*^{+/-p} pups CF to *Zfp57*^{-/-} dams can be classified as severe unless the authors can find a reference for this classification (PMCID: PMC3750667 PMID: 24209967).

Thank you for this thoughtful and detailed comment. We agree that the complexity of Figure 5B, with its multiple genotypes and fostering conditions, can make interpretation challenging. To improve clarity, we have now explicitly stated the relevant comparison groups in the figure and legend, and clarified the comparisons in the main text. We have also added reference markers in the panel to guide the reader through the key comparisons.

Regarding the decision to show only male offspring in the main figure: we analysed both sexes independently and observed consistent trends across males and females. To reduce figure complexity and maintain readability, we chose to present male data in the main figure and moved the female data to Supplementary Figure 7C. This has now been clearly stated in the text.

The main findings we aim to convey are: (1) *Zfp57*^{m/+} pups gain more weight during lactation compared to WT or *Zfp57*^{+/-p} pups, regardless of whether they are raised by *Zfp57*^{-/-} or WT dams; and (2) any pup nursed by a *Zfp57*^{-/-} dam shows attenuated postnatal growth, highlighting a maternal effect.

In the case of the *Zfp57*^{m/+} pups that accumulate excessive weight – this is beneficial, suggested co-adaptation between mother and offspring.

We have removed the word severe from the description of failure to thrive and amended the text to reflect these messages.

They finish the section by concluding that *Zfp57*^{-/-} dams are compromised in their ability to support the normal postnatal growth of offspring. This is followed up by an analysis of milk and the data support the authors' conclusion that *Zfp57*^{-/-} mothers produce abnormal milk. The authors conclude that mother-offspring co-adaptation is associated with milk composition, but it would be helpful if they explain their thinking in terms of mother's genotype, in utero environment, pup epigenetics (which is generally dismissed) and development.

Our conclusion is based on multiple lines of evidence: *Zfp57*^{-/-} dams consistently attenuate postnatal growth of offspring and produce milk with abnormal composition: elevated oxidised triglycerides and reduced phosphatidylcholine levels. These biochemical changes likely reflect the disrupted mammary gland development and altered gene expression observed in *Zfp57*^{-/-} mammary cells. The effects of *Zfp57* loss in the dam are not limited to the lactation period, but may originate from developmental disruptions in the mammary gland during pregnancy. These changes

could impact milk composition and the dam's ability to support healthy offspring growth.

Furthermore, the enhanced postnatal weight gain observed in *Zfp57^{m/+}* pups appears to be an intrinsic feature and is attenuated when these pups are nursed by *Zfp57^{-/-}* dams. This supports the idea of mother-offspring co-adaptation, whereby pups benefit more from milk produced by their biological mother or a dam sharing her genotype. We now clarify this reasoning more explicitly in the discussion.

Figure 6 examines life-long health outcomes. The authors do a better job explaining their at least some of their comparisons. But, there are many comparisons to be made, and the authors do not call out all of them so again being explicit and indicating on the panel that comparisons that are being called out in the text would be very helpful. I also suggest that the salient results be tabulated in a way that makes them accessible to the reader. Sorting through the information, the authors find some differences and make conclusions concerning pup metabolic changes in response to the maternal environment and mother-offspring co-adaptation

We have now marked the key statistical comparisons directly on the figure using characters (#, €, § and ∞) to help guide the reader to the relevant findings described in the text.

To further aid accessibility, we have also included a new supplementary table (Supplementary Table 4) that summarises the main long-term health outcomes across groups.

The limitation to the study pointed out in the discussion of not using a Cre-line to specifically delete *Zfp57* in mammary cells because other organs will retain their *Zfp57* levels misses the point about learning whether the alterations observed in the mammary gland are intrinsic to the gland (i.e. “the stemness of luminal progenitor cells during development”) or systemic influences (i.e. hormone levels, which the authors measured and found no differences in). Obtaining this level of understanding could help the authors draw mechanistic conclusions, something that is lacking in this manuscript. Thank you for this suggestion, we have amended this sentence in the discussion accordingly.

Minor:

“Data show” for n/v agreement because the word data is a plural noun Corrected.

Spell and punctuation check: there are many examples. Here is one: expressing (Fig Legend 1) Thank you for pointing this out. We have corrected this typo and carefully proofread the manuscript to address any additional spelling or punctuation errors.